# Spatial transcriptomics identifies molecular niche dysregulation associated with distal lung remodeling in pulmonary fibrosis

Annika Vannan [1,13], Ruqian Lyu[2,3,4,13], Arianna L. Williams [1], Nicholas M. Negretti [5], Evan D. Mee[1], Joseph Hirsh [5], Samuel Hirsh [5], Niran Hadad[1], David S. Nichols[6], Carla L. Calvi[6], Chase J. Taylor [6], Vasiliy. V. Polosukhin[6], Ana P. M. Serezani[6], A. Scott McCall[6], Jason J. Gokey[6], Heejung Shim[3,4], Lorraine B. Ware [6,7], Matthew J. Bacchetta [8], Ciara M. Shaver[6], Timothy S. Blackwell[6,9,10,11], Rajat Walia [12], Jennifer M. S. Sucre[5,9], Jonathan A. Kropski [6,9,10,14], Davis J. McCarthy [2,3,4,14] & Nicholas E. Banovich [1,14] ✉

Large-scale changes in the structure and cellular makeup of the distal lung are a hallmark of pulmonary fibrosis (PF), but the spatial contexts that contribute to disease pathogenesis have remained uncertain. Using image-based spatial transcriptomics, we analyzed the gene expression of 1.6 million cells from 35 unique lungs. Through complementary cell-based and innovative cell-agnostic analyses, we characterized the localization of PF-emergent cell types, established the cellular and molecular basis of classical PF histopathologic features and identified a diversity of distinct molecularly defined spatial niches in control and PF lungs. Using machine learning and trajectory analysis to segment and rank airspaces on a gradient of remodeling severity, we identified compositional and molecular changes associated with progressive distal lung pathology, beginning with alveolar epithelial dysregulation and culminating with changes in macrophage polarization. Together, these results provide a unique, spatially resolved view of PF and establish methods that could be applied to other spatial transcriptomic studies.

The human lung is structurally complex, with a diversity of specialized epithelial, stromal and immune cells having specific functional roles in anatomically distinct locations. Large-scale changes in the structure and cellular makeup of the distal lung are a hallmark of pulmonary fibrosis (PF) and other chronic lung diseases[1]. PF is a progressive syndrome that can occur in the setting of known environmental exposures, systemic disorders, monogenic syndromes or can be idiopathic. Idiopathic pulmonary fibrosis (IPF) remains the most common and severe form of PF; most patients succumb to their disease or require lung transplantation within 3–5 years of diagnosis, and

available antifibrotic treatments only modestly slow the inexorable decline of lung function[2,3].

A hallmark of histopathologic findings in the lungs of patients with IPF (described as 'usual interstitial pneumonia'[4]) is spatial heterogeneity, where extensively remodeled regions can be found immediately adjacent to relatively preserved alveolar architecture. This spatial variability of pathology has been hypothesized to represent asynchronous disease evolution in the lung ('temporal heterogeneity'). In addition to the spatial heterogeneity of pathology, IPF lungs are characterized by the following: 'proximalized epithelial metaplasia', wherein cell

types typically found in conducting airways are observed in the distal lung epithelium; development and accumulation of cystic-appearing structures filled with mucus ('honeycomb cysts'); and the emergence of 'fibroblastic foci' (subepithelial collections of fibroblasts), which have been speculated to represent the 'leading edge' of disease pathology in the lung[5]. Along with genetic evidence linking IPF susceptibility to the lung epithelium[6–9] and data from experimental models, these classical histopathologic features support the prevailing model of IPF pathogenesis[10], whereby chronic/recurrent injury to the distal lung epithelium results in dysfunctional alveolar repair and culminates in progressive fibrotic remodeling.

Although the cellular complexity and spatial heterogeneity of disease present challenges when using bulk-tissue methods for genomic analysis, single-cell approaches are well-suited for such investigations. Large collaborative studies using droplet-based single-cell RNA sequencing (scRNA-seq) have refined our understanding of the cellular makeup of the normal human lung[11–14] and highlighted dramatic changes in the cellular makeup and molecular programs in IPF lungs, including disease-emergent and disease-perturbed cell types and states[9,15–24]. The spatial heterogeneity of pathology implies that within a given IPF lung, multiple distinct pathologic programs may be simultaneously occurring in distinct spatial regions (niches); thus, it is critical to understand the spatial context within which cellular and molecular programs mediate disease pathogenesis. To this end, we used image-based spatial transcriptomics with subcellular resolution to investigate the evolution of alveolar niche dysregulation in idiopathic and other forms of PF.

## Results

### Diverse cellular landscape of the lung

Using the Xenium platform, we profiled 343 genes across 45 lung tissue samples from nine unaffected donors and 26 participants who underwent lung transplant for PF, measuring 299,018,086 transcripts at subcellular resolution (Fig. 1, Supplementary Figs. 1 and 2 and Supplementary Table 1). Of the 26 PF participants, the most frequent diagnosis was IPF (n = 12, 46.2%). The majority of donors self-reported European ancestry (29; 82.9%), and 17 (48.6%) reported current or prior tobacco use. To enable spatially resolved single-cell analysis, we partitioned transcripts into cells using automated cell segmentation boundaries (Fig. 1). As cell segmentation remains a challenge in the field due to uncertainty around cellular boundaries and reliable assignment of transcripts to the correct cell, we focused only on transcripts overlapping nuclear boundaries. After this quality filtering, we retained 1,630,319 cells containing 121,794,939 transcripts (Fig. 1 and Supplementary Fig. 2). Single-cell analysis was carried out using modified versions of standard approaches (Methods).

We identified a total of 47 cell types, including considerably larger numbers of cell types underrepresented in scRNA-seq studies, such as endothelial and mesenchymal cells[9,12] (Fig. 2a, Supplementary Fig. 2 and Supplementary Table 2), better reflecting the cellular composition of the lung in disease states. For example, prior work using electron and light microscopy estimated the ratio of alveolar type 2 (AT2) to alveolar type 1 (AT1) cells in healthy distal lungs to be 1.68 (ref. 25). From these spatial transcriptomic analyses, we observed a mean AT2/AT1 ratio at 2.5 in unaffected samples, substantially closer to the anticipated ratio than what is found in scRNA-seq (AT2/AT1 ratio = 13.2; Fig. 2b). Canonical cell-type markers[9,11–13] localized in expected cells with generally high fidelity (Fig. 2c); nevertheless, even when restricting the analysis to transcripts within the nuclear boundary, some 'contamination' of gene expression from adjacent cells persists. This appears to be a fundamental property of spatial transcriptomic technologies, resulting from the performance of 2D cellular segmentation of 3D tissue (Fig. 2c and Extended Data Figs. 2–5). Assessing the spatial location of cells within specific lung structures (for example, airways, alveoli and vasculature) supported high confidence in annotated cell identities (Fig. 2d–i).

For example, we observe basal and multiciliated cells and their marker genes oriented toward the basement membrane and inner lumen of airways, respectively (Fig. 2d,e). Similarly, spatial data allowed us to identify fibroblast subtypes, including alveolar fibroblasts dispersed broadly across the lung, myofibroblasts around conducting airways and alveolar ducts, 'fibrotic' activated fibroblasts in patchy foci and 'subpleural' fibroblasts (expressing PLIN2) adjacent to mesothelial cells at the pleural surface (Fig. 2h,i).

To capture the regional heterogeneity of PF pathology, for a subset of participants, we profiled paired samples reflecting different degrees of pathologic remodeling, while from other participants we focused on 'transitional' regions representing a 'border' between severely remodeled and relatively preserved alveoli. Including a spectrum of diseases, pathology provides an opportunity to explore the molecular evolution of disease beyond that which can be captured in studies focusing on highly remodeled/end-stage samples[26,27]. PF samples were labeled as 'less affected' or 'more affected' based on the relative degree of overall pathology as described in the Methods (Supplementary Table 1). While our group and others have profiled samples from different regions within PF lungs using scRNA-seq, a key advantage of spatial transcriptomics is to have molecular measurements perfectly matched to the histologically assessed gradations of pathologic remodeling[9,18].

Focusing first on establishing the spatial context for cell types/ states that have been recently described in scRNA-seq, we observed that (consistent with other reports[18,28–30]) SCGB3A2+ epithelial cells were restricted to small/terminal airways in control lungs but found widely in remodeled areas of PF lungs where they frequently (but not exclusively) co-expressed SFTPC[18,29] (Extended Data Fig. 6a). Among mesenchymal cells, PI16+/MFAP5+ adventitial fibroblasts were largely associated with vasculature, while WNT5A+ myofibroblasts were observed in ductal regions and at the interface between the lamina propria and adventitia of conducting airways (Fig. 2h,i and Extended Data Fig. 6b,c). A spectrum of activated 'fibrotic' fibroblasts expressing varying levels of CTHRC1, FAP and/or POSTN was most concentrated in subepithelial regions underlying areas of extensive epithelial metaplasia but was also found more diffusely in some samples. COL15A1+ 'systemic venous' endothelial cells were found relatively widely in PF samples, particularly those with the most advanced structural remodeling (Extended Data Fig. 6d). Consistent with prior work from ourselves and others[18,19,26,27], we found KRT5−/KRT17+ 'aberrant basaloid' cells were located in close proximity to activated fibrotic fibroblasts (Supplementary Fig. 3). In contrast to one recent report[26], in these analyses, which include a broad range of pathologic remodeling, we found KRT5−/KRT17+ cells are most likely to be located adjacent to alveolar type rather than airway-type epithelial cells, although they are also found in proximity to epithelial cells found in small airways (Supplementary Figs. 3 and 4 and Supplementary Table 3). Furthermore, we found that KRT5−/KRT17+ cells were most abundant in less affected regions (Supplementary Fig. 5) and more likely to be adjacent to airway-type cells in severely remodeled regions.

We additionally performed a series of analyses comparing cellular and molecular changes between categorical designations and across a quantitative metric of 'percent pathology' and also characterized 27 specific PF histopathological features based on annotations by a clinician (Supplementary Note, Extended Data Fig. 7, Supplementary Figs. 5–9, Supplementary Tables 4–8 and Supplementary Data).

### Niche analyses reveal disease-emergent cellular interactions

We next sought to extend beyond a priori-defined pathological features to comprehensively define spatially integrated cellular/molecular units in the lung and characterize their evolution in disease. We used two complementary computational approaches to partition samples into regions of molecular and cellular similarity (that is, spatial 'niches'; Figs. 1 and 3a; Methods). First, we used a cell-based approach using Seurat v5 (refs. 31,32), building a local neighborhood based on spatial proximity and cell type annotation, followed by k-means clustering.

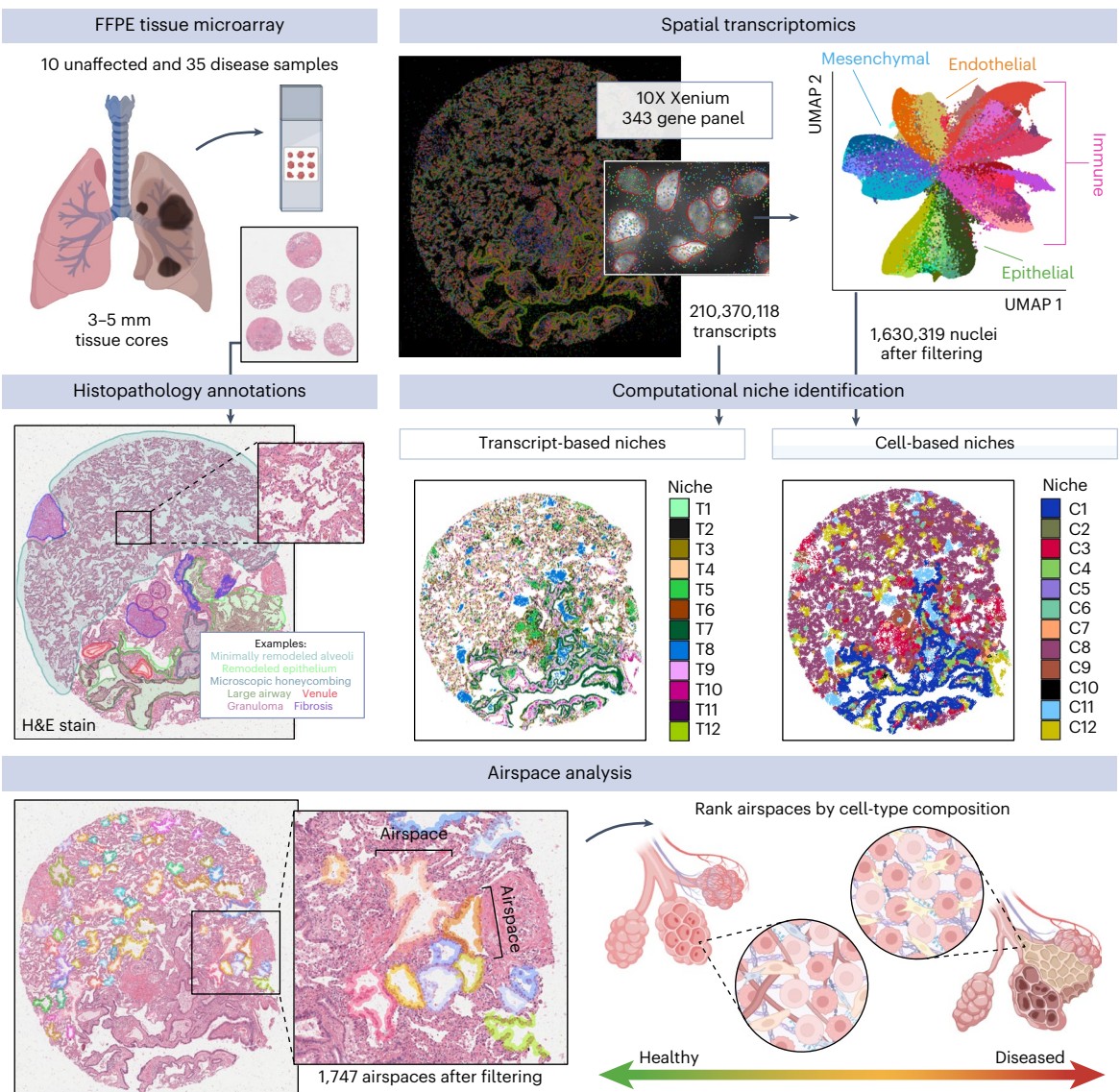

**Fig. 1 | Outline of spatial transcriptomics processing and analysis pipeline.**
In total, 45 lung tissue cores (3–5 mm) from unaffected and PF donors were
processed on the Xenium Analyzer instrument on a combination of four TMAs of
three to nine samples each and one additional replicate TMA with 17 additional
samples and the inclusion of cell-bound stains. We quantified the expression
of 343 genes at subcellular resolution using a custom panel. After filtering,
we retained 299,018,086 high-quality transcripts for identifying transcript
niches with GraphSAGE across all five TMAs. Of these, 210,370,118 transcripts
on the original TMAs 1–4 were used to build the initial GraphSAGE model. After
additional filtering, we annotated cell types for 1,630,319 segmented nuclei
across the endothelial, epithelial, immune and mesenchymal lineages. An
example sample shown is VUILD96LA (sarcoidosis diagnosis). The figure was
created with BioRender.com.

This approach was limited by the following two factors: (1) it is influ-
enced by the 'granularity' of cell annotation, and (2) it only uses data
from transcripts assigned to nuclei. To overcome these limitations,
we also developed a new approach to identify niches agnostic of cell
assignment by directly using transcript data. Using GraphSAGE[31], we
trained a graph neural network model based on the spatial location
of transcript data to aggregate local neighborhood information and
define an embedding space that provides a new representation for all
individual transcripts in the dataset. We then applied Gaussian mixture
models to cluster transcripts in the embedding space and identify
niches, assigning cells to these niches using a consensus approach. In
both analyses, we identified 12 niches (C1–C12 for cell-based and T1–T12
for transcript-based clustering), which displayed distinct gene expres-
sion signatures and cell-type compositions (Fig. 3b, Supplementary
Figs. 14–16 and Supplementary Table 9).

The objective of this approach was to, in an unbiased manner,
identify and characterize conserved relational patterns of cellular and
molecular features that represent recurring patterns of inhomogene-
ous cellular/transcript groupings that are often in close proximity to
one another (Fig. 3b and Supplementary Fig. 16). For example, airway
niches C1 and T7 and lymphoid niches T2 and C9 describe the close
spatial relationship between specific airway and lymphoid cell types,
respectively (Fig. 3b, Supplementary Fig. 3 and Supplementary Table 3).
The niches also capture more complex relationships between cells of
different lineages, including 'healthy' alveolar niches (T4; C8), which
include AT1, AT2, capillary cells and alveolar fibroblasts among other
cell types. Surprisingly, we also observed neutrophils in this niche,
likely a result of patchy acute inflammation in tissue from declined
donors. Unaffected samples were primarily defined by these healthy
alveolar niches, which had substantially lower relative abundance in PF

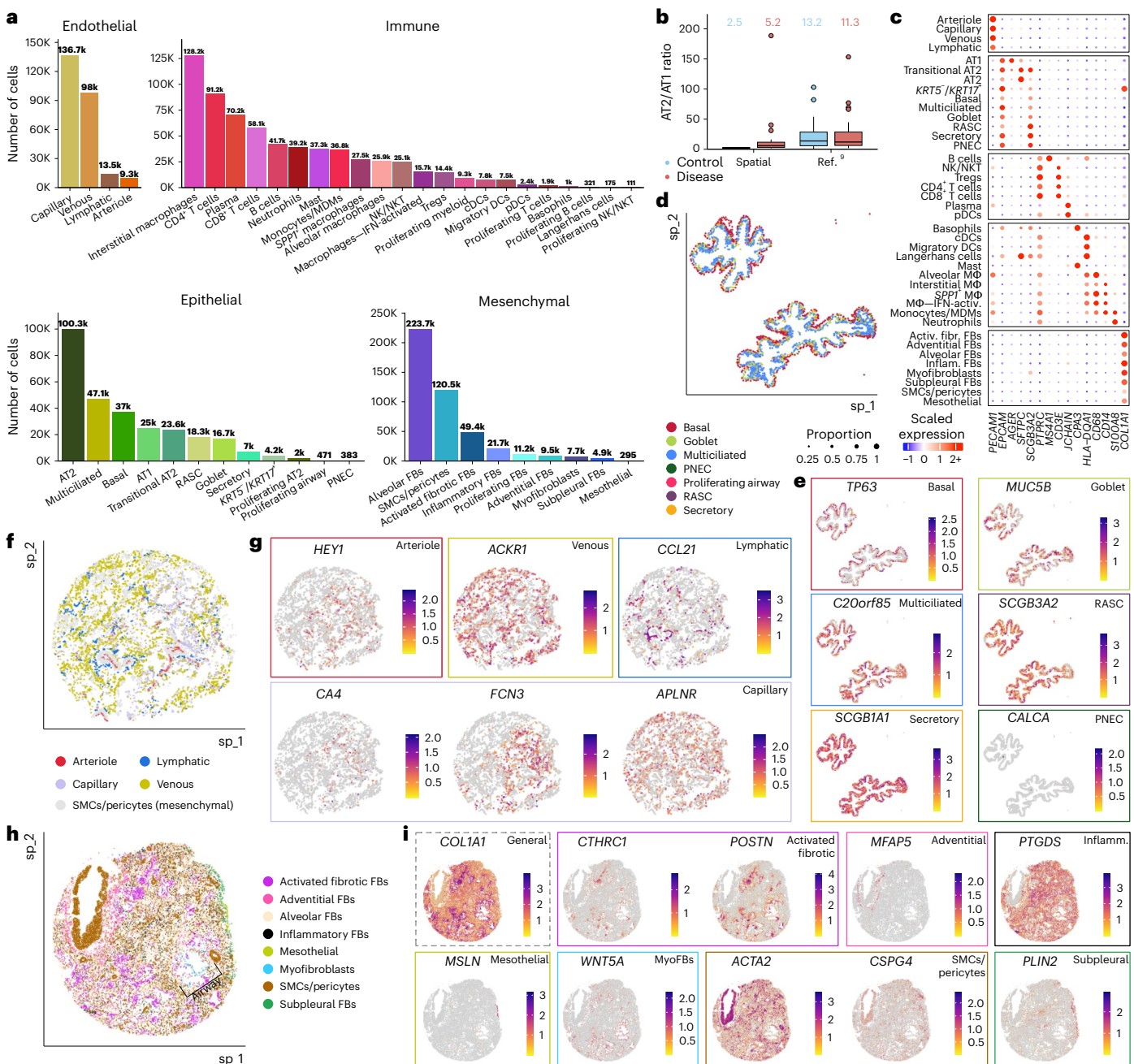

**Fig. 2 | Cell-type composition of unaffected and PF lung tissue determined using marker gene expression and spatial information. a,** Frequency of each cell type found across all samples. **b,** Ratio of AT2 versus AT1 cell counts across samples compared between the present spatial dataset and a recent scRNA-seq dataset[9]. Outlier samples are shown as individual points. Samples without counts for both AT1 and AT2 cells were not included in the analysis. Mean ratios across all samples are listed above. Boxplots show the median, with box hinges extending to the first and third quartiles and the whiskers extending to the largest (upper whisker) or smallest (lower whisker) value with a maximum of 1.5× IQR above and below. Sample sizes—spatial control, n = 10; disease, n = 34; ref. 9 control, n = 16 and disease, n = 22. **c,** Dotplot heatmap showing select genes used to annotate cell types in the dataset. See Extended Data Fig. 1 for the expanded version with additional genes. **d,e,** Example airways (TILD299MA; IPF) showing epithelial cells (**d**) and select marker genes (**e**). Basal cells reside on the outer edge of the airways,

while other airway cell types, particularly multiciliated cells, are oriented toward the inner lumen. **f,g,** Alveolar cell types (**f**) and marker gene expression (**g**) in a selection of alveoli (VUHD113; unaffected). **h,i,** Mesenchymal cells (**f**) and marker gene expression (**g**); VUILD106MA; IPF) proliferating fibroblasts omitted for clarity. An airway surrounded by myofibroblasts is marked by brackets. **e,g,i,** Boxes around plots are colored by the cell type they mark as indicated in **d,f,h**. The gray-dashed box for *COL1A1* (**i**) indicates this gene is a general fibroblast marker. Activ., activated; cDCs, classical dendritic cells; DCs, dendritic cells; FBs, fibroblasts; Fibr., fibrotic; IFN, interferon; Inflam., inflammatory; MΦ, macrophages; MDMs, monocyte-derived macrophages; NK, natural killer; NKT, natural killer T cells; pDCs, plasmacytoid dendritic cells; PNEC, pulmonary neuroendocrine cell; SMCs, smooth muscle cells; T_regs, regulatory T cells; IQR, interquartile range.

samples. Additionally, many niches were either disease-emergent or enriched in disease, including immune (T2, C7, C9), fibrotic (T6, T9, C4) and transitional epithelial niches (T3, C2; Fig. 3c and Supplementary Fig. 14). In particular, niches enriched for lymphocytic inflammation

(T2, C9) were much more prevalent in more remodeled samples, potentially implying a role in later (rather than early) disease pathogenesis. The transcript-based niche classification also resolved two distinct niches enriched for mesenchymal cells. T6 included activated fibrotic

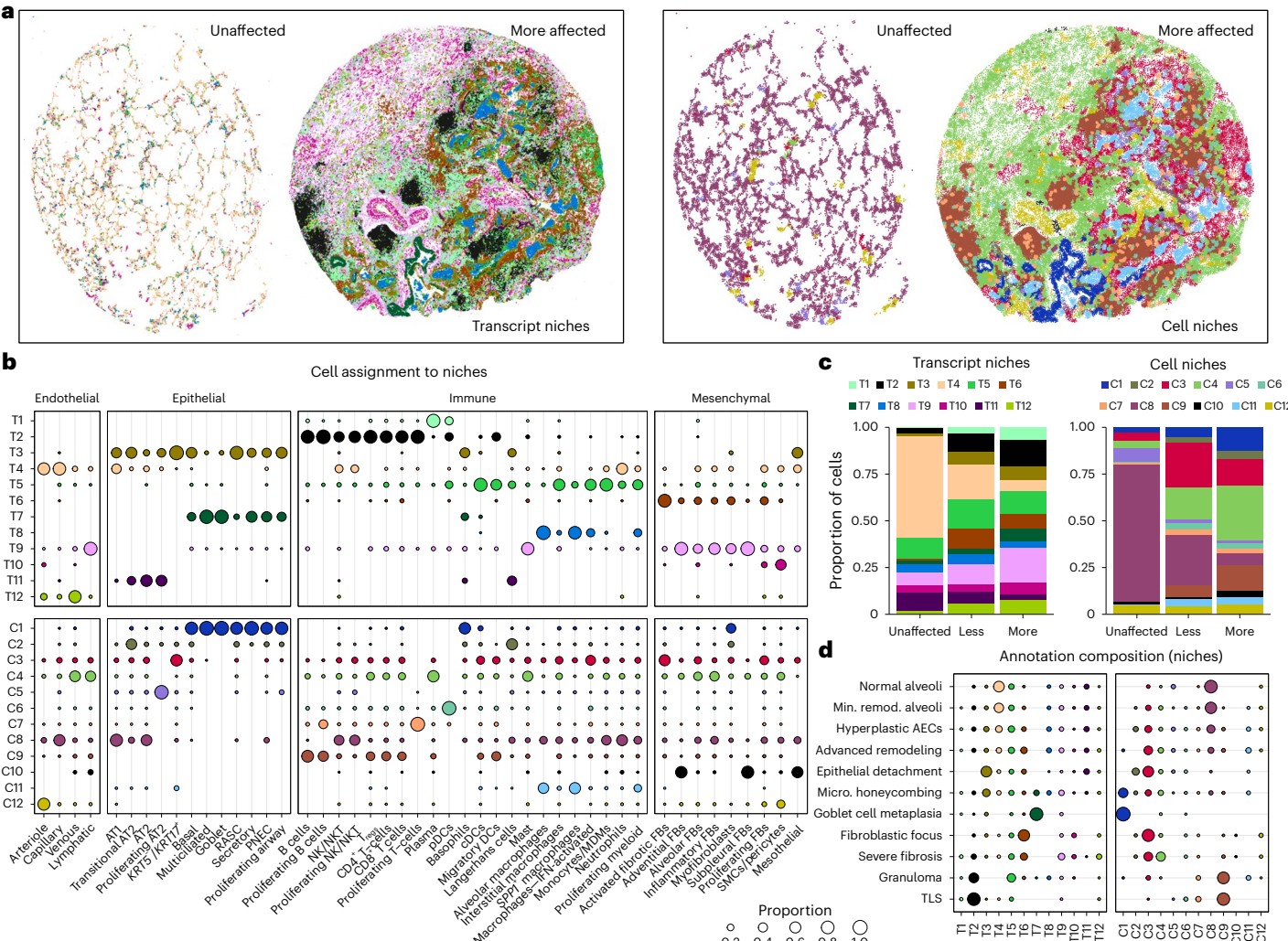

**Fig. 3 | Complementary spatial niche analyses provide comprehensive annotation of tissue remodeling in PF. a**, Representative examples from both unaffected and PF samples showing transcript- (left) and cell-based niches (right). VUHD113 and VUILD107MA (IPF diagnosis) are shown. For transcript niches, hexbin plots are shown (Methods). For cell niches, each point is a cell centroid. **b**, Cell assignment to transcript- (top) and cell-based niches (bottom), as a proportion of the number of cells of each type (each column sums to 1; columns indicated by gray lines). **c**, Bar plots depicting the total proportion of cells across the unaffected, less affected and more affected sample types assigned to each transcript and cell niche. **d**, The niche composition of select annotations, as a proportion of the number of cells across an annotation (each row sums to 1; rows indicated by gray lines). For **b** and **d**, proportions under 0.01 are not shown, and the proportion legend applies to both panels. See Supplementary Fig. 17b for expanded version (**d**) with all annotations listed. Different colors in **c** indicate niche colors for all panels. Micro., microscopic; Min., minimally.

fibroblasts and was largely restricted to regions around/adjacent to remodeled epithelium, whereas T9 was more broadly located, exhibited lower expression of active fibrotic markers and appeared to reflect more 'end-stage fibrosis'. Transitional epithelial niches were nearly absent from control samples but included the majority of epithelial cells from diseased samples. Strikingly, we observed that the niche composition of minimally remodeled regions of PF lungs more closely resembled that of more affected PF than control lungs. This unexpected finding implies that despite relative structural preservation, there is extensive molecular pathology in less severely remodeled regions and challenges the paradigm that spatial heterogeneity allows for true 'early disease' biology to be observed in relatively less remodeled areas of end-stage PF lungs.

We next set out to understand how these niches align with specific pathologic features (Supplementary Note). While a number of features included a heterogeneous mix of niches, some were predominantly or near-exclusively marked by a single niche (Fig. 3d). For example, granulomas and tertiary lymphoid structures (TLSs) were included

predominantly in the disease-enriched C9 and T2 immune niches, and multinucleated cells were marked by C11 and T8 (Fig. 3b,d and Supplementary Figs. 16 and 17). Of particular interest was our observation of patchy epithelial detachment from its underlying basement membrane (Fig. 4 and Extended Data Figs. 7–9). This feature, which we annotated as 'epithelial detachment', has also been described previously in IPF and other forms of PF and can be observed in histology elsewhere[33,34]. In our data, epithelial detachment was strongly associated with the transcript- and cell-based niches T3 and C3, respectively, which contain the vast majority of the detected *KRT5⁻/KRT17⁺* cells (96% and 64%, respectively; Figs. 3b,d and 4a). To further validate our findings, we generated matched sequencing-based spatial transcriptomic data using the Visium HD platform from two tissue sections that had already undergone Xenium-based profiling (Methods). Using a broader set of genes, we found highly concordant signal localization for both *KRT5⁻/KRT17⁺* cells and activated fibrotic fibroblasts (Extended Data Fig. 10 and Supplementary Table 10). As our annotations of epithelial detachment were not comprehensive by design (due to impracticalities around

comprehensive annotation of each sample; Supplementary Note), we postulated the niche analysis could aid in the rapid identification of additional examples of specific features. Indeed, directed by our niche analysis, we identified other regions exhibiting epithelial detachment in additional samples (Fig. 4b). These findings highlight the potential of spatial transcriptomic data to identify specific disease-associated pathologic features directly from the molecular data.

In one sample, we observed a striking region of dense fibrosis almost completely lined by the T3 and C3 niches (Fig. 4c,d). Adjacent to this area, we found multiple examples of epithelial detachment. We also identified structurally intact epithelium in the same transcriptionally assigned niche (Fig. 4e–g), which appear to be examples of subpathologic remodeling, suggesting we can identify molecular and cellular changes that precede histopathology. While the molecular signature of epithelial detachment was prominent at the interface between the putatively advancing fibrotic front (marked by activated fibrotic fibroblasts) and alveolar epithelium, the larger fibrotic region was marked by stable fibrotic niches and pan-fibroblast marker *COL1A1* (Fig. 4c). Furthermore, we observed open structures reminiscent of alveoli but completely devoid of epithelium (Fig. 4c,d). At a single time point, we cannot establish the origin of these 'remnant' alveoli, but one possible explanation is that they follow epithelial detachment at the fibrotic front. These observations raise the possibility that at least some *KRT5⁻/KRT17⁺* cells may represent a cell state that precedes epithelial detachment and other progressive pathology. We find that the detachment-associated niches are present across disease samples (with slightly increased proportion in less affected biopsies) but virtually absent in controls (Fig. 4h). Interestingly, while both niches contained a high proportion of *KRT5⁻/KRT17⁺* cells, the T3 transcript niche predominantly marked a suite of cell types associated with transitional alveolar epithelium, including transitional AT2 cells and respiratory airway secretory cells (RASCs), and often appeared near architecturally normal alveoli adjacent to active fibrotic fronts (Fig. 4 and Extended Data Figs. 7–9), while the C3 cell niche captured the relationship between *KRT5⁻/KRT17⁺* cells and activated fibroblasts expressing *CTHRC1* and *FAP* (Figs. 3b,d and 4) and marked both epithelial detachment and fibrotic foci. Indeed, within the C3 niche, we see a significantly increased likelihood for proximity between *KRT5⁻/KRT17⁺* cells and activated fibrotic fibroblasts and vice versa, providing a clear association between these cells that is mutually strongest in this detachment-associated niche (Fig. 4i, Supplementary Figs. 3, 18 and 19 and Supplementary Tables 2 and 11). Looking specifically at genes upregulated in *KRT5⁻/KRT17⁺* cells compared to other epithelial cell types, we observe similar patterns of expression to prior literature[18,19,26], including increased *COL1A1*, known PF blood biomarker *MMP7*, and transcription factor-encoding genes *SOX4* and *SOX9* (Fig. 4j). Notably, these and other specific *KRT5⁻/KRT17⁺* genes correspond to Gene Ontology (GO) pathways related to extracellular matrix organization along with cell adhesion and motility, suggesting that dysregulation of these gene programs may contribute to detachment from the basement membrane and adjacent cells.

### Disease-emergent macrophages accumulate in airspaces

In addition to identifying niches closely linked to specific pathologic features, our analyses also revealed widespread molecular pathology that did not specifically correspond to classical PF disease features. We identified a cell-based niche associated with macrophage accumulation within airspaces that was found across all disease samples irrespective of diagnosis (C11; Figs. 3b and 5a,b and Supplementary Fig. 14). Interestingly, the macrophages associated with airspace accumulation appear to include two mostly distinct populations, one marked predominately by *FABP4* and another by *SPP1* (Fig. 5c–e). Accumulation of *SPP1⁺* macrophages in airspaces is consistent with other studies in IPF[17,26]. In addition to airspace-accumulated macrophages, in PF samples with less substantial alveolar remodeling, we observed smaller

populations of *FABP4⁺* (alveolar) macrophages within alveoli and *SPP1⁺* macrophages in the interstitium (Fig. 5c). Despite accumulation within airspaces, *FABP4⁺* macrophages were only modestly increased in less affected samples as a proportion of the total macrophage population and became less frequent with increasing percent pathology (Fig. 5f, Extended Data Fig. 7 and Supplementary Fig. 7). Meanwhile, *SPP1⁺* macrophages are observed in control lungs but comprise a higher proportion of the total macrophage population in more affected samples, and increased expression of *SPP1* was associated with higher pathology scores (Fig. 5f, Supplementary Fig. 7 and Supplementary Tables 2, 7 and 8), suggesting an evolution of macrophage phenotypes characterizes progressive PF. We additionally confirmed the presence of both *FABP4⁺* and mixed *FABP4⁺/SPP1⁺* accumulations in distal airspaces using a broader set of genes with matched Visium HD data (Fig. 5g,h and Supplementary Table 10). In numerous participants, discrete regions of *FABP4⁺* and *SPP1⁺* macrophage accumulation were observed within the same 3–5 mm biopsy. It is not yet clear whether these distinct macrophage subtypes directly promote local remodeling (for example, via *SPP1*-mediated promotion of transforming growth factor β activity[35]) or result from differential polarization related to microenvironmental cues. Indeed, while a number of studies have described phenotypic and compositional changes of macrophages in PF[16,17,23,24], in this study we provide spatial contextualization and characterization of macrophage diversity in PF lungs at single cell resolution.

### A timeline of alveolar dysregulation

Finally, we sought to leverage the spatial heterogeneity of disease features across samples to recreate a 'molecular natural history' of PF progression. We hypothesized that, across these samples, leveraging the ability to specifically analyze each airspace as an independent unit would make it possible to capture much of the molecular evolution of alveolar remodeling and begin to establish an 'order of events' on the path to end-stage PF.

To this end, we first developed a machine learning approach to identify and segment lumens across samples based on spatial patterns of transcript expression (Figs. 1 and 6a; Methods). We then assigned cells to each lumen and filtered the analysis space to include only lumens likely to be alveolar in origin (Supplementary Table 13). Next, we ordered the remaining 1,747 airspaces on a continuum of most normal in composition (that is, 'homeostatic') to most remodeled based on the proportion of transcripts corresponding to the healthy alveolar transcript niche T4 (ref. 36; Fig. 6b). Supporting the validity of this pseudotime strategy, alveoli from unaffected samples are enriched at the start of the trajectory, and percent pathology tends to increase across the trajectory. After ordering alveoli by disease severity in this manner, we identified gene expression, cell-type composition and niche proportions significantly associated with pseudotime using generalized additive models (GAMs; Fig. 6b, Supplementary Figs. 20 and 21 and Supplementary Tables 14–17; Methods). Supported by both the cell-type and niche analyses, we find that initial loss of alveolar homeostasis was marked by a loss of capillary endothelial cells and AT1 cells. We observe an initial increase in proliferating AT2 cells (consistent with the classical description of 'hyperplastic alveolar epithelial cells' (AECs)), which then becomes less frequent as the remaining epithelium has increasing 'transitional' and terminal airway-type features/niches. Emergence of activated (*CTHRC1⁺/FAP⁺*) fibrotic fibroblasts appears around the transition to a more airway-like epithelium, while progressive accumulation of *FABP4⁺* and then *SPP1⁺* macrophages are later events.

We then used spectral clustering to group expression changes into four broad categories as follows: homeostasis and early, intermediate and late remodeling (Fig. 6b and Supplementary Table 18). Given limited knowledge of early disease mechanisms, we focused on genes in the homeostatic and early remodeling stages. We used a pseudobulk approach, aggregating gene expression levels across each airspace. To

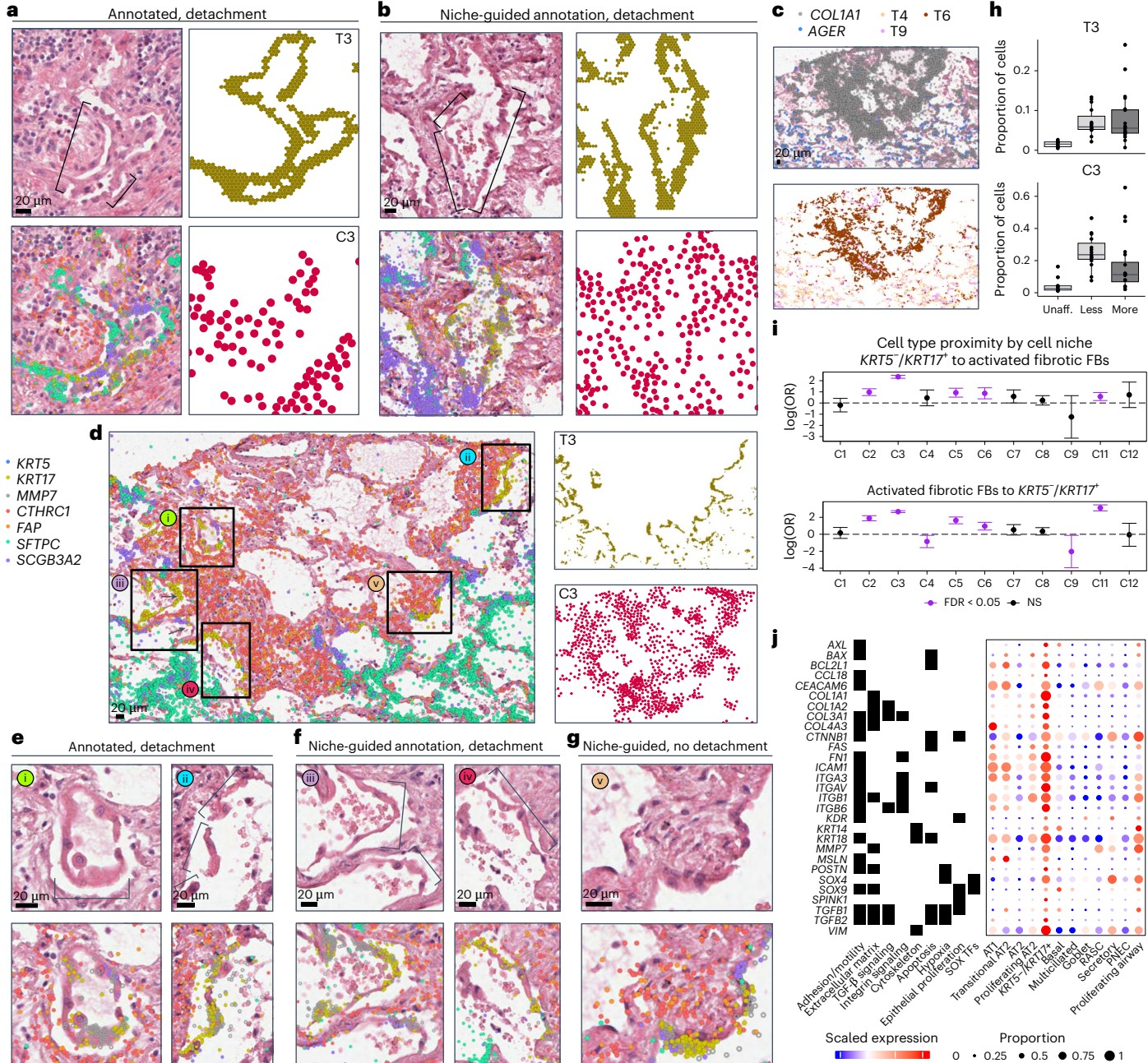

**Fig. 4 | *KRT5⁻/KRT17⁺* cells detach at sites of active fibrosis identified by spatial niches. a,b,** H&Es of epithelial detachment (brackets) directly annotated by a clinician (**a**) and not annotated (**b**), overlaying transcript expression of listed genes (colors shown left of **d**) and compared with the T3 (green) and C3 (red) niches. **c,** In one sample, we observed a dense fibrotic region marked by fibrotic niches (T6/T9; brown/pink) and *COL1A1* expression (gray) lined with epithelial detachment marked by T3/C3 adjacent to normal alveolar niches (T4/C8; light beige/latter not shown) expressing AT1 marker *AGER* (light blue). **d,** The same area is depicted, with expression is shown for the same genes as **a,b,e−g**, except for *MMP7*, which is omitted for clarity. **e−g,** This region contained two sites of epithelial detachment originally annotated by the clinician (**e**), additional examples of detaching *KRT5⁻/KRT17⁺* and transitional epithelial cells (**f**) and an instance of nondetaching *KRT5⁻/KRT17⁺* cells flanked by activated fibrotic fibroblasts (**g**). Scale bars = 20 μm. Samples−(**a**) VUILD107MA, (**b**) VUILD91MA and (**c−g**) VUILD91LA, all IPF-diagnosed. **h,** Proportion of cells assigned to

the T3 and C3 niches for each sample, split by disease state−unaffected, less affected and more affected (*n* = 10, *n* = 15 and *n* = 20, respectively). Boxplots show the median, hinges extend to first/third quartiles and whiskers extend to the largest/smallest (upper/lower) value to a maximum of 1.5× IQR. Outliers are shown as individual points. **i,** Proximity of *KRT5⁻/KRT17⁺* cells and activated fibrotic fibroblasts for each cell niche (*n* = 6−29,535 per cell type per niche). log OR and error bars (5−95% confidence interval) indicate the likelihood that a *KRT5⁻/KRT17⁺* cell's single nearest neighbor was an activated fibroblast (top) or vice versa (bottom). Significant (false discovery rate (FDR) < 0.05, purple) results above or below 0 (dashes) indicate increased or decreased likelihood for the cell types to be in close proximity within a niche. **j,** Heatmap of selected differentially expressed genes for *KRT5⁻/KRT17⁺* cells versus all other cell types. Genes belonging to a specific GO term or PANTHER category are marked (full groupings are provided in Supplementary Table 12). OR, odds ratio.

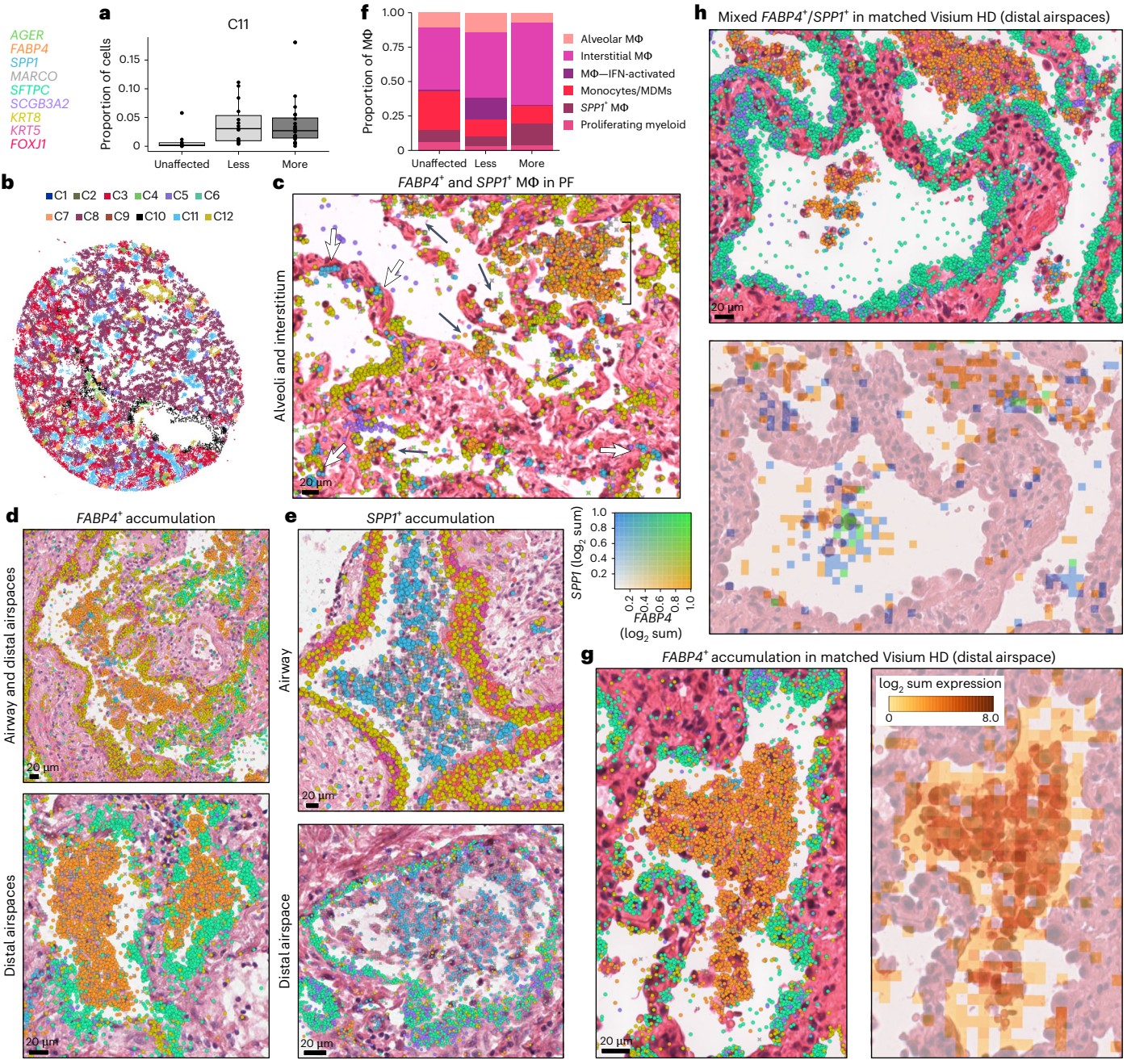

**Fig. 5 | *FABP4*⁺ and *SPP1*⁺ macrophages accumulate in PF airspaces and are characterized by a spatial niche. a,** Boxplot showing the proportion of cells assigned to the C11 niche across disease severity, including unaffected (*n* = 10), less affected (*n* = 15) and more affected (*n* = 20) samples. **b,** Representative example of cell niches, including the C11 macrophage accumulation niche (light blue) in sample VUILD102LA (IPF). **c,** *FABP4*⁺ and *SPP1*⁺ macrophages in minimally remodeled alveoli, including *FABP4*⁺ macrophages accumulated within an alveolus (denoted by a square bracket). Instances of individual *FABP4*⁺ macrophages that have migrated into alveolar lumens and small *FABP4*⁺ accumulations (black arrows) and *SPP1*⁺ macrophages in the interstitium (white arrows) are marked noncomprehensively. **d,e,** H&E images of *FABP4*⁺ (**d**) and *SPP1*⁺ (**e**) macrophage accumulations within airways and substantially remodeled distal airspaces overlain with transcript expression for listed genes. **f,** Distribution of macrophage subtypes as a proportion of the total population of macrophages across disease severity. **g,h,** Matched Xenium and Visium HD

images of *FABP4*⁺ (**g**) and mixed *FABP4*⁺/*SPP1*⁺ (**h**) macrophage accumulations within distal airspaces. For **g**, the Visium HD image (right) shows the sum of the log₂ expression of a list of genes marking alveolar macrophages as a density map overlain on the H&E. In **h**, the sum log₂ expression of a list of genes marking alveolar macrophages is compared to a list marking *SPP1*⁺ macrophages. Genes that were strong markers for both alveolar and *SPP1*⁺ macrophages were not included in **h**. See Methods for the gene selection process and Supplementary Table 10 for a list of marker genes. For **c**–**e**,**g**,**h** Xenium images, all listed genes are potentially visible in each example image if expressed, except *SCGB3A2*, which for clarity is not shown on the two figures that include airways, and *AGER*, which is only shown in **d**. Scale bars on the bottom left of each H&E = 20 μm. **c,** Sample TILD130LA (IPF); **d,** examples from VUILD91MA (top; IPF) and VUILD96LA (bottom; sarcoidosis); **e,** samples VUILD78MA (top; IPAF) and VUILD96MA (bottom; sarcoidosis) and **g,h,** sample VUILD49LA (cHP).

deconvolve these dynamics in the early remodeling genes, we quantified the expression level in each cell type from all cells within the 1,747 airspaces (Fig. 6c and Supplementary Table 19). Furthermore, we

carried out a cell-type-level GAM analysis for 25 cell types with sufficient counts, accounting for cell-type composition. In both analyses, we see a clear localization of signal to epithelial cells, with 30.9% and 32.3% of the

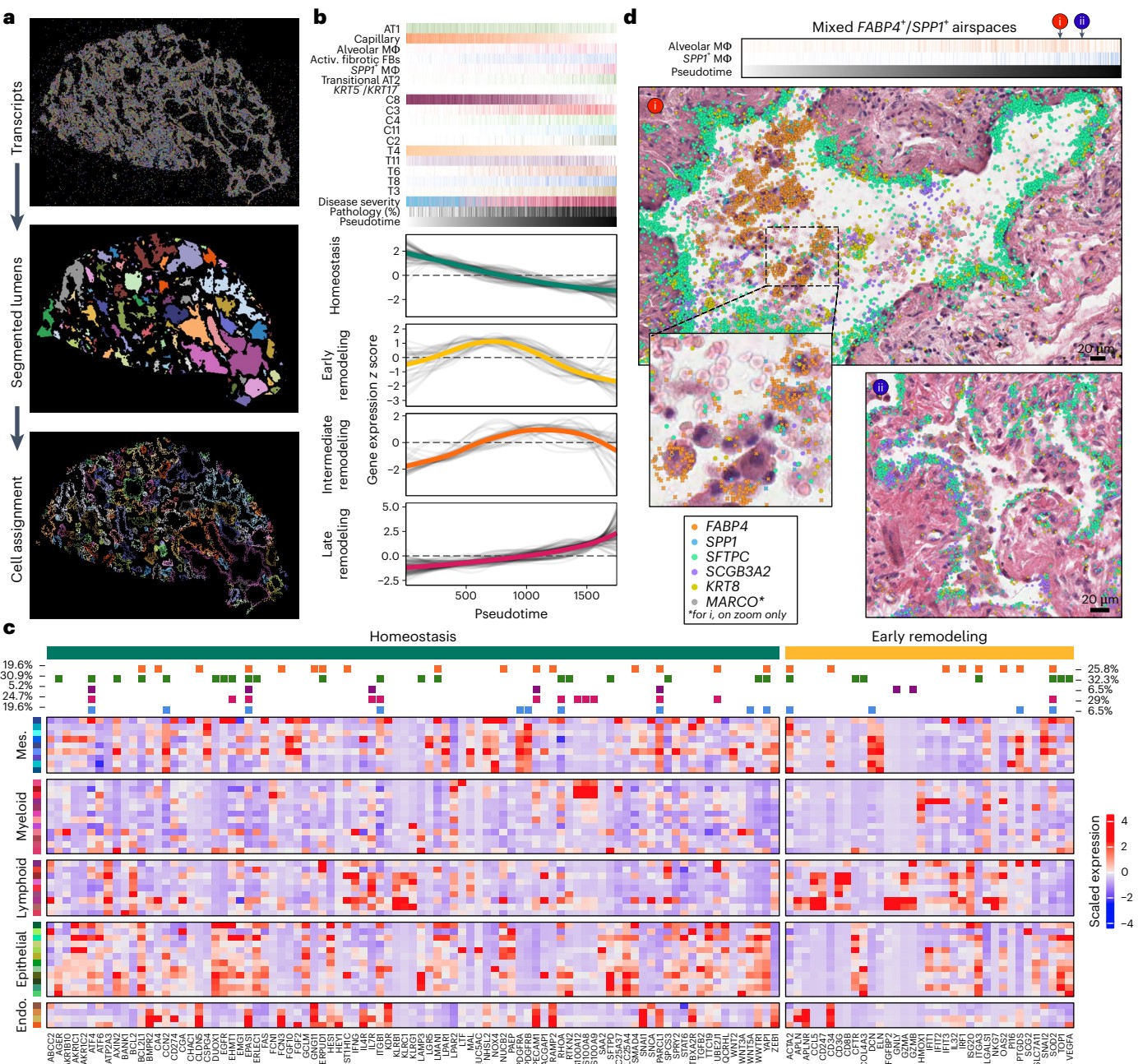

**Fig. 6 | Alveolar remodeling at airspace resolution. a**, Representation of lumen segmentation pipeline. **b**, Heatmap of predicted expression of each gene that was significantly associated with pseudotime. The top annotation shows select cell types, cell niches and transcript niches that were associated with pseudotime, with the darkest shade of each color representing the maximum proportion of that cell type or niche found across all airspaces. Disease severity is split into unaffected (blue), less affected (pink) and more affected (red). **c**, Scaled expression across cell types for the 124 genes with altered expression in the homeostatic (89 genes) or early remodeling stages (35 genes) from **b**, using only cells that were contained within one of the 1,747 airspaces. Cell-type colors are by lineage as in Fig. 2. On the top, boxes are filled in for each gene if it showed a significant change in expression in at least one cell type across the pseudotime of each of the following lineages: endothelial (orange), epithelial (green), lymphoid (purple), myeloid (pink) and mesenchymal (blue). Percentages were calculated as the number of significant tests (FDR < 0.05) in either the homeostasis (green, left) or early remodeling (yellow, right) stages that occurred across the pseudotime in all cell types of each lineage divided by the total number of significant tests for that stage of alveolar remodeling. On the *y* axes, *z* score of 0 has been marked on each plot with a dashed line. **d**, H&E images of mixed alveolar (*FABP4*+) and *SPP1*+ macrophage accumulations in two alveoli ranked near the end of the pseudotime trajectory, overlaid with transcript expression for all listed genes. Above the H&Es, each alveolus is marked by its position in pseudotime, and the proportion of alveolar and *SPP1*+ macrophages is shown for each airspace across pseudotime as in **b**. The example alveoli shown are VUILD115MA_90 (cHP diagnosis; top) and VUILD78MA_27 (IPAF, bottom). Scale bars on the bottom right of each H&E = 20 μm.

significant associations in the cell-type-level GAM analysis coming from this lineage for homeostasis and early remodeling, respectively, with 18.6% and 16.1% localized specifically to AT2 cells. This result implies that molecular pathology in the alveolar epithelium is evident before

extensive architectural remodeling occurs and supports the conceptualization of epithelial injury and dysregulation as central to PF risk and disease initiation. This conclusion contrasts significant associations identified in late remodeling, where 34.5% of total associations are

from epithelial cells, but 37.8% are from myeloid cells (Supplementary Fig. 22). Interestingly, when focusing on myeloid cells in late remodeling, we observed two ordered peaks of *FABP4*+ macrophages followed by *SPP1*+ macrophages. At the intersection of these two peaks, we identified several airspaces with both *FABP4*+ and *SPP1*+ macrophages (Fig. 6d), suggesting the possibility that *FABP4*+ macrophage accumulation leads to recruitment of, or differentiation to, *SPP1*+ macrophages. Indeed, prior research has suggested both possibilities[17,37]. Together, these results support a conceptual model where initial alveolar remodeling is driven by disruption of the alveolar-capillary interface with activation of regeneration-associated programs in the epithelium, followed by a wave of subepithelial fibroblast activation, then subsequent myeloid cell recruitment/proliferation.

## Discussion

Building upon prior lung molecular atlas projects[11–13,18,19,38], we generated an integrated, single-cell resolution, spatially contextualized characterization of the cellular diversity of the adult distal lung in health and chronic fibrotic lung disease. Beyond contextualizing individual cell types, we established the molecular basis of a diversity of classical histopathologic features of PF. Provocatively, we identified numerous regions where *KRT5*−/*KRT17*+ cells detached en masse from their basement membrane when overlying areas adjacent to activated fibroblasts. This finding has been rarely reported previously[33,34], although there remains debate as to whether this occurs in vivo or reflects an ex vivo artifact. While we cannot conclusively exclude the possibility, we suggest it is unlikely the same feature would be identified in multiple samples, in an analogous cellular and spatial niche context, by chance if this were a stochastic ex vivo artifact. In these regions, *KRT5*−/*KRT17*+ cells often exhibit an elongated, squamous-type morphology, raising the possibility that such detachment could be akin to sloughing of squamous-type epithelia in other tissues/conditions, in this case occurring in a pathologic state where repair and re-epithelialization may be ineffective. A similar finding was recently reported in an independent preprint, including evidence of sloughed airway epithelium in live-imaged surgical biopsies, suggesting this phenomenon can occur in vivo[39]. The implications of this process are not yet clear, but one possibility is that exposed basement membranes could be prone to patchy fusion or permit migration of fibroblasts into the airspace where they elaborate pathologic extracellular matrix that leads to eventual airway obstruction, yielding cystic structures distal to the point of fusion/obstruction. In light of recent adoptive transfer studies suggesting IPF basal/basal-like cells can potentiate fibrosis when instilled into the airway[40], these findings raise the possibility that paracrine effects of *KRT5*−/*KRT17*+ cells could extend beyond immediately adjacent neighbors in vivo.

We also found that whether using cell-aware or cell-agnostic approaches, there are a determinable number of conserved, molecularly definable spatial 'niches' in the human lung. As key cellular processes occur in a spatially and temporally coordinated manner, conceptualizing these niches as distinct functional units allows for directed interrogation of cellular and molecular programs in a specific context. We found that there were substantial shifts in the relative abundance of a given niche across disease pathology. Perhaps most strikingly, even within relatively preserved regions of fibrotic lungs, the molecular signature of 'normal alveoli' was virtually absent, suggesting that substantial molecular pathology precedes extensive tissue/architectural remodeling.

We then extended this concept further by developing a new approach to segment individual alveoli/airspaces and explore the evolution of molecular pathology in progressively more remodeled regions. Rather than an initial influx of inflammatory cells or fibroblast activation, these results suggest that disruption of the alveolar epithelium and adjacent capillary network are observed before another structural remodeling is detected. This concept is supported by additional evidence that suggests PF risk is mediated primarily through the lung epithelium[9,41]. Other recognized disease-associated features, including the emergence of abundant activated fibroblasts and accumulation of macrophages, appear to be later events in remodeling. These findings imply that precision therapeutic strategies will likely require concurrent assessment of which cellular mechanisms are most prominent in an individual at a given time. This not only presents potential challenges but also raises the possibility of improving outcomes (and minimizing toxicities) by better-aligning therapeutics with individual patient disease biology.

There are several limitations to this study. First, while this is the largest imaging-based spatial transcriptomic study of the human lung reported to date, this study ultimately reflects a relatively small number of individuals (*n* = 35), samples collected from organ donors or end-stage disease and participants who were predominantly of European ancestry. Imaging-based spatial transcriptomic platforms are also inherently semi-targeted; the probe set used for this study was informed by prior scRNA-sequencing datasets and developed specifically for cell identification and examination of established PF-related molecular programs and pathways. Additionally, cell segmentation remains a challenge, particularly in organs (including the lung) where many cell types have irregular shapes and/or sizes. While emerging cell-boundary staining procedures can improve this somewhat, we anticipate this will remain a challenge given the 3D structural relationships in the distal lung. We attempted to mitigate these issues for cell-aware analyses by restricting our dataset to transcripts overlying nuclei, but some degree of transcript 'contamination' from adjacent/overlying cells remained, requiring post hoc filtering for gene-level analyses. Unlike scRNA-seq, this contamination is nonrandom; thus, 'denoising' will require new computational approaches.

In brief, this study provides a comprehensive characterization of the cellular diversity and molecular pathology of the adult distal lung in both health and PF. The identification of conserved, molecularly definable spatial niches and their evolution across disease provides insights into PF pathogenesis, and the development of new analytical approaches for quantification and interrogation of multicellular niches using spatial transcriptomic approaches serves as a valuable resource for the lung biology community.

## Online content

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

[1]Division of Bioinnovation and Genome Sciences, Translational Genomics Research Institute, Phoenix, AZ, USA. [2]St. Vincent's Institute of Medical Research, Fitzroy, Victoria, Australia. [3]Melbourne Integrative Genomics, University of Melbourne, Parkville, Victoria, Australia. [4]School of Mathematics and Statistics, Faculty of Science, University of Melbourne, Parkville, Victoria, Australia. [5]Division of Neonatology, Department of Pediatrics, Vanderbilt University Medical Center, Nashville, TN, USA. [6]Division of Allergy, Pulmonary and Critical Care Medicine, Department of Medicine, Vanderbilt University Medical Center, Nashville, TN, USA. [7]Department of Pathology, Microbiology and Immunology, Vanderbilt University Medical Center, Nashville, TN, USA. [8]Department of Cardiac Surgery, Vanderbilt University Medical Center, Nashville, TN, USA. [9]Department of Cell and Developmental Biology, Vanderbilt University, Nashville, TN, USA. [10]Department of Veterans Affairs Medical Center, Nashville, TN, USA. [11]Department of Internal Medicine, University of Michigan School of Medicine, Ann Arbor, MI, USA. [12]Department of Thoracic Disease and Transplantation, Norton Thoracic Institute, Phoenix, AZ, USA. [13]These authors contributed equally: Annika Vannan, Ruqian Lyu. [14]These authors jointly supervised this work: Jonathan A. Kropski, Davis J. McCarthy, Nicholas E. Banovich. ✉e-mail: nbanovich@tgen.org

## Methods

### Participants and samples

The studies described here comply with ethical regulations as approved under local institutional review boards, including providing written informed consent (Vanderbilt University Medical Center (VUMC) Institutional Review Board (060165 and 171657) and Western Institutional Review Board (20181836)). Peripheral PF lung samples were obtained from lungs removed at the time of lung transplant surgery at VUMC or Norton Thoracic Institute as previously described[9,18,42]. Control lung tissue samples were obtained from lungs declined for organ donation. Diagnoses were determined by the local treating clinicians and affiliated multidisciplinary committee and confirmed by review of explant pathology according to the current American Thoracic Society/European Respiratory Society consensus guidelines[5,43]. Sample names were generated based on their collection site (Vanderbilt University—'VU'; Translational Genomics Research Institute—'T'), disease status (healthy donor—'HD'; interstitial lung disease—'ILD'), unique number assigned to the patient and where applicable, whether the sample was taken from a 'less affected' or 'more affected' tissue section as determined by percent pathology scores described later in the Methods. Replicate healthy samples were labeled 'A' and 'B' to distinguish (for example, VUHD116A and VUHD116B), and replicate disease samples with the same 'less affected' or 'more affected' designation were labeled '1' and '2' (for example, VUILD105MA1 and VUILD105MA2).

### Tissue microarray (TMA) construction

To maximize efficiency per run, multiple samples could be placed on a single Xenium slide (10.45 mm × 22.45 mm) using a TMA design. A 5-µm section of each lung formalin-fixed paraffin-embedded (FFPE) block was hematoxylin and eosin (H&E) stained and presented to a physician who identified areas of interest and labeled them as less or more fibrotic (LF or MF) relative to the sample. For TMAs 1–4, blocks were designed in a 3 × 3 or 2 × 2 pattern for 3 mm and 5 mm cores, respectively (Supplementary Table 1). Sample cores were punched and placed manually using a 'Quick-Ray Manual Tissue Microarrayer Full Set' and 'Quick-Ray Molds', according to the manufacturer's directions. Empty core spaces were filled with core punches taken from blank paraffin blocks (VWR, 76548-194). Once complete, blocks are placed face down on a clean glass slide and briefly heated in a warm drawer (~45 °C) to slightly melt the paraffin together and even the block face. TMA blocks were then cooled to room temperature, removed from the slide, sealed and stored at 4 °C.

An additional TMA was constructed (TMA5) after improvement to the 10X Xenium processing and software. TMA5 was designed in a 3 × 6 pattern of 3 mm cores with the top left core space remaining empty for orientation, with 17 samples total. A special 3 mm square tip was created by the Arizona State University Instrument Design and Fabrication Core to fit the Quick-Ray Manual Tissue Microarrayer and allow us to produce square tissue cores. Due to the new core shape, this TMA was constructed by placing each core individually on double-sided adhesive tape (Gorilla Double-Sided Mounting Tape) in a paraffin block mold[44]. The entire mold is warmed to 37 °C, and a P1000 is used to dispense 400 µl of melted paraffin wax in the lanes between the cores. As more wax was poured into the back of the mold, the mold was gently tapped to dislodge air bubbles. Once solidified, the block was extracted from the mold, and the tape was removed. TMA5 then underwent the same warm drawer and storage conditions as all others.

During TMA preparation, some samples were placed closely together such that a subset of transcripts and nuclei could not be distinguished as belonging specifically to either sample. These transcripts and nuclei were removed from downstream analyses.

All samples were processed through the Xenium workflow, and four specific samples from TMA5 (one unaffected and three PF) were then additionally run through the Visium HD protocol after Xenium processing to generate orthogonal data for verifying phenomena of interest. Both workflows are described below.

### Xenium in situ workflow

**Gene panel design.** Xenium in situ technology requires the use of a predefined gene panel. Each probe contains two paired sequences complementary to the targeted mRNA as well as a gene-specific barcode. Upon binding of the paired ends, ligation occurs. The now circular probe is amplified via rolling circle amplification, increasing the signal-to-noise ratio for target detection and decoding. A total of 343 unique genes were included in the analysis of this dataset. A total of 246 genes came from an early version of the Xenium human lung base panel (PD_277) and 97 genes from a custom-designed panel (CVEVZD). The custom panel was curated based on human lung single-cell analysis data[18], selecting genes useful for cell-type identification and/or suspected to be involved in IPF.

**Xenium sample preparation.** As Xenium in situ technology examines RNA, all protocol workstations and equipment were cleaned using RNase AWAY (RPI, 147002) followed by 70% isopropanol. All reagents, including water, were molecular-grade nuclease-free. Sample preparation began with rehydrating and sectioning FFPE blocks on a microtome (Leica, RM2135). Sections measuring 5 µm were placed onto Xenium slides (10X Genomics). Following overnight drying, slides with placed samples were stored in a sealed desiccator at room temperature for ≤10 days. Slides were then placed in imaging cassettes for the remainder of the preparation. Tissue deparaffinization and decross-linking steps made subcellular RNA targets accessible. Gene panel probe hybridization occurred overnight for 18 h at 50 °C (Bio-Rad DNA Engine Tetrad 2). Subsequent washes the next day removed unbound probes. Ligase was added to circularize the paired ends of bound probes (2 h at 37 °C) and followed by enzymatic rolling circle amplification (2 h at 30 °C). TMA5 was prepped to include Xenium Multi-Tissue Stain (10X Genomics, 1000662) according to 10X Genomics' Demonstrated Protocol CG000749. All slides were washed in Tris-EDTA (TE) buffer before background fluorescence was chemically quenched; autofluorescence is a known issue in lung tissue as well as a byproduct of formalin fixation[45]. Following PBS and PBS-T washes, DAPI was used to stain sample nuclei. Finalized slides were stored in PBS-T in the dark at 4 °C for ≤5 days until being loaded onto the Xenium Analyzer instrument. Stepwise Xenium FFPE preparation guidelines and buffer recipes can be found in 10X Genomics' Demonstrated Protocols CG000578 and CG000580.

**Xenium Analyzer instrument.** The Xenium Analyzer is a fully automated instrument for decoding subcellular localization of RNA targets. The user marks regions for analysis by manually selecting the sample location on an initial low-resolution, full-slide image. After loading consumable reagents and a maximum of two slides per run, internal sample and liquid handling mechanics control experiment progression. Data collection occurs in cycles of fluorescently labeled probe binding, image acquisition and probe stripping. Images of the fluorescent probes are taken in 4,240 × 2,960-pixel fields of view (FOVs). Localized points of fluorescence intensity detected during the rounds of imaging are then defined as potential RNA puncta. Each gene on the panel has a unique fluorescence pattern across the image channels. Puncta that match a specific pattern is then decoded and labeled according to the gene ID. Finally, all image FOVs and associated detected transcripts are computationally stitched together via the DAPI-stained image. Onboard analysis pipelines present quality values for each detected transcript, based on variable confidence in the signal and decoding process. Data for TMA1–TMA4 were acquired on instrument software version 1.1.2.4 and analysis version xenium-1.1.0.2. Data for TMA5 were acquired on instrument software version 2.0.1.0 and analysis version xenium-2.0.0.10. TMA5 had additional fluorescent images taken of

each Xenium Multi-Tissue Stain channel to allow for cell segmentation. Detailed instructions on instrument operation and consumable preparation can be found in 10X Genomics' Demonstrated Protocol CG000582.

**Postrun histology.** After the run, slides were removed from the Xenium Analyzer instrument and had quencher removed according to 10X Genomics' Demonstrated Protocol CG000613. Immediately following, the slides were H&E stained according to the following protocol: xylene (x3, 3 min ea), 100% alcohol (x2, 2 min ea), 95% alcohol (x2, 2 min ea), 70% alcohol (2 min), deionized (DI) water rinse (1 min), hematoxylin (1 min; Biocare Medical CATHE), DI water rinse (1 min), bluing solution (1 min; Biocare Medical HTBLU-M), DI water rinse (3 min), 95% alcohol (30 s), eosin (5 s; Biocare Medical HTE-GL), 95% alcohol (10 s), 100% alcohol (x2, 10 s ea) and xylene (x2, 10 s ea). Coverslipping was performed using Micromount (Leica, 3801731) and cured overnight at room temperature. Histology images were taken on a ×20 Leica Biosystems Aperio CS2.

## Data preprocessing

**Cell segmentation.** Cell segmentation was performed with 10X Xenium onboard cell segmentation (10X Genomics). DAPI-stained nuclei from the DAPI morphology image were segmented, and boundaries were consolidated to form nonoverlapping objects. For samples on TMAs 1–4, nuclear boundaries were expanded by 15 µm or until they reached another cell boundary to approximate cell segmentation. For samples on TMA5, cells were segmented using a cell boundary stain. For all TMAs, nuclear segmentation was used to define cells.

**Image registration.** Registration was performed between Xenium DAPI morphology images (that is, nuclei) and H&E-stained images of the same slice with the BigWarp plugin implemented in ImageJ (2.14.0)[46,47]. To place both images in the coordinate space of the DAPI morphology image, the H&E stained image was specified as the moving image, and the DAPI morphology was specified as the target. Anchors were placed on identifiable landmarks on both images (approximately 200 landmarks per image pair). A thin plate spline warp was then applied to align the corresponding anchor points between the two images. Registered images were manually reviewed for potential visual artifacts resulting from the registration process. These registered images were then visualized in Xenium Explorer 3.0.0 software, which was used to generate figures showing transcript expression overlain on H&E stains.

**Quality filtering and data preprocessing.** For each sample, Xenium generated an output file of transcript information, including $x$ and $y$ coordinates, corresponding gene, assigned cell and/or nucleus and quality score. Low-quality transcripts (quality value (QV) < 20)) and transcripts corresponding to blank probes were removed. Seurat v5 was used to perform further quality filtering and visualization on nuclei gene expression data[32]. Nucleus-by-gene count matrices were created for each sample based on the expression of transcripts that fell within segmented nuclei. A single merged Seurat object was created for all samples based on these count matrices and metadata files with nuclei coordinates and area. Nuclei were retained according to the following criteria: ≥12 transcripts corresponding to ≥10 unique genes, percentage of high-quality transcripts corresponding to negative control probes, negative control codewords, unassigned codewords, or the cumulative percentage of these ≤5, and nucleus area ≥6 and ≤80 µm. Because Xenium outputs coordinates based on each slide, which results in samples with shared coordinates across multiple slides, the nuclei coordinates were manually adjusted for visualization so that no samples overlapped during plotting. These adjusted nucleus coordinates were added to the Seurat object as a dimension-reduction object.

**Dimensionality reduction, clustering and cell-type annotation.** Scanpy[48] in Python was used to perform dimensionality reduction and clustering, while Seurat v5 (ref. 32) in R was used for visualization of these results[49]. Gene expression was normalized per cell using a log1p transformation. Dimensionality reduction was performed with principal component analysis. Nuclei were clustered using the Leiden algorithm[50] based on this dimensionality reduction and computing a nearest neighbors distance matrix. Uniform Manifold Approximation and Projection (UMAP) plots of the data were then generated for visualization. For speed, these calculations were performed using a container[51] with the RAPIDS (v21.8.1) implementation[52] of Scanpy (v1.8.1).

For cell-type annotation, we used both marker genes and spatial information. We did not rely solely on gene expression because some level of gene 'contamination' is expected with spatial data, as transcripts located in one cell may be assigned to an adjacent cell in sufficiently close proximity. Therefore, we incorporated spatial data including cell morphology and histological features to label clusters. Initial Leiden clusters were primarily segregated by cell lineage based on marker genes assessed using the Seurat v5's FindMarkers function, including *PECAM1* (endothelial), *EPCAM* (epithelial), *PTPRC* (immune), *DCN, LUM* and *COL1A1* (all mesenchymal)[32]. New Seurat/AnnData objects were created for each of the four lineages, and dimensionality reduction and clustering were performed again as described above. Clusters were then given first-pass cell-type labels based on marker genes. The epithelial and immune lineages were further split into sublineages for alveolar and airway cells (epithelial) as well as myeloid and lymphoid cells (immune), and new Seurat/AnnData objects were generated and reprocessed for these sublineages. As lineage and subgroup splitting became more accurate, clusters were relabeled with revised annotations. 'Stray' clusters that did not belong to the lineage or sublineage they were originally assigned to, as well as clusters that were marked by genes from more than one lineage, received final annotations based on shared marker genes and spatial patterns with confidently labeled clusters. Low-quality clusters with extremely low transcript counts and/or with conflicting marker genes that could not be resolved (similar to 'doublets' in scRNA-seq data) were removed. This resulted in 47 final cell types, including 4 endothelial, 12 epithelial, 22 immune and 9 mesenchymal (Extended Data Figs. 2–5 and Supplementary Table 2).

## H&E image annotation

**Percent pathology assignment.** Percent pathology was assessed by visual estimation of the fraction of the total imaged area of the sample with architectural remodeling. Disease samples were labeled 'more affected' if their percent pathology was greater than or equal to 75%. All other disease samples were labeled 'less affected', and control samples were labeled 'unaffected' regardless of percent pathology score.

**Annotation of histological features.** We annotated representative examples of 27 histological features across all 45 samples, including 12 epithelial or of likely epithelial origin (normal alveoli, minimally remodeled alveoli, hyperplastic AECs, emphysema, remodeled epithelium, advanced remodeling, epithelial detachment, remnant alveoli, small airway, large airway, microscopic honeycombing and goblet cell metaplasia), 3 vascular (artery, muscularized artery and venule), 3 immune (granuloma, mixed inflammation and TLS), 5 mesenchymal/interstitial (interlobular septum, airway smooth muscle, fibroblastic focus, fibrosis and severe fibrosis) and 2 general (multinucleated cell and giant cell) feature types. Annotations were marked on the registered H&E images using QuPath (v.0.4.3)[53]. Cells were then assigned to annotated regions using a custom Python (v.3.12.3) script. Briefly, annotated regions were scaled by a factor of 0.2125 µm per pixel to harmonize units with the cell centroid coordinates. Cells were assigned a boolean value for each annotated region corresponding to whether the cell centroid fell within the annotated region. After filtering to annotations containing at least one cell, there were 712 annotations across the 27 histological features.

## Comparison to scRNA-seq datasets

We compared cell lineage recovery from the present spatial dataset to two scRNA-seq sources. The first was ref. 9, a recent sc-eQTL study from our lab[9], and the other was the Human Lung Cell Atlas (HLCA), which included aggregated data from multiple lung scRNA-seq studies[12]. Studies from the HLCA varied in the disease studied and whether the dataset contained control and/or lung disease samples. We narrowed the scope of our comparison to studies (lung atlases) or samples (sc-eQTL study) that did not specifically enrich or deplete cells from a specific lineage in their preprocessing and did not exclusively contain data from nasal samples. We included 47 samples from the sc-eQTL study (19 control and 28 ILD) and 14 datasets from the HLCA. For each data source, the proportion of cells from each lineage (endothelial, epithelial, immune and mesenchymal) was calculated first for all samples and then for control and disease samples separately. If an annotated cell type did not have a clear lineage association, it was removed from the analysis. Sample-level data from the present study was compared to sample-level data from the sc-eQTL study and dataset-level information from HLCA.

## Post-Xenium Visium HD workflow for select samples

**Post-Xenium Visium HD sample preparation.** All protocol workstations and equipment were cleaned using RNase Away (RPI 147002) followed by 70% isopropanol. All reagents, including water, were molecular-grade nuclease-free. The Xenium slide containing IPFTMA5 was H&E stained postrun as described in 'Xenium in situ workflow—Postrun histology'. The slide was stored for 9 days postcoverslipping in a sealed desiccator at 4 °C. The slide was then decoverslipped in xylene, loaded into a Visium tissue slide cassette (10X Genomics), and destained using 0.1 N HCl. The decross-linking step of the archived slide protocol was skipped due to this already being performed as part of the Xenium workflow. Stepwise archived slide preparation guidelines and buffer recipes can be found in 10X Genomics' Demonstrated Protocol CG000684.

**Visium HD CytAssist preparation.** The tissue slide within the Visium HD tissue cassette was then prepared for loading onto the CytAssist instrument. It underwent human probe hybridization (10X Genomics, 1000466) and a posthybridization wash to remove unbound probes, followed by probe ligation and subsequent wash. The Visium HD slide containing the probe capture areas was prepared. Both the Visium HD slide and the tissue slide were loaded onto the CytAssist instrument to enable the probe release and capture. Four samples were targeted as a representative subset (one unaffected and three fibrotic) for the Visium HD analysis (VUHD049, VUILD49LA, TILD111LA and TILD113LA) as it has a smaller capture area than the Xenium slide. Once captured on the Visium HD slide, probes were extended, eluted into 0.08 M KOH and then transferred into 1 M Tris–HCl (pH 8). Pre-amplification cleanup was performed using SPRIselect reagent (Beckman Coulter, B23318). The sample was stored at 4 °C overnight. Detailed instructions regarding Visium HD CytAssist operation, tissue cassette and Visium HD slide preparation can be found in 10X Genomics' Demonstrated Protocol CG000685.

**Visium HD library construction and sequencing.** The probe-based library was constructed using Dual Index Plate TS Set A (10X Genomics, 1000251) with a sample index PCR cycle number of 16, as determined by the Cq value from qPCR (Applied Biosystems QuantStudio 5). A final cleanup step with SPRIselect reagent was followed by storage at −20 °C until sequencing. Library quality control was performed on an Agilent TapeStation 4200. Next-generation sequencing was carried out on Illumina's NovaSeq X platform using a 10 billion read, 300 cycle flow cell. Minimum sequencing recommendations were met using the product of tissue coverage percentage estimations and a constant maximum read figure of 275 million read pairs. Optimal cluster density was achieved with a lane loading concentration of 235 pM. All steps were performed according to the instructions found in 10X Genomics' Demonstrated Protocol CG000685.

**Visium HD data processing.** A high-resolution image was aligned using the Visium HD Manual Alignment tool on Loupe Browser (v8.0.0; 10X Genomics)[54]. Five matched landmarks per sample within the capture area were selected for the alignment and then algorithmically refined by the software. Visium HD sequence data, high-resolution image and alignment file were processed using the count function of Spaceranger (3.0.0; 10X Genomics)[55] with default parameters. Reads were mapped against the GRCh38-2020-A reference.

**Visium HD data analysis.** To compare the Xenium and Visium HD outputs, the Loupe Browser (v8.0.0) software was used to visualize gene expression over H&E images for the Visium HD data. Composite gene expression scores were used to visualize marker gene expression of several cell types of interest. These composite scores were created by summing the $\log_2$-transformed unique molecular identifier (UMI) counts of 7–26 genes per cell type that were selected based on the dot-plot heatmap in Extended Data Fig. 1 and prior literature ($KRT5^-/KRT17^+$ 'aberrant basaloid' cells[18,19]; activated fibrotic fibroblasts[56]; alveolar $FABP4^+$ macrophages[24,37]; $SPP1^+$ macrophages[24,37]; Supplementary Table 10). Three strong canonical marker genes for $KRT5^-/KRT17^+$ cells or activated fibroblasts ($COL1A1$, $FN1$ and $ACTA2$) were not included in the composite scores for these cell types because they mark both cell types ($COL1A1$ and $FN1$) or because they strongly mark another cell type ($ACTA2$ marks smooth muscle). For the comparison between alveolar macrophages and $SPP1^+$ macrophages (Fig. 6h), genes that were strong markers for both macrophage populations in the present Xenium dataset according to Supplementary Figs. 3 and 6 were excluded from the alveolar macrophage marker list, except for $PPARG$, which is known to distinguish these populations in scRNA-seq and had low expression in $SPP1^+$ regions in the Visium HD data.

## Computational niche identification

**Transcript-based niches.** Transcript-based niches were characterized using a graph neural network model GraphSAGE (implementation in StellarGraph v1.2.1 + Python 3.8.0)[31,57] that directly modeled detected transcripts without cell segmentation. GraphSAGE learns the structures in the graph data via sampling and aggregating neighbors, scaling well to large graphs. Detected transcripts from each sample were used to build a sample graph after removing low-quality transcripts (QV < 20) and blank probes. Each sample graph consisted of nodes representing individual transcripts, with edges drawn between nodes that had Euclidean distances (based on spatial coordinates) smaller than a threshold ($d = 3.0$) similar to a previous study[57]. Each node was associated with an input feature vector that was the one-hot encoding of the node's gene label, and small components with nodes with fewer than ten nodes were removed ($n = 299,018,086$ after filtering).

A two-hop GraphSAGE model was applied to learn node embeddings. It sampled 20 and 10 neighboring nodes at the first-hop and second-hop neighbors, respectively, to learn 50-dimensional embeddings at each hop for each node in the graph. To embed all nodes across samples into the same embedding space, we trained the model on a joined graph consisting of subgraphs from 28 samples (TMAs 1–4) with each subgraph containing 5,000 randomly sampled root nodes along with their three-hop neighbors as well as the existing edges among them. The joined graph contained 18,594,542 nodes and 421,316,086 edges for model training. The model was trained in an unsupervised manner by solving a binary classification task for 'positive' and 'negative' node pairs, in which the model predicts whether or not two nodes should have an edge connection based on their local neighborhood structures. We trained the model with ten epochs, achieving a training accuracy of 0.82 (ref. 31).

Embedding representations of all nodes (transcripts) across the 28 samples from TMAs 1–4 were obtained using the trained model, and we performed unsupervised clustering on all nodes using the Gaussian mixture model ($k = 12$) as implemented in the PyCave library (https://github.com/borchero/pycave). We subsequently obtained the embeddings of the 17 samples from TMA5 using the same trained model and projected their transcripts into the existing 12 transcript clusters.

*Transcript-based niche plots and assignment of nuclei to niches.* We created a hexbin summarization plot for each sample using all transcripts with a bin width of 5 for generating the transcript-based niche plots to overcome overplotting issues. Each bin was labeled and colored with the major cluster label among the transcripts falling in the bin area. Bins with fewer than ten transcripts were not included in the plots. To assign cells to transcript-based niches, we assigned each cell to its closest hexbin by calculating the Euclidean distance of each cell centroid to the hexbin centroids.

### Cell-based niches
Seurat v5's[32] BuildNicheAssay function was used to partition cells into 12 spatial niches using $k$-means clustering based on the cell-type composition of their 25 closest neighboring cells.

### Cell-type proximity analysis
To assess proximity between cell types, each cell was anchored, and its distance and angular direction were calculated between the anchored cells and their neighboring cells within a fixed 60-µm radius circle. Only first-degree neighbors were considered proximal to the anchored cell. The probability of cell types being proximal to one another was assessed using logistic regression by assigning cells to binarized proximal and nonproximal categories and using cell type as a covariate. This analysis was performed both for all samples and cell types and within each cell and transcript niche individually.

### Differential expression and composition analyses
**Differential cell-type composition by disease severity.** We transformed the cell-type proportion per sample using logit transformation and tested whether the cell-type proportion was different among three disease groups (unaffected, less affected and more affected samples) using the propeller.anova function implemented in propeller[58].

**Representative gene detection in transcript-based niches.** We used the linear model framework implemented in propeller[58] and limma[59,60] to test differences in gene proportions across niches and thereby identify representative genes of each niche. We first investigated the proportion of transcripts assigned to each niche within each sample. Samples were excluded from testing for a niche when a few transcripts from that sample (proportion < $5 \times 10^{-4}$) were assigned to the niche. Gene proportions were logit-transformed, and a linear model was fitted to each gene that modeled the mean transformed gene proportion in each niche while accounting for sample differences. Differentially abundant genes were then derived by contrasting the mean gene proportion from one niche with the average mean proportion from the other niches.

**Differential expression, cell-type composition and niche proportions by percent pathology.** We used the same propeller[58] and limma[59,60] frameworks for finding gene, cell type and niche proportions that correlated with changes in percent pathology across samples. Proportions were logit-transformed, and linear models were fit to model the relationship between the transformed proportions and percent pathology across samples to find features (that is, genes, cell types and niches) that varied significantly with percent pathology (FDR < 0.01).

We also performed a cell-type-aware differential gene expression along percent pathology across samples by pseudo-bulking gene expression per identified cell type. For each cell type, we filtered the list of testing genes to remove contamination signals. To identify contamination genes per cell type, we first scaled the gene counts using a count-per-1K normalization across cells and kept genes with at least five counts in 30% of cells within a cell type for testing. The remaining genes per cell type were then tested with the propeller[58] and limma[59,60] frameworks.

**Differential expression analysis across annotated pathology features.** We aggregated gene counts from cells included in each pathology annotation instance and detected differentially expressed genes among pathology annotations by fitting linear models implemented in limma[59,61]. Gene expression was converted to counts per million (CPM) and $\log_2$-transformed after adding a 0.5 pseudocount. Lowly expressed genes ($\log_2$(CPM) < 8) in 50% of the annotation instances were excluded. The limma voom function was applied to model the mean–variance relationship and assign weights to each $\log_2$CPM observation value, which was subsequently used in the linear regression model to account for heteroskedasticity in count data. We controlled for the TMA effects by adding TMA origins as a covariate in the linear model. Differentially expressed genes for each annotation type were detected by comparing the gene expression per annotation type with the rest. Separately, we also compared epithelial annotations to each other to identify dysregulated genes in pathologic epithelium.

### Lumen segmentation and airspace identification
**Initial lumen segmentation.** To segment individual lumens from the spatial sequencing data, a custom graphics processing unit (GPU)-accelerated image processing algorithm was developed using scikit-image (v.0.19.3), RAPIDS cuCIM (v.22.12.00) and CuPy (v.11.2.0) in Python (v.3.9.7) to assign unique identifiers to lumens in the spatial transcriptome data[62–64]. Briefly, the location of each detected transcript, excluding transcripts associated with immune cells, was binarized to a 2D image followed by a series of dilation, erosion and closing operations to define the location of the tissue and segment the lumens. The outer boundaries of the tissue were defined using Alpha Shapes (alphashape package v.1.3.1) on a denoised, closed shape representing the entire tissue. Unique identifiers were then assigned to the negative space (lumens) using scikit-image, and metrics were calculated for each individual lumen. Cell centers (nuclei) within a cutoff distance, defined as the thickness of a normal alveolar wall, were then assigned to the unique identifier of the closest lumen by searching for the nearest label in a restricted zone using the K-D Tree nearest-neighbor lookup algorithm implemented in scikit-image. All cells that were not close to a lumen were given an identifier of zero.

**Quality filtering to isolate alveolar airspaces.** Lumens were first filtered by size to remove false positive segmentations. We retained lumens that contained between 25 and 500 cells, including at least 5% epithelial cells, for which the maximum distance between any two nuclei in the lumen was at least 110 µm. The high cell number and low epithelial cell thresholds were selected to retain lumens with large macrophage accumulations. For each lumen, the proportion of cells belonging to each cell niche was calculated, and the cell niche(s) with the highest proportion of cells was recorded as the 'maximum cell niche'. To discard lumens corresponding to endothelial and airway structures, lumens were required to have an endothelial niche proportion of C10 < 0.3 and C12 < 0.3 along with an airway niche proportion of C1 < 0.2. To further isolate lumens likely to be alveolar in origin, lumens were retained only if their maximum cell niche(s) were C2/C5 (transitional epithelium), C8 (healthy alveolar), C3 (*KRT5⁻/KRT17⁺* fibrotic niche) and/or C11 (airspace macrophage accumulation niche).

**Assessing changes in cell types, niches and gene expression along pseudotime.** Pseudotime analysis was performed on the remaining 1,747 alveolar airspaces based on the proportion of transcripts overlapping nuclei that were assigned to the T4 healthy alveolar niche. This allowed us to rank airspaces along a continuum of disease severity and tissue remodeling. To find cell types, niches and genes with varied abundances or expression along the pseudotime, we applied GAMs using the fitGAM function from tradeSeq[65] on the aggregated feature counts across airspaces under a negative binomial distribution. We subsequently applied associationTest[65] to test the association of features with pseudotime.

**Cell-type changes along pseudotime-ordered airspaces.** We counted the number of each cell type across airspaces and fitted a GAM model per cell type against pseudotime with an offset of log-transformed total cell counts. We excluded cell types if they were not present with at least three counts in at least ten airspaces (36 cell types tested). We obtained the list of significant cell types (28 in total) using the associationTest[65].

**Cell-type-based and transcript-based niches.** We applied similar analysis for cell-type-based and transcript-based niches as for cell-type analysis given above. Niches were excluded if they were not present with at least three counts in at least 10% of segmented airspaces. We obtained the smoothed niche abundance changes along 2,000 time points after model fitting and visualized their patterns in heatmaps after ordering them by their peak time point. The peak time point for each niche was determined using a rolling mean approach (window size, $n = 100$). Features with earlier peak times were plotted in the top rows.

**Classifying gene expression patterns along pseudotime.** We characterized gene expression patterns along pseudotime-ordered airspaces using GAM models by modeling the gene count along pseudotime under negative binomial distribution. We added the total number of detected transcripts per lumen as an offset term as well as a TMA covariate for controlling TMA effects. With the fitted model, we predicted the gene expression along pseudotime and clustered the gene expression patterns first using hierarchical clustering ($k = 20$) after taking $z$ scores per gene. Genes were separated into bimodal and unimodal patterns by visual inspection of the 20 hierarchical clusters. For the unimodal genes, we further clustered them by spectral clustering ($k = 4$) using the R package kernlab[66]. We ordered the four gene clusters by their gene expression peak times and labeled them into four categories, including homeostasis, early remodeling, intermediate remodeling and late remodeling. The gene expression dynamics of the four gene clusters were then visualized in heatmaps.

**Altered gene expression along pseudotime within each cell type.** We tested which genes had altered expression along pseudotime within a specific cell type using GAMs as well. We tested 25 cell types that were observed with more than three cells in at least 80 airspaces. For each cell type, we aggregated their gene expression across segmented airspaces. Within a specific cell type, for the expression of each gene $\mu$, we fit a negative binomial GAM model to model the gene expression across lumens using the gam function with cubic regression splines 'bs="cr"' implemented in mgcv[67] as

$$\log(\mu) = s(t) + \sum_{i=1}^{15} s(p_i) + \log(n) + U,$$

where $t$ represents pseudotime, $p_i$ represents the proportion of cell type $i$, $n$ represents the number of total transcripts per lumen and $U$ represents the TMAs. We applied the same strategies in knot selection as in tradeSeq[65] and selected five knots per smooth term. We included proportions of 15 major cell types that had more than three cells in more than 300 airspaces in the model to control for the contamination in detected gene expression signals. When testing genes for significant association with pseudotime per cell type, we restricted genes to the set of significant genes in the overall gene association test across all cell types. For each cell type, we then tested genes that were detected with more than three copies in 50% of the testing airspaces. Genes that had significant pseudotime terms after multiple testing corrections (FDR < 0.05) were summarized and plotted.

### Statistics and reproducibility

Sample sizes were chosen based on sample availability, although data generation was randomized with respect to disease and control samples. The study was unblinded. Any data exclusions are noted above, including filtering cells of a large size or unexpectedly low or high gene counts based on current best practices. Key findings from Xenium data were replicated with orthogonal technology (Visium HD) as described above. For figures depicting H&E images of described phenomena (for example, cell types in annotations in Fig. 3d and macrophage accumulations in Fig. 6c–e,g–h), at least three similar examples were observed across samples, and the images shown are intended to be representative but not exhaustive. For representative H&E images overlain with transcripts or cell types, see the accompanying dotplot heatmaps for additional context. Full data are available at the Gene Expression Omnibus (GEO), including histopathology, gene expression data and cell-type annotations.

### Reporting summary

Further information on research design is available in the Nature Portfolio Reporting Summary linked to this article.

### Data availability

Raw and processed data are deposited at GEO under accession GSE250346.

### Code availability

Custom R, Python and bash scripts for this project are available on GitHub at https://github.com/Banovich-Lab/Spatial_PF and on Zenodo[68].

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

## Acknowledgements

This work was supported by National Institutes of Health (NIH) R01HL145372 (to J.A.K. and N.E.B.), U01HL175444 (to J.A.K., N.E.B. and J.M.S.S.), R01HG011886 (to N.E.B. and D.J.M.), W81XWH1910415 (to N.E.B. and J.A.K.), T32HL094296 (to A.M.S.), R01160551 (to C.M.S.), P01HL092870 (to T.S.B.), the Francis Family Foundation (to J.J.G.), Vanderbilt Faculty Research Scholars (to J.J.G. and A.S.M.), K08143051 (to J.M.S.S.), R01HL168556 (to J.M.S.S.), the Department of Veterans Affairs (to T.S.B. and J.A.K.), T32HL094296 (to A.S.M.), National Health and Medical Research Council (NHMRC) (GNT1195595 and GNT1162829 to D.J.M.) and NIH (HL158906 and HL126176 to L.B.W.). We thank A. Dietrich for her thoughtful feedback on the manuscript.

## Author contributions

N.E.B., D.J.M., J.A.K., A.V. and R.L. conceptualized the study. R.L., N.M.N., J.H. and S.H. developed the methodology of this study. A.V., R.L., N.H. and E.D.M. carried out formal analysis. A.V., R.L., A.L.W., N.H., D.S.N., C.L.C., C.J.T., V.V.P., A.P.M.S., A.S.M., J.J.G., H.S., M.J.B., C.M.S., T.S.B. and J.M.S.S. conducted the investigation. N.E.B., D.J.M., J.A.K., L.B.W., R.W., T.S.B. and C.M.S. arranged the resources. R.L., A.V., E.D.M. and A.L.W. curated the data. A.V., R.L., N.E.B., D.J.M. and J.A.K. wrote the original draft and did the writing, reviewing and editing of the manuscript. A.V., R.L., E.D.M., N.H. and A.L.W. handled visualization. N.E.B., D.J.M. and J.A.K. provided supervision. N.E.B., D.J.M., J.A.K., L.B.W. and J.M.S.S. secured funding.

## Competing interests

J.A.K. reports grants/contracts from Boehringer Ingelheim and Bristol Myers Squibb, stock options from APIE Therapeutics and consulting fees from ARDA Therapeutics. R.W. reports consultant fees from Genentech and Boehringer Ingelheim. T.S.B. reports grants/contracts from Boehringer Ingelheim, Bristol Myers Squibb and Morphic. The other authors declare no competing interests.

## Additional information

**Extended data** is available for this paper at https://doi.org/10.1038/s41588-025-02080-x.

**Correspondence and requests for materials** should be addressed to Nicholas E. Banovich.

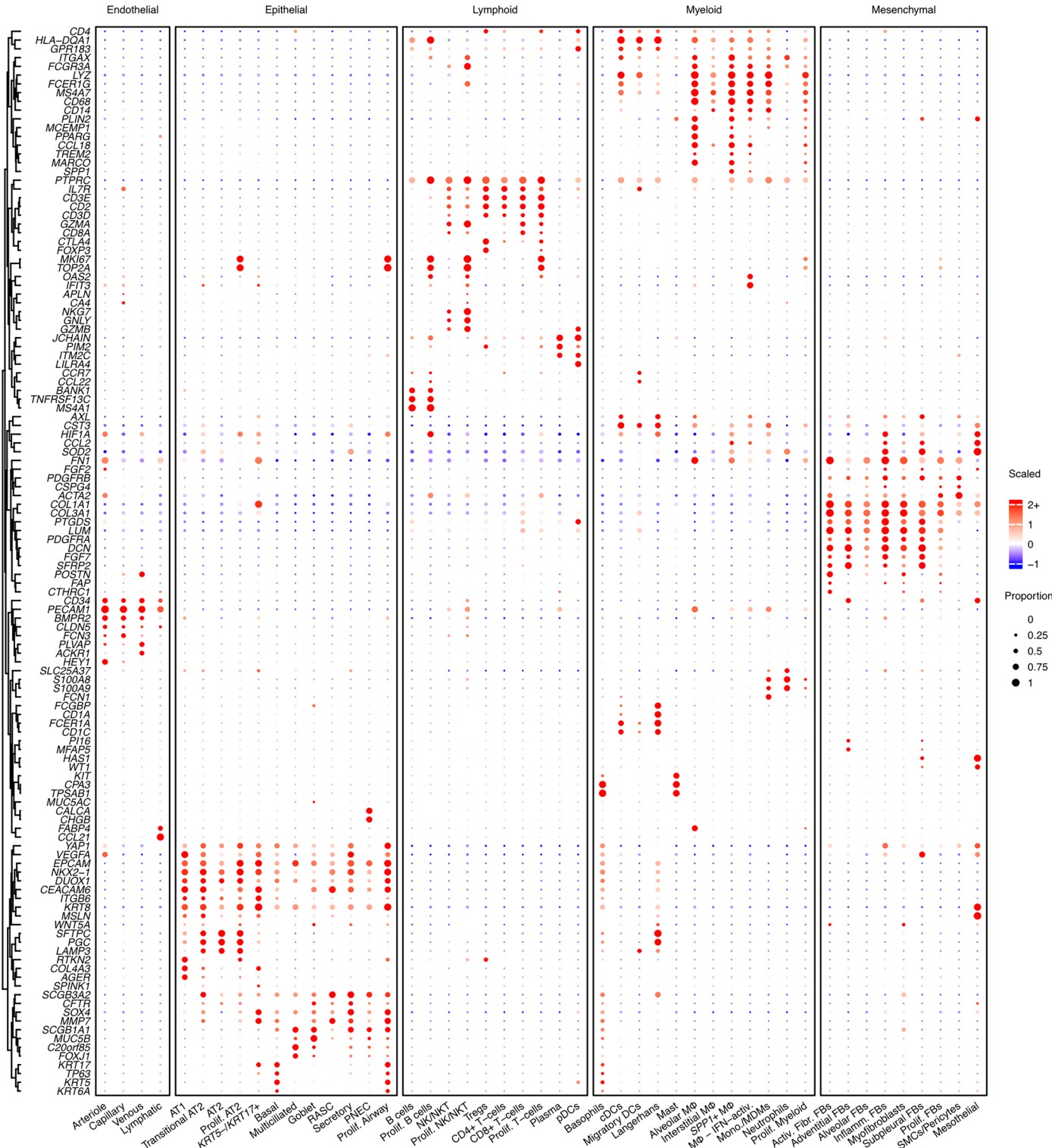

**Extended Data Fig. 1 | Marker genes for all cell types.** Of the 343 genes, 122 select genes are shown here to demonstrate cell-type annotation of the 47 identified cell types.

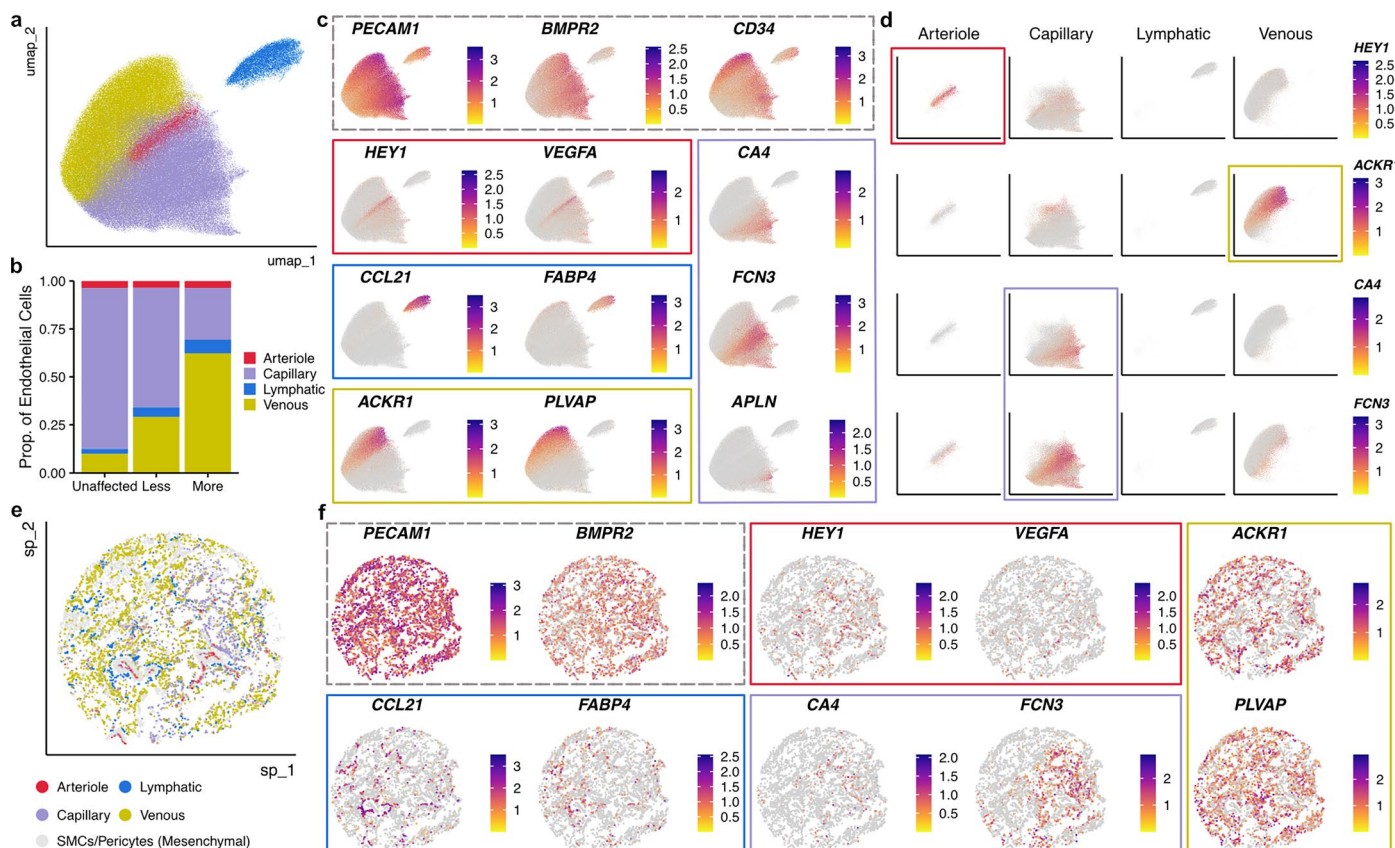

**Extended Data Fig. 2 | Cell-type annotation for the endothelial lineage using UMAP plots and spatial data. a**, Lineage-level UMAP for endothelial cells generated after final cell-type annotation. Cell types are colored based on the legend in **b**. **b**, Proportion of the total number of endothelial cells by disease state (unaffected, less affected or more affected). Capillary cells are the majority endothelial cell type in unaffected samples, but capillary cells are lost in PF. **c**,**d**, Select cell-type gene markers visualized in UMAP space across all endothelial cells (**c**) and split by cell type (**d**). **e**, An example spatial plot of sample

VUILD107MF (IPF diagnosis). Each point represents a cell, colored by cell type. Smooth muscle cells (SMCs)/pericytes of the mesenchymal lineage are included to provide additional spatial context for the location of endothelial cells. **f**, Plots of the same sample showing expression of select endothelial cell marker genes. Only endothelial cells are included in these plots. For **d** and **f**, boxes indicate the cell type of the gene marks. Gray-dashed lines indicate general endothelial lineage markers.

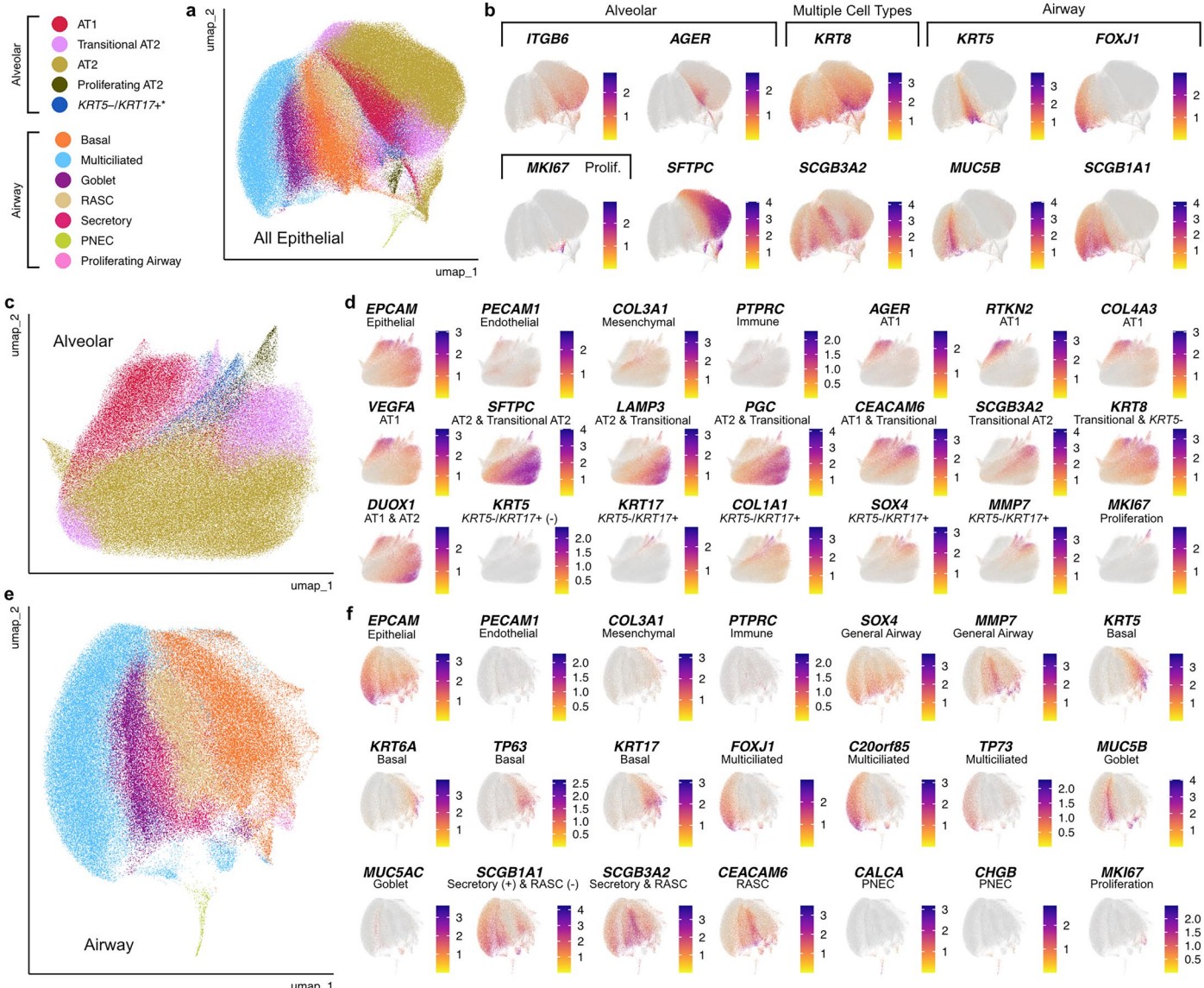

**Extended Data Fig. 3 | Cell-type annotation for the epithelial lineage using UMAP plots. a**, Lineage-level UMAP for epithelial cells generated after final cell-type annotation. **b**, Select cell-type gene markers visualized in UMAP space across epithelial cells. **c,e**, UMAPs were additionally visualized for alveolar (**c**) and airway cell types (**e**) separately. Please note that *KRT5⁻/KRT17⁺* cells were arbitrarily grouped with alveolar cell types, although they are found in PF in both remodeled airways and alveoli. **d,f**, Specific gene markers for alveolar (**d**) and airway cell types (**f**) visualized in UMAP space. For **b**, **d** and **f**, the specific cell types or groups of cells marked by each gene are listed near the gene name. 'Transitional' refers to general transitional epithelial marker genes, except where noted as a specific marker for only the transitional AT2 cell type.

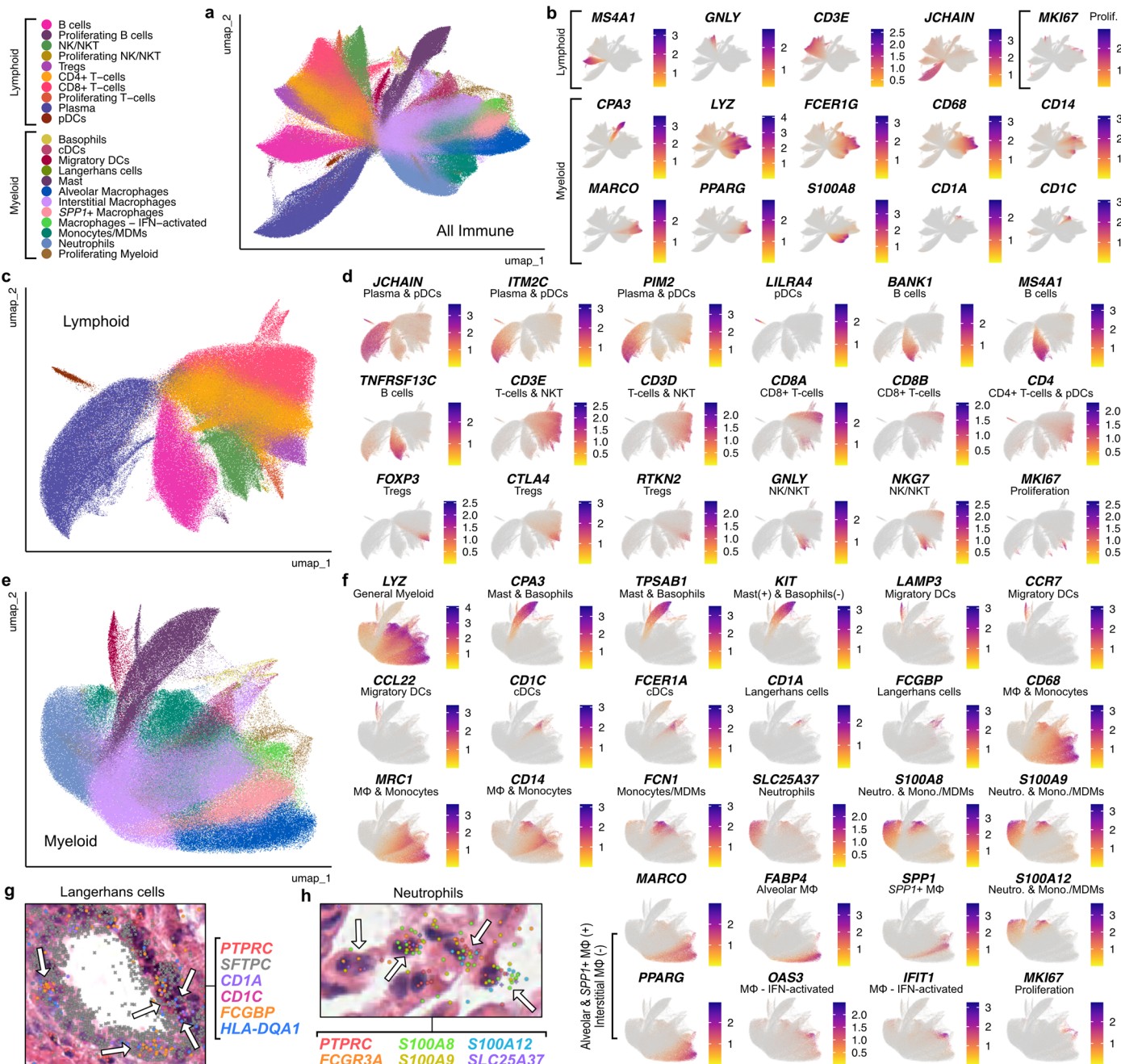

**Extended Data Fig. 4 | Cell-type annotation for the immune lineage using UMAP plots and spatial data. a**, Lineage-level UMAP for immune cells generated after final cell-type annotation. **b**, Select cell-type gene markers visualized in UMAP space across immune cells. **c,e**, UMAPs were additionally visualized for lymphoid (**c**) and myeloid cell types (**e**) separately. **d,f**, Specific gene markers for lymphoid (**d**) and myeloid cell types (**f**) visualized in UMAP space. For **b, d** and **f**, the specific cell types or groups of cells marked by each gene are listed near the gene name. **g,h**, Individual examples of Langerhans cells (**g**) and neutrophils (**h**)

indicated by white arrows with individual transcripts overlain onto H&E tissue stains. **g**, Langerhans cells in sample TILD117MA1 (IPF diagnosis) expressing immune marker *PTPRC*, dendritic cell markers *CD1C* and *HLA-DQA1*, and specific Langerhans cell markers *CD1A* and *FCGBP* in a remodeled airspace marked by *SFTPC*. **h**, Neutrophils in unaffected sample VUHD116B expressing general myeloid/monocytic markers *FCGR3A*, *S100A8*, *S100A9* and *S100A12* in addition to specific neutrophil marker *SLC25A37*. See Extended Data Fig. 1 for a dotplot heatmap showing expression of marker genes in each cell type.

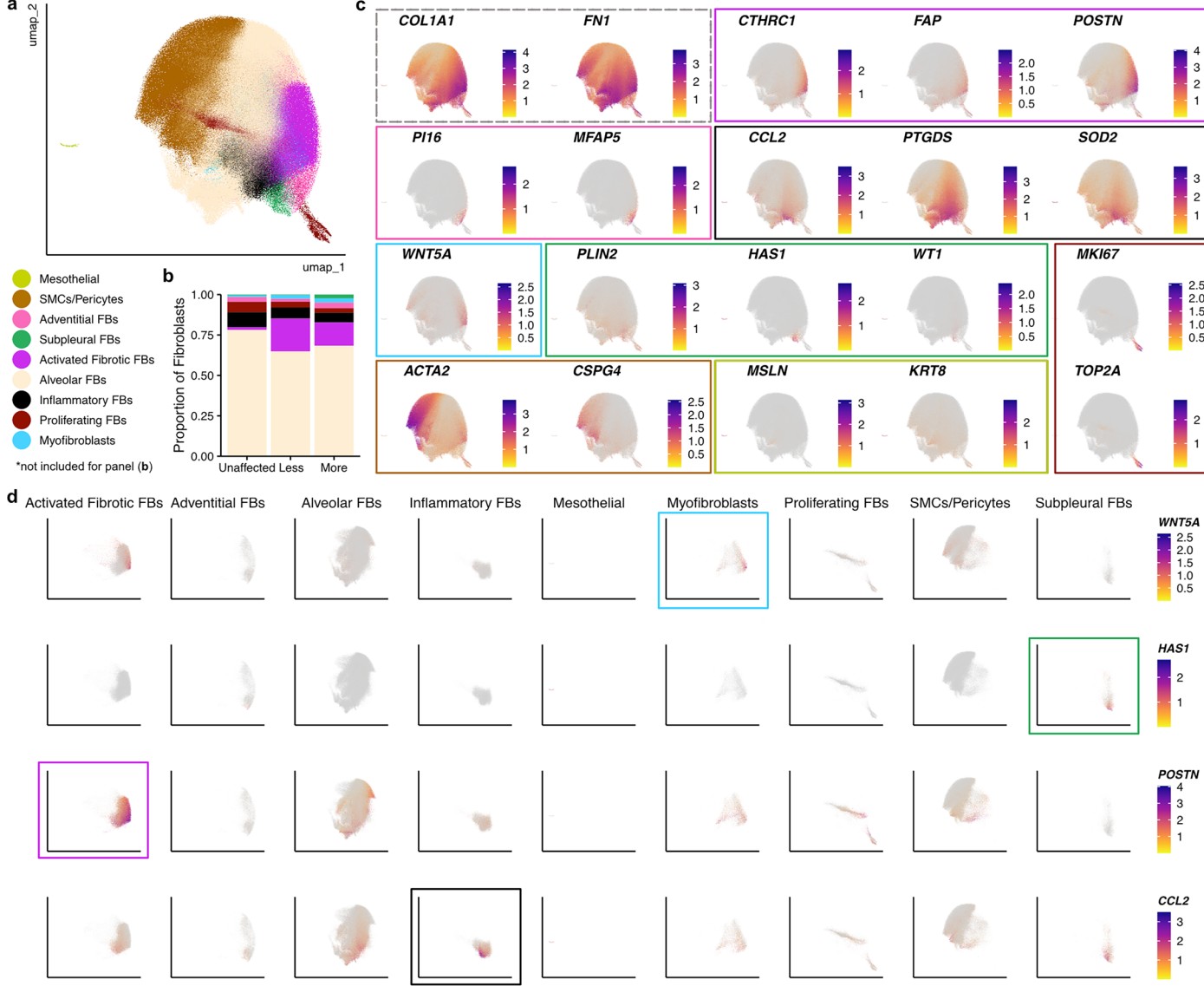

**Extended Data Fig. 5 | Cell-type annotation for the mesenchymal lineage using UMAP plots. a**, Lineage-level UMAP for mesenchymal cells generated after final cell-type annotation. **b**, Proportion of the total number of mesenchymal cells by disease state (unaffected, less affected or more affected). Alveolar fibroblast cells are the majority mesenchymal cell type across all groups, although the proportion of disease-associated, activated fibrotic fibroblasts increases in PF. **c,d**, Select cell-type gene markers visualized in UMAP space across all mesenchymal cells (**c**) and split by cell type (**d**). For **c** and **d**, boxes indicate the cell type the gene marks. Gray-dashed lines indicate general mesenchymal lineage markers.

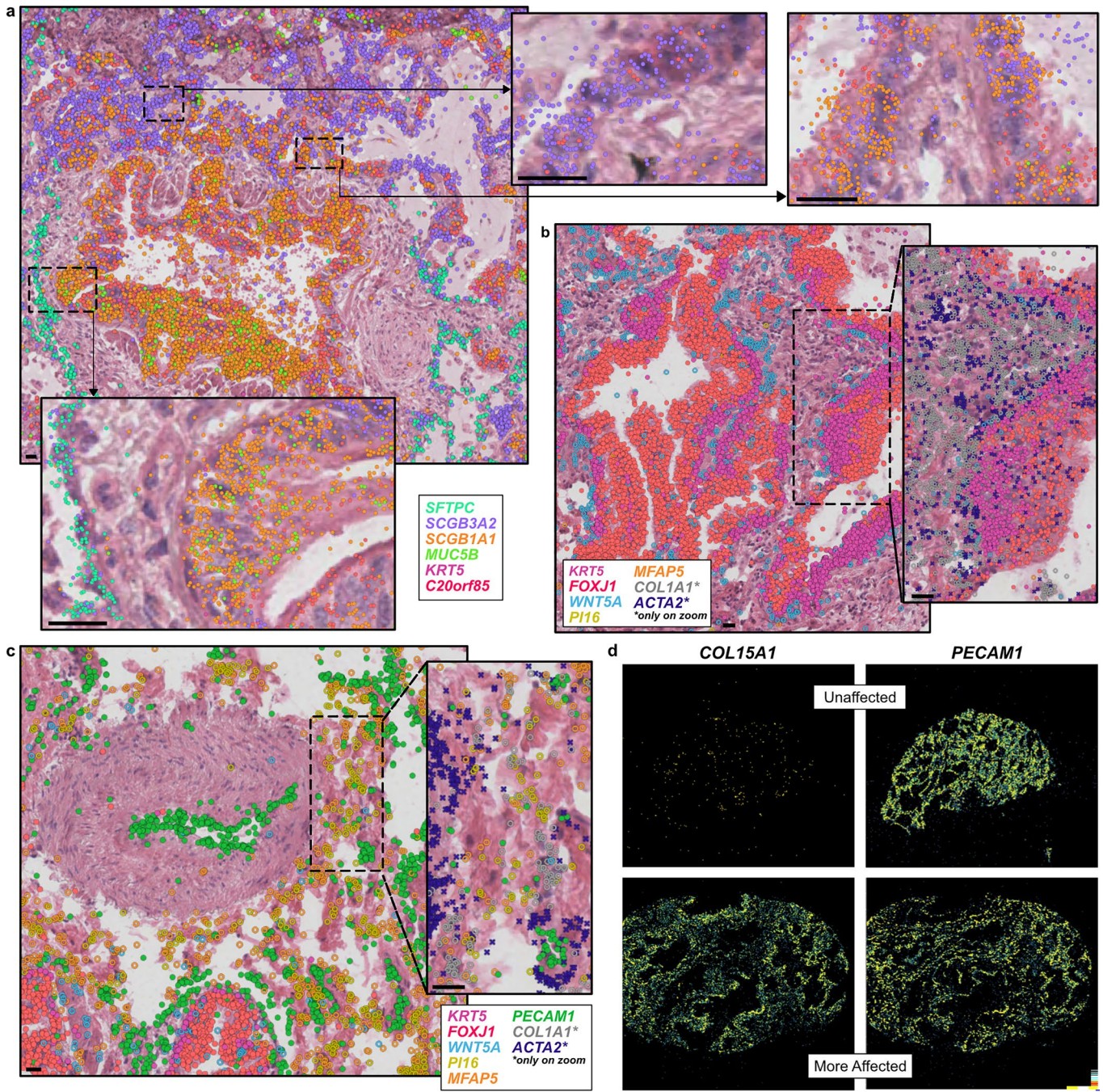

**Extended Data Fig. 6 | Localization of key cell types. a–c,** Representative examples of (**a**) pathologic *SCGB3A2*⁺/*SCGB1A1*⁻ airways near airways with mixed populations of *SCGB3A2*⁻/*SCGB1A1*⁺ and *SCGB3A2*⁺/*SCGB1A1*⁺ (RASC) dual-positive cells and *SCGB3A*⁺/*SFTPC*⁺ (transitional AT2) cells in remodeled alveoli; (**b,c**) *WNT5A*⁺ myofibroblasts surrounding conducting airways; (**c**) *PI1*⁺/*MFAP5*⁺ adventitial fibroblasts near vasculature. Transcript expression is overlain on H&E images. **d,** Density plots showing expression of disease-enriched endothelial marker *COL15A1* and pan-endothelial marker *PECAM1* in representative unaffected and PF samples. Samples depicted are as follows: (**a**) VUILD78MA (IPAF diagnosis), (**b–d**) TILD175MA (IPF) and (**d**) VUHD095 (unaffected). Scale bars represent 20 μm.

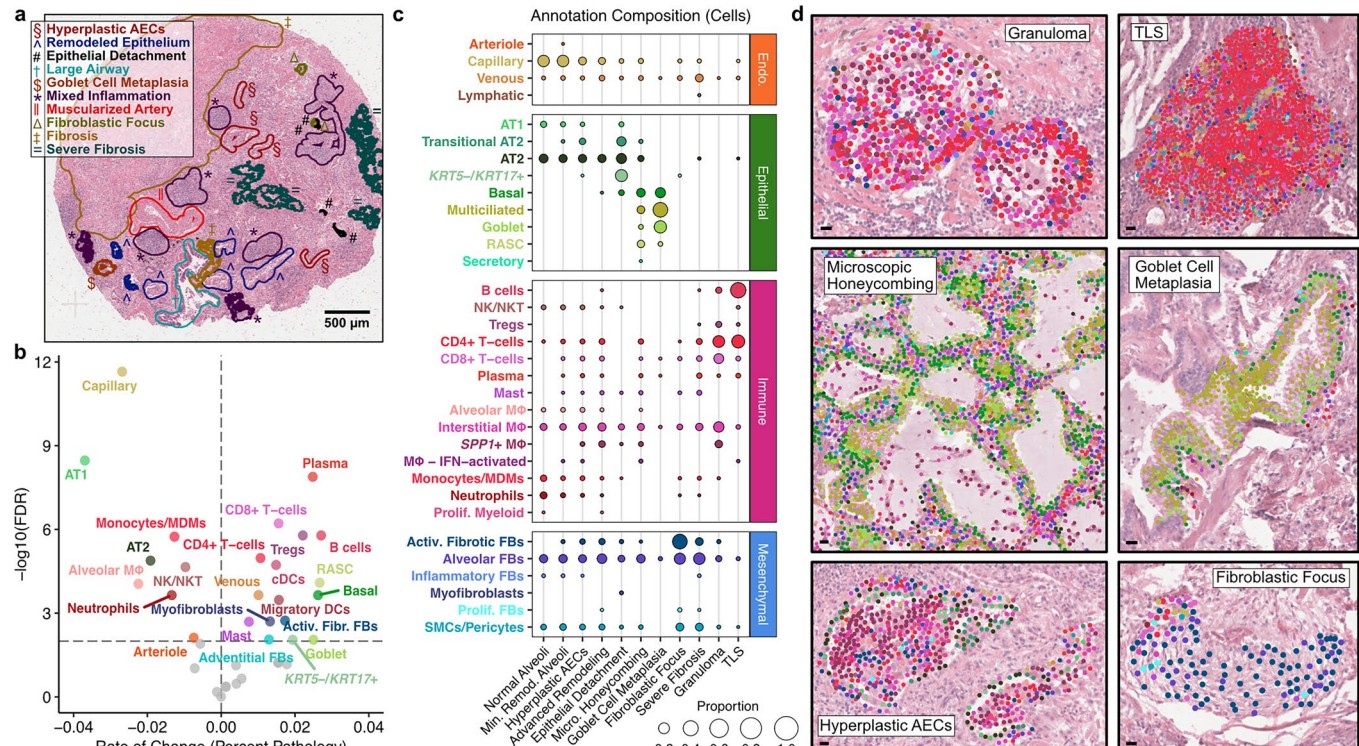

**Extended Data Fig. 7 | The molecular and cellular basis of clinically relevant PF histopathologies. a**, Each sample was assigned a pathology score based on a semiquantitative scale assessing the number/severity of pathologic features. Here we show a representative sample (VUILD107MA; IPF diagnosis) with annotated features. Features are represented by colors and symbols as shown: §, hyperplastic airway epithelial cells (AECs); ^, remodeled epithelium; #, epithelial detachment; †, large airway; $, goblet cell metaplasia; *, mixed inflammation; ‖, muscularized artery; Δ, fibroblastic focus; ‡, fibrosis; =, severe fibrosis. **b**, Cell types significantly associated with pathology score are labeled in the volcano plot. The horizontal and vertical dashed lines show the significance threshold (FDR < 0.01) and split the plot into cell types negatively and positively associated with pathology score, respectively. Cell types on the right are present in higher proportions in samples with high pathology scores. **c**, The cell-type composition of select annotations of interest, as a proportion of the number of cells across an annotation (each column sums to 1; columns indicated by gray lines). See Supplementary Fig. 17a for an expanded version of this plot with all cell types and annotations. **d**, Examples of select annotations on semi-transparent H&E images overlain with cell types in the annotated region. Each point represents a cell centroid. Cell-type colors are matched to **b**. Scale bars represent 20 µm. Example annotations were taken from the following samples: granuloma–VUILD96MA (sarcoidosis); tertiary lymphoid structure (TLS)–VUILD110LA (CTD-ILD); microscopic honeycombing (partial)–VUILD78MA (IPAF); goblet cell metaplasia–VUILD104MA2 (IPF); hyperplastic AECs–VUILD107MA (IPF); fibroblastic focus–VUILD104MA1 (IPF).

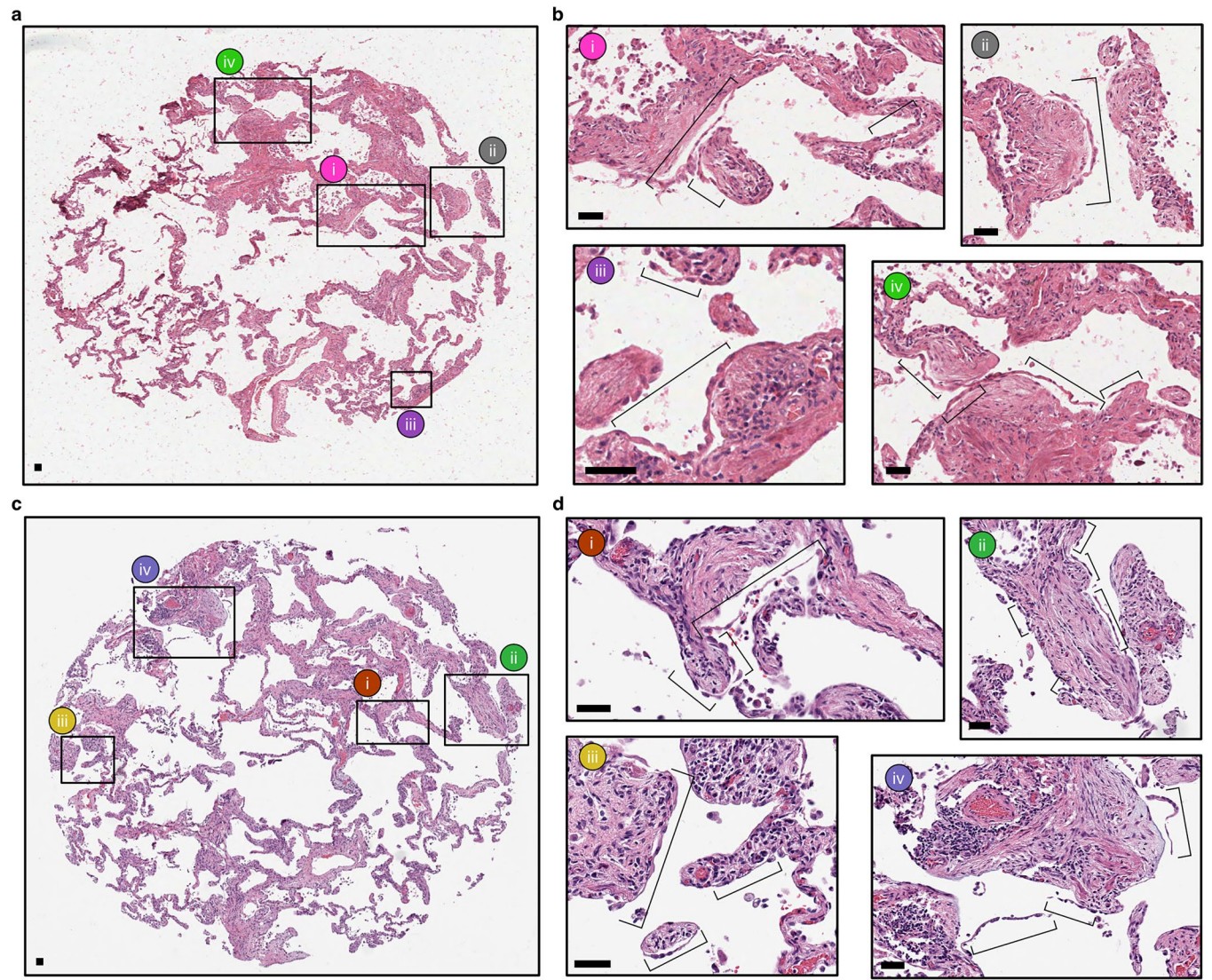

**Extended Data Fig. 8 | Epithelial detachment on a serial section that did not undergo Xenium processing.** To verify that the epithelial detachment feature was not an artifact of Xenium processing, we took serial sections approximately 45 μm from the processed tissue section. H&E stains, shown here for sample VUILD48LA (NSIP diagnosis), show areas of epithelial detachment are present on both the original section (**a**,**b**) and the section that did not undergo Xenium processing (**c**,**d**). **b** and **d** show the areas marked with boxes in **a** and **c**, respectively. Brackets denote epithelial detachment. Scale bars represent 50 μm.

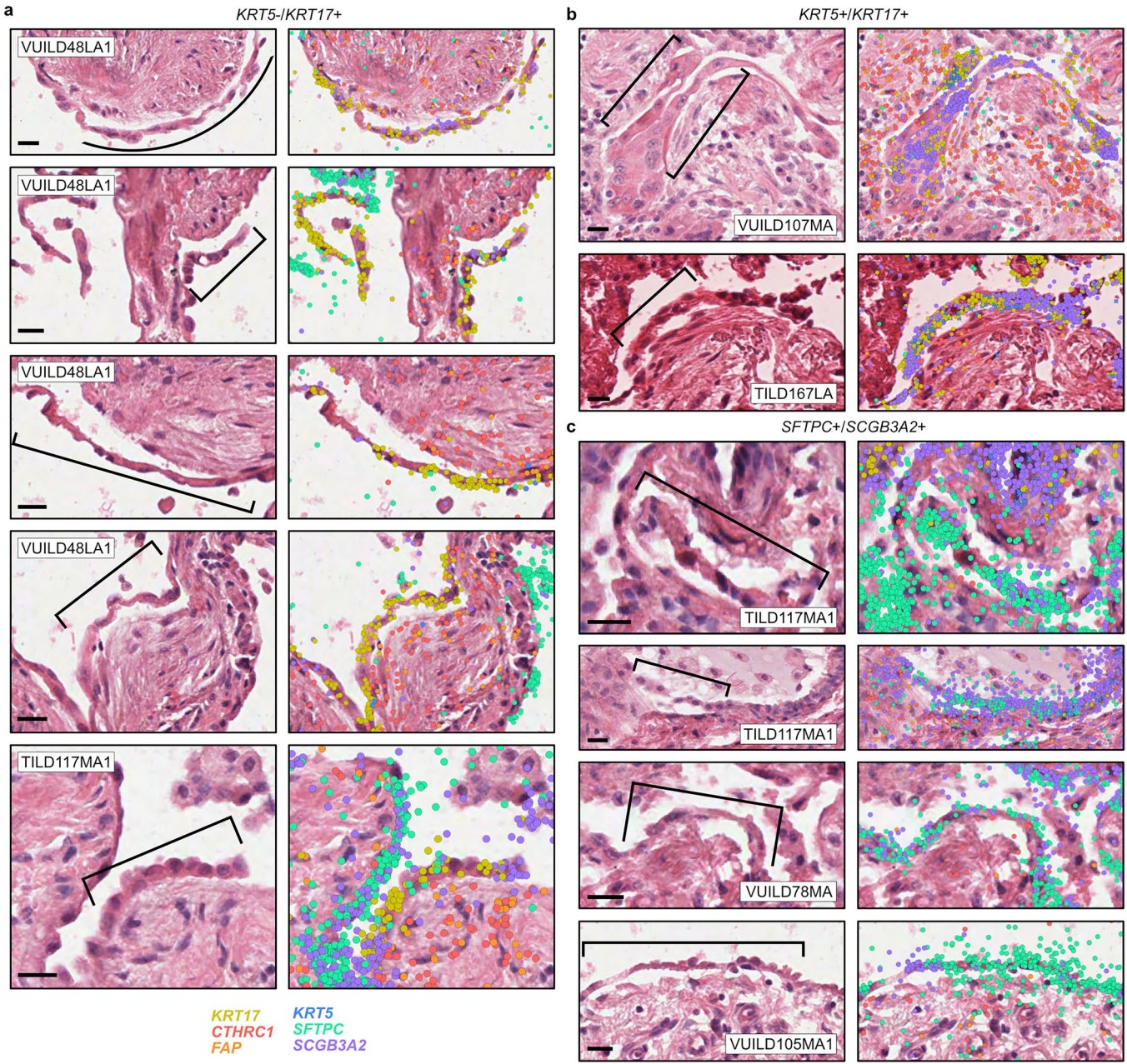

**KRT17** **KRT5**
**CTHRC1** **SFTPC**
**FAP** **SCGB3A2**

**Extended Data Fig. 9 | Additional examples of epithelial detachment.** The majority of epithelial detachment annotations contained *KRT5⁻/KRT17⁺* cells (**a**), basal (*KRT5⁺/KRT17⁺*) cells (**b**) and AT2-like (*SFTPC⁺/SCGB3A2⁺*) cells (**c**). A select set of examples are shown for these groups. For each example, an H&E stain is shown with brackets denoting areas of epithelial detachment, side-by-side with transcript expression for the listed genes overlain on H&E images. Scale bars in the bottom left of each H&E image represent 20 µm. Sample names for the depicted annotations are noted on each H&E.

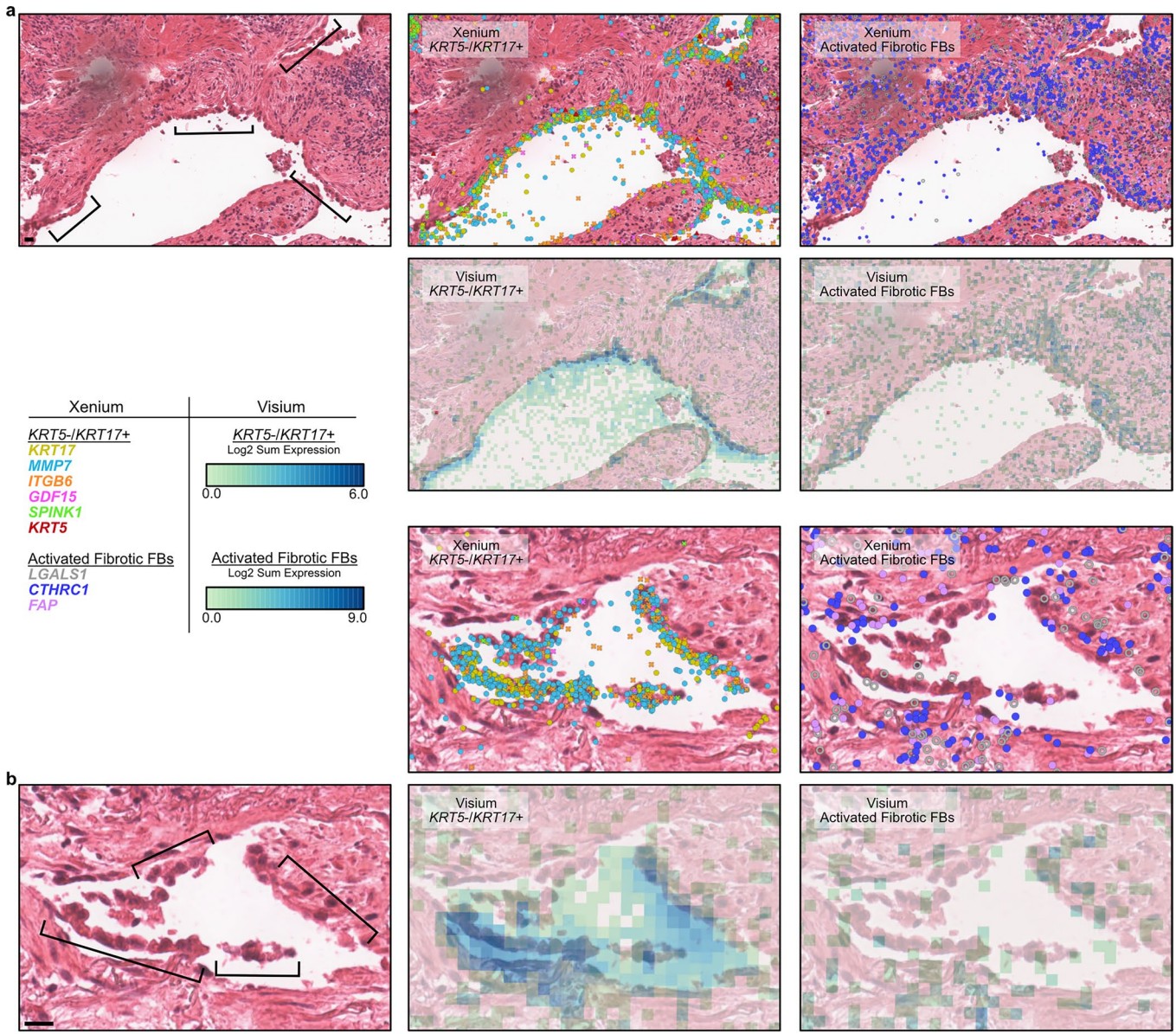

**Extended Data Fig. 10 | KRT5⁻/KRT17⁺ cell detachment in matched Xenium and Visium HD data. a,b,** Examples of *KRT5⁻/KRT17⁺* cells detaching from the basement membrane are shown for two samples (**a**, VUILD49LA−cHP diagnosis; **b**, TILD113LA−IPF) that were processed with both Xenium and Visium HD. For each panel, an H&E stain is shown with brackets denoting areas of epithelial detachment. Scale bars represent 20 μm. For Xenium images (top), select transcripts are shown to mark *KRT5⁻/KRT17⁺* cells (left) and activated fibrotic fibroblasts (right). Each transcript is a point or other shape, colored by gene as listed in the legend. For Visium HD images (bottom), the sum of the log₂ expression of a list of genes (Supplementary Table 11) is shown as a density map overlain on the H&E.

# Reporting Summary

## Statistics

For all statistical analyses, confirm that the following items are present in the figure legend, table legend, main text, or Methods section.

| n/a | Confirmed | |
|---|---|---|
| ☐ | ☒ | The exact sample size (*n*) for each experimental group/condition, given as a discrete number and unit of measurement |
| ☐ | ☒ | A statement on whether measurements were taken from distinct samples or whether the same sample was measured repeatedly |
| ☐ | ☒ | The statistical test(s) used AND whether they are one- or two-sided *Only common tests should be described solely by name; describe more complex techniques in the Methods section.* |
| ☐ | ☒ | A description of all covariates tested |
| ☐ | ☒ | A description of any assumptions or corrections, such as tests of normality and adjustment for multiple comparisons |
| ☐ | ☒ | A full description of the statistical parameters including central tendency (e.g. means) or other basic estimates (e.g. regression coefficient) AND variation (e.g. standard deviation) or associated estimates of uncertainty (e.g. confidence intervals) |
| ☐ | ☒ | For null hypothesis testing, the test statistic (e.g. *F*, *t*, *r*) with confidence intervals, effect sizes, degrees of freedom and *P* value noted *Give P values as exact values whenever suitable.* |
| ☒ | ☐ | For Bayesian analysis, information on the choice of priors and Markov chain Monte Carlo settings |
| ☐ | ☒ | For hierarchical and complex designs, identification of the appropriate level for tests and full reporting of outcomes |
| ☐ | ☒ | Estimates of effect sizes (e.g. Cohen's *d*, Pearson's *r*), indicating how they were calculated |

*Our web collection on statistics for biologists contains articles on many of the points above.*

## Software and code

Policy information about availability of computer code

| Data collection | Xenium instrument versions 1.1.2.4 and 2.0.1.0<br>Xenium Analyzer versions xenium-1.1.0.2 and xenium-2.0.0.10<br>Xenium Explorer 3.0.0<br>CytAssist instrument version 2.0.1.6<br>10X Genomics Loupe Browser v8.0.0<br>10X Genomics SpaceRanger 3.0.0 |
|---|---|
| Data analysis | Github code link: https://github.com/Banovich-Lab/Spatial_PF<br><br>ImageJ 2.14.0<br>RStudio Server 2023.06.0+421 "Mountain Hydrangea" Release and RStudio Server 2023.12.1+402 "Ocean Storm" Release<br>R 4.3.0-4 and 4.2.1<br>Python 3.9.7 and 3.12.3<br>Seurat 5.0.1<br>Scanpy v1.8.1<br>RAPIDS v21.8.1<br>RAPIDS cuCIM version 22.12.00<br>CuPy version 11.2.0<br>scikit-image version 3.9.7<br>Alpha shape version 1.3.1<br>QuPath version 0.4.3 |

```
GraphSAGE implementation in stellargraph(v1.2.1+python3.8.0)
PyCave (python3.8.4+pycave3.2.1)
propeller 0.99.0
limma 3.50.1
SingleCellExperiment 1.16.0
tradeSeq 1.8.0
kernlab 0.9-32
```

For manuscripts utilizing custom algorithms or software that are central to the research but not yet described in published literature, software must be made available to editors and reviewers. We strongly encourage code deposition in a community repository (e.g. GitHub). See the Nature Portfolio guidelines for submitting code & software for further information.

## Data

Policy information about availability of data

All manuscripts must include a data availability statement. This statement should provide the following information, where applicable:
- Accession codes, unique identifiers, or web links for publicly available datasets
- A description of any restrictions on data availability
- For clinical datasets or third party data, please ensure that the statement adheres to our policy

All data from this study are fully available as supplementary tables or in raw/processed format on GEO: GSE250346. This includes all image data and necessary accompanying files needed to interact with data outputs on the 10X Genomics Explorer software.

## Research involving human participants, their data, or biological material

Policy information about studies with human participants or human data. See also policy information about sex, gender (identity/presentation), and sexual orientation and race, ethnicity and racism.

| | |
|---|---|
| Reporting on sex and gender | Clinically annotated genetic sex was available for 34 of the 35 individuals in the study. Of the reported sex, 14 were female and 20 were male. |
| Reporting on race, ethnicity, or other socially relevant groupings | Self reported race/ethnicity was available for 34 of the 35 individuals in the study. Of the reported race/ethnicities, 3 reported as African American, 1 as American Indian, and 1 as Hispanic. The remaining subjects reported European ancestry. |
| Population characteristics | Of the 35 samples, 9 were individuals unaffected by lung disease (declined organ donors). The remaining 26 samples had a form of pulmonary fibrosis including IPF (n = 12), ILD (n = 4), CTD-ILD (n = 2), cHP (n = 4), CPFE (n = 1), NSIP (n = 1), IPAF (n = 1) and Sarcoidosis (n = 1). Ages ranged between 31-64 for controls and 47-70 for pulmonary fibrosis patients. Approximately half (48.6%) of PF patients reported current or prior tobacco use. |
| Recruitment | Participants were patients of the Clinical Investigator scheduled for a lung transplant surgery. The Clinical Investigator or a member of his research staff approached individuals to discuss the study and invite them to participate. |
| Ethics oversight | Studies were approved by the local Institutional Review Boards (Vanderbilt IRB nos. 060165 and 171657 and Western IRB no. 20181836) |

Note that full information on the approval of the study protocol must also be provided in the manuscript.

# Field-specific reporting

Please select the one below that is the best fit for your research. If you are not sure, read the appropriate sections before making your selection.

☒ Life sciences ☐ Behavioural & social sciences ☐ Ecological, evolutionary & environmental sciences

For a reference copy of the document with all sections, see nature.com/documents/nr-reporting-summary-flat.pdf

# Life sciences study design

All studies must disclose on these points even when the disclosure is negative.

| | |
|---|---|
| Sample size | The final sample size was 45 samples from 35 donors, consisting of 9 unaffected controls and 26 samples with pulmonary fibrosis. Sample size calculations were not performed. Sample size was chosen based on sample availability and the number of samples feasible to run together in the same Xenium runs. While this dataset contains a relatively small number of individuals, it is the largest imaging-based spatial transcriptomic study of the human lung reported to date and was sufficient to generate a large dataset containing more than 1.6 million cells. |
| Data exclusions | Cells with a large size, low number of counts, or unexpectedly high numbers of counts were filtered out based on current best practices. |
| Replication | Key findings from Xenium data were replicated with orthogonal technology - Visium HD. |
| Randomization | Data generation was randomized with respect to disease and control samples. |

| Blinding | This study was unblinded. Blinding would not be possible because major differences in histology are readily observable both between control and disease and between different disease diagnoses. |

# Reporting for specific materials, systems and methods

We require information from authors about some types of materials, experimental systems and methods used in many studies. Here, indicate whether each material, system or method listed is relevant to your study. If you are not sure if a list item applies to your research, read the appropriate section before selecting a response.

## Materials & experimental systems

| n/a | Involved in the study |
|-----|----------------------|
| ☒ | Antibodies |
| ☒ | Eukaryotic cell lines |
| ☒ | Palaeontology and archaeology |
| ☒ | Animals and other organisms |
| ☒ | Clinical data |
| ☒ | Dual use research of concern |
| ☒ | Plants |

## Methods

| n/a | Involved in the study |
|-----|----------------------|
| ☒ | ChIP-seq |
| ☒ | Flow cytometry |
| ☒ | MRI-based neuroimaging |

## Plants

| Seed stocks | *Report on the source of all seed stocks or other plant material used. If applicable, state the seed stock centre and catalogue number. If plant specimens were collected from the field, describe the collection location, date and sampling procedures.* |
|---|---|
| Novel plant genotypes | *Describe the methods by which all novel plant genotypes were produced. This includes those generated by transgenic approaches, gene editing, chemical/radiation-based mutagenesis and hybridization. For transgenic lines, describe the transformation method, the number of independent lines analyzed and the generation upon which experiments were performed. For gene-edited lines, describe the editor used, the endogenous sequence targeted for editing, the targeting guide RNA sequence (if applicable) and how the editor was applied.* |
| Authentication | *Describe any authentication procedures for each seed stock used or novel genotype generated. Describe any experiments used to assess the effect of a mutation and, where applicable, how potential secondary effects (e.g. second site T-DNA insertions, mosiacism, off-target gene editing) were examined.* |

