## [Peer Review File · Nature Genetics]

Spatial transcriptomics identifies molecular niche dysregulation associated with distal lung remodeling in pulmonary fibrosis

Corresponding Author: Professor Nicholas Banovich

Version 0:

Decision Letter:

21st Feb 2024

Dear Professor Banovich,

Your Article, "Image-based spatial transcriptomics identifies molecular niche dysregulation associated with distal lung remodeling in pulmonary fibrosis" has now been seen by 3 referees. You will see from their comments copied below that while they find your work of considerable potential interest, they have raised quite substantial concerns that must be addressed. In light of these comments, we cannot accept the manuscript for publication, but would be very interested in considering a revised version that addresses these serious concerns.

We hope you will find the referees' comments useful as you decide how to proceed. If you wish to submit a substantially revised manuscript, please bear in mind that we will be reluctant to approach the referees again in the absence of major revisions.

To guide the scope of the revisions, the editors discuss the referee reports in detail within the team, including with the chief editor, with a view to identifying key priorities that should be addressed in revision and sometimes overruling referee requests that are deemed beyond the scope of the current study. We hope that you will find the prioritised set of referee points to be useful when revising your study. Please do not hesitate to get in touch if you would like to discuss these issues further.

If you choose to revise your manuscript taking into account all reviewer and editor comments, please highlight all changes in the manuscript text file. At this stage we will need you to upload a copy of the manuscript in MS Word .docx or similar editable format.

*2) If you have not done so already please begin to revise your manuscript so that it conforms to our Article format instructions, available here. Refer also to any guidelines provided in this letter.

Please be aware of our guidelines on

digital image standards.

Link Redacted

If you wish to submit a suitably revised manuscript we would hope to receive it within 6 months. If you cannot send it within this time, please let us know. We will be happy to consider your revision so long as nothing similar has been accepted for publication at Nature Genetics or published elsewhere. Should your manuscript be substantially delayed without notifying us in advance and your article is eventually published, the received date would be that of the revised, not the original, version.

Nature Genetics is committed to improving transparency in authorship. As part of our efforts in this direction, we are now requesting that all authors identified as 'corresponding author' on published papers create and link their Open Researcher and Contributor Identifier (ORCID) with their account on the Manuscript Tracking System (MTS), prior to acceptance. ORCID helps the scientific community achieve unambiguous attribution of all scholarly contributions. You can create and link your ORCID from the home page of the MTS by clicking on 'Modify my Springer Nature account'. For more information please visit please visit www.springernature.com/orcid.

Thank you for the opportunity to review your work.

Sincerely,

Safia Danovi
Editor
Nature Genetics

Referee expertise:

Referee #1: ST, lung

Referee #2: IPF, single-cell transcriptomics

Referee #3: lung fibrosis

Reviewers' Comments:

Reviewer #1:

Remarks to the Author:

In their manuscript Vannan et al present image-based transcriptomics of IPF samples and the computational analysis of these data sets. Using the Xenium platform they examine a total of 28 lung tissue samples from 6 unaffected donors and 13 IPF donors. For 9 of the IPF donors, samples from more or less fibrotic regions were analysed. In total the authors examine gene expression for 343 transcripts and annotated a total of 39 different cell types on a total of 1.1 million nuclei, including aberrant cell types expected in IPF.

They report substantial changes in cell type composition with disease, and disease associated histopathology, largely in line with previous studies. For transcript niche identification the authors used graph neural networks without cell segmentation. In addition, the authors also used cell-based niche identification. Whilst most of the analysis seems to be based on published methods (GraghSAGE) it was nice to see these two complementary analyses. Having identified these niches, the authors were then able to compare cell type abundances across niches and made some surprising findings: For example, fibrotic foci were not associated with aberrant basaloid cells, but aberrant basaloid cells were associated with hyperplastic aerocytes and epithelial detachment. Aberrant basaloid cells were also found in association with SFTPC+/SCBG3A2 remodelled airspaces as expected. Honeycombing was associated with mixed airway and alveolar cell types. As expected, fibroblasts were most abundant in regions associated with severe fibrosis and fibroblastic foci. Overall, their analysis gives a very detailed picture of cellular landscape of the remodelled IPF lung, with different niches capturing specific interaction between different cell types.

Overall, the findings are well supported by the presented data, and the computational pipeline looks sensible to me. However, it is striking that two other recent paper preprints concluded that aberrant basaloid cells are part of the fibroblastic foci/fibrotic niche. The preprints (<https://www.biorxiv.org/content/10.1101/2023.12.21.572330v1>; <https://www.biorxiv.org/content/10.1101/2023.12.13.571464v1>) use lower resolution methods (Visium) so in that sense these

are not contradictory findings. Nevertheless, I feel the authors need to comment on this and either provide some additional validation, for example absence of KRT17 staining in fibrotic foci, or possibly refer to other studies that may already be able to provide evidence that is in line with their conclusions.

The novel observation that aberrant basaloid cells are associated with epithelial detachment is already well documented in the the figures. (e.g. Fig4e-g)

Furthermore, the authors use their data to show that different macrophage subtypes are associated with different niches , a very interesting finding; although there is some overlap of niches.

Lastly, the authors identified lumens likely to have originated from alveoli and then ordered these airspaces based on cell type composition, from most normal to most disease, thus creating a pseudotime of disease progression. In a sense check this aligned with a progression from unaffected to low to more fibrosis. This allowed the authors to define early and late events in disease progression. Placing macrophages into this timeline showed that FABP4 macrophage were present in pseudotime before SPP1+macrophages, which appear to be associated with late events.

Comment: As part of this analysis the authors show a differential gene expression analysis Fig 6d. could the authors comment on the pathways that are differentially expressed, rather than just genes?

Overall, this study suggests that disruption of the alveolar epithelium and adjacent capillary network is observed first, before fibrotic remodelling, and that activated fibroblasts and infiltration by different subsets of macrophages are late events. These are original results, they are clearly presented and supported by the data.

All of this already makes a very nice paper, but it is a shame that the authors did not use their rich data to examine cell-cell interaction networks. They beautifully define niches with distinct cell populations, that follow a pseudotime. Would it not be possible to use the matching single cell data to make predictions about the most likely interactions that are disease associated and likely to drive the pathologic process?

Reviewer #2:

Remarks to the Author:

General Comments

This is an interesting manuscript that describes initial efforts to use the Xenium platform for spatial transcriptomics on sections from normal lungs and the lungs of patients with interstitial lung diseases. The authors used a standard Xenium package including probes for 246 "lung" genes together with 97 genes chosen by the investigators to evaluate 21 samples from 12 patients with pulmonary fibrosis (7 with IPF and 5 from patients with a variety of other fibrosing interstitial lung disease) and 7 from individual normal lungs. For each sample the authors performed spatial transcriptomics followed by staining with Hematoxylin and Eosin. They then used a variety of bioinformatic approaches to identify cell types/states of interest that had been previously described based on single cell RNA sequencing and the relationships between those cells and areas of specific pathology identified by lung pathologists. Data linking multiple recently identified disease-associated cell states to specific areas of distinct pathology could fill an important current knowledge gap. Furthermore, spatial understanding of how and where these distinct cell states interact could generate new hypotheses about the steps underlying the development of pulmonary fibrosis, an important but currently poorly treated set of diseases with considerable morbidity and mortality.

However, there are a number of shortcomings to the current manuscript that limit the impact of the work described. Importantly, the authors don't provide sufficient detail for readers to understand how they decided what to call each cell and how well their calling method behaved. Most of the figures that show spatial data directly are shown at insufficient magnification for readers to evaluate co-expression of multiple genes (or even two genes) by any single cell. Dot blots in the supplemental figures clearly show that most of the genes analyzed (as expected) were detected in multiple cell types. Somewhat surprisingly, many of the genes that would be expected to be restricted to one broad cell category (like CD3 that should be restricted to T cells, and PDGFRA that should be restricted to fibroblasts) were listed as detected in cell types that have never been shown to express these genes by scRNAseq. Some of these problems might be due to acknowledged limitations of the 10x method for computational cell segmentation, but it would be important for the manuscript to show examples of how these errors might have occurred and what steps the authors took to mitigate them. Since this is the first study described utilizing their unique probe set, it would also be important to include high power images demonstrating cells that clearly co-express multiple genes that establish their unique identities.

Seurat software also allows users to generate uMAP plots from Xenium outputs which the authors say they used during an iterative process of re-clustering defined subsets of cells that the authors used to develop their system for identifying and counting individual cell states. Inclusion of these uMAPs would be very helpful for readers to evaluate how well the methods used in this paper really distinguish individual cell types or states. The one uMAP plot shown, in Figure 1, does not even seem to clearly distinguish among the major cell categories labeled (epithelial cells, endothelial cells, mesenchymal cells and immune cells).

Another limitation of this paper is that it does not seem to provide any novel insights into the biology underlying pulmonary fibrosis. Although the authors show examples (in figure 2) of 3 pathologic structures that are well known to occur in subsets of fibrotic diseases, fibroblastic foci, granulomas and tertiary lymphoid structures and also show examples of honeycombing,

goblet cell metaplasia and hyperplastic “alveolar” epithelial cells, there are no new insights into the cellular composition or organization of these structures than those previously described by standard immunostaining. Actually, since the authors don’t show images of the primary data with patterns of co-expression in individual cells in these structures, this paper contains less molecular information than many previous descriptions. It was also a bit disappointing that the small lung regions analyzed in total seemed to include only two fibroblastic foci, two tertiary lymphoid structures and a few large granulomas.

The Seurat algorithms for identifying cellular niches don’t add much to this paper, since this approach mainly identified groups of cells that are all the same class of cell, rather than “niches” in which multiple cell types interact. Although the GraphSAGE generated transcript-based “niches” did not completely overlap with the cell-based niches, the tendency was similar for them to mostly identify homogeneous groups of cells. At the very least, it would have been important for the authors to try to quantitatively evaluate close physical relationships among different cell types or the “niches” they identify. There are well-established methods by which they could have done that.

The authors spend quite a bit of space in the results and discussion describing what they think might be a novel finding – the presence of KRT17+/KRT5- epithelial cells overlying areas of detachment of the epithelium from the underlying tissue. As above, it would be helpful for the authors to show high power images demonstrating the expression of KRT17, the absence of KRT5 and the presence of other markers of so-called “aberrant basal cells” in these regions. More important is the concern that the detachment might be an artifact caused by detachment of epithelium from areas of stiff underlying tissue during embedding, sectioning and processing of these samples. At the very least it would be important to characterize the cells and genes in the regions that these epithelial cells were likely in contact with before processing. In this regard, it is important to note that previous publications have shown enrichment of these cells in the epithelium overlying fibroblastic foci. In the current manuscript the authors are likely to have missed this event because they only sampled two fibroblastic foci, and the one shown in Figure 2 itself seems to be artifactually separated from the surrounding tissue. Cell proximity calculations might at least have allowed the authors to determine if aberrant basaloid cells were likely to be in close proximity to aggregates of activated fibroblasts, as has been shown by others. However, the limitations of the authors tools to distinguish distinct fibroblast subsets (the vast majority of fibroblasts were simply lumped together as “fibroblasts”) might limit their ability to perform this analysis.

Specific comments

Figure 1 – As noted above, the uMAP plot shown does not appear to show clear separation of even the major cell categories labeled on the figure. The authors should explain how they chose to label specific regions of the uMAP as specific cell types, along with uMAPs showing overlay of a number of genes of interest that should be specific to specific cell types, ideally including uMAPs based on sub-clustering of each major cell type. That would provide more confidence that the labels are accurate.

Figure 4 – It is hard to see gene expression in individual cells with the large spots used to mark individual genes on relatively low power images. It would be important to show individual cells at much higher power so readers can see individual amplified transcripts and their relationship with individual nuclei and calculated cell boundaries. It is difficult to know what to make of the finding of cell detachment since from the supplementary figures it looks like very few regions of KRT17+/KRT5- cells were seen. As noted above, detachment might also be an artifact of fixation and sectioning.

Figure 5 – The structures called alveoli don’t really look like alveoli and the SFTPC signal seems to be present in stratified epithelial cells surrounding the lumens shown. In IPF, many of the macrophages are in the interstitium, but the sections shown seem to only show large clusters of macrophages within large lumens. Why aren’t more than single genes shown for each cell type, to allow readers to get a better sense of how cell types were named?

S2 – the cell numbers shown suggest some significant biases in cell naming. For example, although AT2 cells and AT1 cells are present in equal numbers in normal lungs, the authors identified more than 4 x more AT2 cells. This could be because AT1 cell numbers are reduced in fibrotic lungs, but it would be reassuring to show equal numbers in normal lungs. Previous scRNA sequencing and immunostaining have shown a reduction in AT2 cells in fibrotic lungs, but this paper suggests there is a large increase. I wonder if that is because of spill-over of the SFTPC signal, much as the authors saw for the highly expressed AT1 marker AGER (which I presume they ignored in their naming process). For mesenchymal cells, from scRNAseq in normal lungs most fibroblasts are alveolar fibroblasts and in fibrotic lungs they are mostly “activated” fibroblasts, with the remaining fibroblasts divided among peribronchial and adventitial. However, the overwhelming majority of fibroblasts shown are identified as just “fibroblasts” suggesting that much of the heterogeneity of this population of cells was missed or underestimated with the probes used for this study. This point should at least be acknowledged and discussed.

S3 - many of the cell types listed seem to have largely overlapping patterns of expression of the 70 genes shown within each broad cell type, so it would be important for the authors to show sections demonstrating with arrows how they distinguished among endothelial subsets, fibroblast subsets, epithelial subsets and lymphoid subsets. For example, differences were very subtle between peribronchial fibroblasts, fibroblasts and alveolar fibroblasts. Lumican was the only marker distinct for adventitial fibroblasts. It is not clear why proliferating mesenchymal cells aren’t called proliferating fibroblasts, since they don’t resemble mesothelial cells or SMCs at all. Where are pericytes? Did the authors not include any distinguishing markers?

S4 – only shows expression of individual genes described as showing specific fibroblasts in panels b and c and double

positive cells in panel a. However, at the resolution shown it is not possible to tell whether and if the cells are actually double positive and there are no other markers included to show that the cells positive for WNT5A or PI16 were actually fibroblasts. While PI16 is largely restricted to fibroblasts, the heat maps shown in supplementary figures suggest that WNT5A might be more broadly expressed. With 383 markers used, it would better for the authors to show better examples of how they integrated multiple markers to identify each cell type they say they can distinguish. Some uMAPs for subclusters of each major cell type showing how good the distinctions were and how the markers that characterized each cluster allowed separation would be especially helpful.

S6 – seems to show that there are no significant cell composition differences between normal and severe fibrosis. The only significant difference marked were for slides selected from regions thought to include mild fibrosis. Do the authors have any hypothesis for why this was the case? Did they not statistically compare severe fibrosis to normal?

S8 legend says *s represent significance, but I couldn't see any *s on any panel in the figure.

S9 – It is not clear what to make of these volcano plots. In nearly every cell type, AGER, which is mainly expressed in AT1 cells, appears to be upregulated in the normal lung in most cell types. It looks like PPARG, DUOX1, RTKN2 and MUC5B, at least, have similar patterns of non-specific expression. KRT17 shows up high in FABP2 macs in fibrosis but is not shown in KRT5-/KRT17+ epithelial cells, where the only gene highlighted in fibrosis is CXCL9, a chemokine mainly expressed by myeloid cells and fibroblasts.

S10 seems to confirm that very few nuclei were analyzed from fibroblastic foci, and areas of cellular detachment. Severe fibrosis, granulomas, areas of honeycombing and mixed inflammation and normal small airways and other structures predominated. Although there were only two TLS, many cells were analyzed from these two. One sample seems to have been composed entirely of a TLS and another TLS came from one other sample. It looks like both fibroblastic foci might have come from a single sample. It looks like the granulomas, that were extensively analyzed, came from just two samples, with one made up of more than 50% granulomas.

S11a seems to show that many of the annotated histologic features were not well separated from one another by gene expression, with considerable overlap among many of the groups although I may have misread this since some of the colors were not easy to distinguish. The two exceptions seem to be granulomas and to a lesser extent, severe fibrosis.

S11b seems to show ITGB6, an epithelial restricted gene, and COL1A2, a fibroblast gene, as markers of histopathologic regions of airway smooth muscle. The figure also seems to confirm that only two fibroblastic foci and two tertiary lymphoid structures were annotated from the entire dataset.

S11c shows high levels of PDGFRA and ACTA2 in what are called normal alveolar epithelial cells and high levels of MUC5AC in what are called normal alveolar epithelium. The figure also shows MRC1 (expressed mainly in macrophages) and MSLN (mainly in mesothelial cells), in what are called hyperplastic aecs and epithelial cells in areas of detachment and CD68 (a macrophage marker) in normal alveolar epithelial cells.

S13 WNT5a seems to be expressed at low levels in 6 of 12 cellular niches. Yet figure S4 uses this single gene to suggest that subepithelial fibroblasts are distributed around conducting airways. The T3 niche, which is described in the text as the aberrant basaloid cell niche, contains only a single lane (sample?) that is KRT17 high and KRT5 low. All of the other lanes in this niche seem to express equal levels of KRT17 and KRT5. How are these data consistent with the idea that this niche represents aberrant basaloid cells?

S19 – Shows that cell identities were either unreliable or assignment of expressed genes to specific cells was inaccurate. This figure includes numerous examples of patterns that are dramatically different from expression patterns shown from public RNAseq databases including:

PDGFRa and PDGFRb – very low in mesenchymal cells, high in all myeloid cells and some lymphocytes

CD4 in most mesenchymal cells and some myeloid cells

Col3a1 in lymphocytes

Krt6 in a population of mesenchymal cells and most lymphoid cells

Muc5ac high in some mesenchymal cells

Acta2 highest in all myeloid cells, not in mesenchymal cells, even smooth muscle

Fgf2 – mostly in myeloid cells

Itgb6 mostly in most mesenchymal cells and some myeloid cells

WT1 most highly expressed in myeloid cells and some lymphocytes.

Methods – the authors described iterative rounds of clustering and sub-clustering using marker genes including analyzing uMAP outputs to refine clusters as well as analysis of cell size and shape. They should show uMAP plots validating the final clusters since those would help readers to evaluate the validity of the annotation methods, ideally with overlay of multiple well-characterized genes described to characterize clusters previously identified by scRNAseq.

How did the authors decide that alveolar lumens could have up to 500 cells? These spaces seem too large to be defined as alveoli.

Reviewer #3:

Remarks to the Author:

Vannan et al used 97 genes from Haberman et al.'s (PMID: 32832598) plus a pre-existing probe set to measure gene expression in 19 lungs, both control and pulmonary fibrosis. The results as the authors themselves point out are circumscribed, because of the approach of using a small number of markers that were previously discovered, to at best a validation of cell types suggested by the previous scRNAseq work, now with spatial information added. Therefore, while the work complements the published scRNAseq data in terms of spatial information, I am concerned that it in any event merely recognizes structures that are well known to the lung pathology of fibrosis, such as fibroblastic foci and epithelial hyperplasia, etc.; not much was discovered by the study, nor were there surprises. Finally, I'm not sure how this will be a genetic resource, given the limited gene number. A hypothesis-driven gene set would at least uncover the activity of a given pathway or pathways, whereas the number of markers used here, and their apparent limitation to known overrepresented markers in fibrosis globally, limits depth. Some specific queries:

1) Sloughing of the epithelium: Even if this is not artifactual, it is not clear what was learned from reporting this phenomenon, which is anyway observable by standard H&E and as the authors mentioned, previously reported. Is there something about the sloughing that reveals new aspects of the pathophysiology, or is there a clue to its cause?

2) Airway macrophages: I am not clear on how the airway-localized macrophages could be important for progression of fibrosis in the parenchyma.

3) Disease heterogeneity: The sample contains IPF and non-IPF fibrosis. The authors should focus more analyses on the differences between these subgroups globally and with a focus on immune cell types and locations.

4) The pseudotime ordering provides little context for the reasons for the evolution within airways. Are there ligand-receptor pairs that could be interrogated by other methods to understand this progression, if the Xenium markers used are insufficient for this analysis? Can the data shed light on capillary dropout? The loss of interstitial macrophages in favor of later SPP1+ macrophages is based on markers, but it is not clear whether the cells that were there all along—the macrophages—had a physiological role in fibrosis progression but merely began to express the putative profibrotic marker SPP1 “late” in the pseudotime trajectory.

Version 1:

Decision Letter:

Our ref: NG-A64487R

11th Nov 2024

Dear Dr Banovich,

Thank you for submitting your revised manuscript "Image-based spatial transcriptomics identifies molecular niche dysregulation associated with distal lung remodeling in pulmonary fibrosis" (NG-A64487R). It has now been seen by the original referees and their comments are below. The reviewers find that the paper has improved in revision, and therefore we'll be happy in principle to publish it in Nature Genetics, pending minor revisions to comply with our editorial and formatting guidelines.

Sincerely,

Safia Danovi, PhD
Senior Editor, Nature Genetics
ORCID: 0009-0007-7822-5479

Reviewer #1 (Remarks to the Author):

In their revised manuscript, Vannan et al have made substantial changes. There are far more figures that indicate how cell types were annotated showing the levels of expression of marker genes, that now better match the expected expression patterns. (As pointed out as a problem by reviewer 2). In terms of the question I had raised, the authors mostly pointed out that they had tried similar analyses, but that the limited number of genes available within their data set did not allow extensive cell-cell interaction analysis or the imputation of gene expression from single cell data. Therefore it is reasonable that the authors do not present additional results on this front.

However the authors have added VisiumHD data to provide orthogonal validation for some of their conclusions and they find

that the Visium data aligns well with the identified cellular niches.

Reviewer #2 (Remarks to the Author):

The authors have done an outstanding job of responding to all the significant comments raised in my initial review. The paper is greatly strengthened by their correction of 34 mislabeled probes, by the new figures that more clearly show how cells were identified and by a more compelling analysis and description of the novel biology identified by the extensive dataset that serves as the basis for this manuscript. This manuscript should now be a terrific resource for a broad array of scientists studying tissue fibrosis.

Dean Sheppard

Reviewer #3 (Remarks to the Author):

In the revision, the authors have added data from more lungs, as well as some Visium HD data (mainly confirmatory), and have rectified errors in gene ID, which are important improvements. However, I remain concerned about the main approach having been to use the salient cell type-identifying features from the scRNAseq data of Haberman et al for 2D mapping/atlas-ing of the Haberman work. The problem is that these main features identify well-known structures, for the most part, such as fibroblastic foci or bronchovascular cell types (eg, adventitial fibroblasts).

It is true that this work will be valuable within the lung community (and within that, the lung fibrosis community) in laying a groundwork for future focused analyses by validating core marker genes and their associated 10x probes, to be included in panels that will address a more hypothesis-driven question (eg, a specific signaling pathway, etc).

The finding of the Krt5-/Krt17+ sloughing is intriguing and could underlying bronchiolization and ultimately honeycombing, and I regret only that there was no experimental hypothesis testing of the mechanism, even ex vivo.

Response to Referees

Referee expertise:

Referee #1: ST, lung

Referee #2: IPF, single-cell transcriptomics

Referee #3: lung fibrosis

To the editor and reviewers:

We want to thank all three reviewers for their careful and thorough assessment of our work. The feedback has greatly improved our manuscript. In particular, we thank reviewer #2 who correctly identified a number of strange expression patterns in our dataset. While we had also found a number of these results confusing, this was our first dataset using the 10X Xenium platform, and we initially attributed these observations to poor probe performance or other chemistry related issues. We had met with 10X Genomics prior to submission to review some of these issues and at the time felt reassured that these were accurate (albeit in some cases, partially unexpected calls). We did not want to prune or curate our data, and chose the tactic of being more permissive during filtering.

During revision we undertook a number of additional validations to ensure the robustness of our results. In these validation experiments, we compared results from the panel used to generate this dataset – a pre-commercial version of the human lung base panel plus 100 custom add-on genes – to the commercial human multi-tissue Xenium panel and an alternative sequencing-based spatial transcriptomic system, Visium HD. In these analyses, we observed several examples of discordant gene localizations between the panel used for this dataset and the other platforms when comparing adjacent tissue sections, including some key markers such as *ACTA2*. After extensive internal investigations, it was uncovered that 10X had provided an incorrectly coded JSON file which is used to assign genes to codeword indexes. Ultimately, this created a complicated series of round robin swaps for 35 of the genes on our custom panel. The instances of lineage restricted genes identified in incorrect cell types pointed out during the initial reviews were largely driven by this issue. We regret that we did not fully elucidate these issues before the initial submission, but are happy to report since this was a computational error – rather than a probe or chemistry error – we were able to correct these data. Furthermore, we have added an additional TMA of 17 samples to this data (these data were generated during our QC analyses) as well as Visium HD data for four samples. We have appended a table of probe mismatches from the original dataset below, as well as a letter from 10X Genomics outlining the source of the issue.

In addition to correcting this data-related issue and increasing our sample number by 50%, we have extensively revised this manuscript in response to the reviewer comments. Profiling >1.6M cells from 35 unique lungs including controls and a diversity of different forms of pulmonary fibrosis, including samples with a range of disease severity (not only end-stage findings), this is the largest and most representative lung spatial transcriptomic study we are aware of. We have refined our cellular annotation, added a new main figure (Figure 2) and several supplemental figures providing much greater “atlas-level” transcriptomic and spatial context that we anticipate will be a useful resource for the community. In response to the reviewer comments, we have also added new quantitative assessment of spatial proximities between cell types. We have also supplemented the specific pathologic feature assessment with larger numbers of examples of features. The addition of new samples and corrections of the codebook-related error has improved the cell-type specific differential expression results and airspace-segmentation based trajectory analyses. Cumulatively, we suggest these

revisions substantially improve this manuscript, offer new insights into the molecular evolution of pulmonary fibrosis and will be of high-interest for the lung biology and spatial-transcriptomics communities. We have responded to the additional reviewer requests inline below.

Table of mis-labeled genes

Original Label	Actual Gene
ACTA2	MRC1
APLNR	TGFB3
CCL18	GDF15
CCL21	HLA-DRA
CDKN2A	FGF2
CEACAM6	ELN
COL1A2	ACTA2
COL4A3	LAMP3
CSPG4	APLNR
ELANE	WT1
ELN	PDGFRB
FGF2	PLIN2
FGF7	HES1
FN1	COL1A2
GDF15	SFRP4
HES1	TGFB2
HLA-DRA	CCL21
ITGA3	FGF7
ITGAX	ITGB1
ITGB1	ITGA3
ITGB6	FN1
JCHAIN	MUC5AC
LAMP3	CDKN2A
LYZ	ITGB6
MRC1	COL4A3
MUC5AC	PPARG
PDGFRA	ITGAX
PDGFRB	LYZ
PLIN2	CCL18
PPARG	JCHAIN
SFRP4	CEACAM6
TGFB2	PDGFRA
TGFB3	CSPG4
TREM2	ELANE
WT1	TREM2

August 28, 2024

The Editorial Team
Nature Genetics

To Whom It May Concern:

RE: Banovich lab Xenium results in first submission of Nature Genetics paper

In the early stages of R&D, 10x Genomics worked with the Banovich lab to test a pre-commercial version of the Human Lung Xenium panel. In the process of delivering the decoding file that is necessary for the instrument to pair barcodes with genes and decode the panel, a mix-up by 10x Genomics led to the wrong file being provided to the Banovich lab. This meant that wrong codewords were assigned to 35 genes. Codewords are required to pair the signal from the probe barcode to the correct gene, and as such this error led to an erroneous interpretation of the data. However, this was purely a computational error, and the physical panel content (i.e. the probes used to target the genes of interest) were correct.

As soon as we were aware of this error, the 10x Genomics team worked with the Banovich lab to correct this computational error and provided a new decoding file with the corrected mapping of the codewords to the panel genes.

We apologize for any inconvenience caused and trust it will not impact the decision by Nature Genetics to accept the manuscript from the Banovich lab.

Please do not hesitate to contact me if you have any further questions.

Sincerely,

DocuSigned by:
Sarah Taylor
8D8038430BF24F5...

Sarah Taylor, Ph.D.

Senior Director, Applications
sarah.taylor@10xgenomics.com

[10xgenomics.com](https://www.10xgenomics.com)

Reviewer #1:

Remarks to the Author:

In their manuscript Vannan et al present image-based transcriptomics of IPF samples and the computational analysis of these data sets. Using the Xenium platform they examine a total of 28 lung tissue samples from 6 unaffected donors and 13 IPF donors. For 9 of the IPF donors, samples from more or less fibrotic regions were analysed. In total the authors examine gene expression for 343 transcripts and annotated a total of 39 different cell types on a total of 1.1 million nuclei, including aberrant cell types expected in IPF.

They report substantial changes in cell type composition with disease, and disease associated histopathology, largely in line with previous studies. For transcript niche identification the authors used graph neural networks without cell segmentation. In addition, the authors also used cell-based niche identification. Whilst most of the analysis seems to be based on published methods (GraphSAGE) it was nice to see these two complementary analyses. Having identified these niches, the authors were then able to compare cell type abundances across niches and made some surprising findings: For example, fibrotic foci were not associated with aberrant basaloid cells, but aberrant basaloid cells were associated with hyperplastic aerocytes and epithelial detachment. Aberrant basaloid cells were also found in association with SFTPC+/SCBG3A2 remodelled airspaces as expected. Honeycombing was associated with mixed airway and alveolar cell types. As expected, fibroblasts were most abundant in regions associated with severe fibrosis and fibroblastic foci. Overall, their analysis gives a very detailed picture of cellular landscape of the remodelled IPF lung, with different niches capturing specific interaction between different cell types.

Overall, the findings are well supported by the presented data, and the computational pipeline looks sensible to me. However, it is striking that two other recent paper preprints concluded that aberrant basaloid cells are part of the fibroblastic foci/fibrotic niche. The preprints (<https://www.biorxiv.org/content/10.1101/2023.12.21.572330v1>; <https://www.biorxiv.org/content/10.1101/2023.12.13.571464v1>) use lower resolution methods (Visium) so in that sense these are not contradictory findings. Nevertheless, I feel the authors need to comment on this and either provide some additional validation, for example absence of KRT17 staining in fibrotic foci, or possibly refer to other studies that may already be able to provide evidence that is in line with their conclusions.

We agree with the reviewer that the localization of aberrant basaloid cells is a point that warrants more clarity. Our initial analysis demonstrated that aberrant basaloid cells were often associated with a pro-fibrotic niche (originally niche C7 now C3). This niche marked both KRT5-/KRT17+ cells as well as activated fibroblasts. However, we have expanded our analysis to include a formal test of cellular adjacency (Supplemental Figures 9, 27, and 28) which support our initial interpretation that these cells are found in terminal airway/alveolar regions, often but not exclusively adjacent to fibrotic fibroblasts, but not always directly involving fibroblast foci. We suspect there are two reasons for the somewhat nuanced range of findings across studies: 1) As the reviewer notes, the lower-

resolution methods make it more difficult to resolve these cell types, particularly in complex regions with many cell types, and 2) Our dataset includes not only severely remodeled regions, but also areas of less “end-stage” pathology which is distinct from what appears presented in these other reports. We thus suggest these findings are supplementary to and clarifying of the results presented in the referenced preprints using lower-resolution methods. Finally, we have contextualized our results within these two studies, which were initially published after or during our initial submission.

The novel observation that aberrant basaloid cells are associated with epithelial detachment is already well documented in the figures. (e.g. Fig4e-g)

We thank the reviewer for their assessment of this analysis.

Furthermore, the authors use their data to show that different macrophage subtypes are associated with different niches , a very interesting finding; although there is some overlap of niches.

We again appreciate the reviewer’s interest in our findings.

Lastly, the authors identified lumens likely to have originated from alveoli and then ordered these airspaces based on cell type composition, from most normal to most disease, thus creating a pseudotime of disease progression. In a sense check this aligned with a progression from unaffected to low to more fibrosis. This allowed the authors to define early and late events in disease progression. Placing macrophages into this timeline showed that FABP4 macrophage were present in pseudotime before SPP1+macrophages, which appear to be associated with late events.

Comment: As part of this analysis the authors show a differential gene expression analysis Fig 6d. could the authors comment on the pathways that are differentially expressed, rather than just genes?

We agree with the reviewer that conceptually this is an attractive idea, although this particular dataset is unfortunately not well-suited for these sorts of analyses as a large proportion of the 343 genes are lineage/identity markers (rather than genes that change dynamically in response to pathway activity). We are hopeful that we will be able to perform these sorts of analyses in the future using larger probe-sets and/or alternative methods.

Overall, this study suggests that disruption of the alveolar epithelium and adjacent capillary network is observed first, before fibrotic remodelling, and that activated fibroblasts and infiltration by different subsets of macrophages are late events. These are original results, they are clearly presented and supported by the data.

We appreciate the reviewers comments on our findings and their assessment that our results are well supported by the data.

All of this already makes a very nice paper, but it is a shame that the authors did not use their rich data to examine cell-cell interaction networks. They beautifully define niches with distinct cell populations, that follow a pseudotime. Would it not be possible to use the matching single cell data to make predictions about the most likely interactions that are disease associated and likely to drive the pathologic process?

Similar to the above comment, we strongly agree with the reviewer that this is conceptually an exciting idea. The image-based spatial data presented here include a limited number of “communication” molecules, and we did attempt to perform ligand-receptor communication based analyses using these data directly, but found there was limited ability to interpret these findings because the number of ligand-receptor pairs was small. We perceive the reviewer’s comment as asking whether it would be possible to use the spatial data to “impute” location of cells for which disaggregated scRNA-seq is available, and use these “deeper” datasets for more comprehensive differential expression and communication analyses. This is also a very exciting idea, and one we too have considered. We have explored this conceptually, but were disappointed to find that with this specific probe-set, the number of different genes detectable in a given cell is insufficient to capture the high-dimensional variance required for high-fidelity “integration”/“imputation” with existing methods. In short, with this specific dataset, we do not think the performance of the “imputation” is sufficient for this sort of analysis to yield robust and high-confidence findings, but we are enthusiastic about this approach overall and look forward to developing better and larger probe-sets and/or using other modalities that may be better suited for this sort of analysis.

Reviewer #2:

Remarks to the Author:

General Comments

This is an interesting manuscript that describes initial efforts to use the Xenium platform for spatial transcriptomics on sections from normal lungs and the lungs of patients with interstitial lung diseases. The authors used a standard Xenium package including probes for 246 “lung” genes together with 97 genes chosen by the investigators to evaluate 21 samples from 12 patients with pulmonary fibrosis (7 with IPF and 5 from patients with a variety of other fibrosing interstitial lung disease) and 7 from individual normal lungs. For each sample the authors performed spatial transcriptomics followed by staining with Hematoxylin and Eosin. They then used a variety of bioinformatic approaches to identify cell types/states of interest that had been previously described based on single cell RNA sequencing and the relationships between those cells and areas of specific pathology identified by lung pathologists. Data linking multiple recently identified disease-associated cell states to specific areas of distinct pathology could fill an important current knowledge gap. Furthermore, spatial understanding of how and where these distinct cell states interact could generate new hypotheses about the steps underlying the development of pulmonary fibrosis, an important but currently poorly treated set of diseases with considerable morbidity and mortality.

However, there are a number of shortcomings to the current manuscript that limit the impact of the work described. Importantly, the authors don't provide sufficient detail for readers to understand how they decided what to call each cell and how well their calling method behaved. Most of the figures that show spatial data directly are shown at insufficient magnification for readers to evaluate co-expression of multiple genes (or even two genes) by any single cell. Dot blots in the supplemental figures clearly show that most of the genes analyzed (as expected) were detected in multiple cell types. Somewhat surprisingly, many of the genes that would be expected to be restricted to one broad cell category (like CD3 that should be restricted to T cells, and PDGFRA that should be restricted to fibroblasts) were listed as detected in cell types that have never been shown to express these genes by scRNAseq. Some of these problems might be due to acknowledged limitations of the 10x method for computational cell segmentation, but it would be important for the manuscript to show examples of how these errors might have occurred and what steps the authors took to mitigate them. Since this is the first study described utilizing their unique probe set, it would also be important to include high power images demonstrating cells that clearly co-express multiple genes that establish their unique identities.

Seurat software also allows users to generate uMAP plots from Xenium outputs which the authors say they used during an iterative process of re-clustering defined subsets of cells that the authors used to develop their system for identifying and counting individual cell states. Inclusion of these uMAPs would be very helpful for readers to evaluate how well the methods used in this paper really distinguish individual cell types or states. The one uMAP plot shown, in Figure 1, does not even seem to clearly distinguish among the major cell categories labeled (epithelial cells, endothelial cells, mesenchymal cells and immune cells).

Another limitation of this paper is that it does not seem to provide any novel insights into the biology underlying pulmonary fibrosis. Although the authors show examples (in figure 2) of 3 pathologic structures that are well known to occur in subsets of fibrotic diseases, fibroblastic foci, granulomas and tertiary lymphoid structures and also show examples of honeycombing, goblet cell metaplasia and hyperplastic “alveolar” epithelial cells, there are no new insights into the cellular composition or organization of these structures than those previously described by standard immunostaining. Actually, since the authors don’t show images of the primary data with patterns of co-expression in individual cells in these structures, this paper contains less molecular information than many previous descriptions. It was also a bit disappointing that the small lung regions analyzed in total seemed to include only two fibroblastic foci, two tertiary lymphoid structures and a few large granulomas.

The Seurat algorithms for identifying cellular niches don’t add much to this paper, since this approach mainly identified groups of cells that are all the same class of cell, rather than “niches” in which multiple cell types interact. Although the GraphSAGE generated transcript-based “niches” did not completely overlap with the cell-based niches, the tendency was similar for them to mostly identify homogeneous groups of cells. At the very least, it would have been important for the authors to try to quantitatively evaluate close physical relationships among different cell types or the “niches” they identify. There are well-established methods by which they could have done that.

The authors spend quite a bit of space in the results and discussion describing what they think might be a novel finding – the presence of KRT17+/KRT5- epithelial cells overlying areas of detachment of the epithelium from the underlying tissue. As above, it would be helpful for the authors to show high power images demonstrating the expression of KRT17, the absence of KRT5 and the presence of other markers of so-called “aberrant basal cells” in these regions. More important is the concern that the detachment might be an artifact caused by detachment of epithelium from areas of stiff underlying tissue during embedding, sectioning and processing of these samples. At the very least it would be important to characterize the cells and genes in the regions that these epithelial cells were likely in contact with before processing. In this regard, it is important to note that previous publications have shown enrichment of these cells in the epithelium overlying fibroblastic foci. In the current manuscript the authors are likely to have missed this event because they only sampled two fibroblastic foci, and the one shown in Figure 2 itself seems to be artifactually separated from the surrounding tissue. Cell proximity calculations might at least have allowed the authors to determine if aberrant basaloid cells were likely to be in close proximity to aggregates of activated fibroblasts, as has been shown by others. However, the limitations of the authors’ tools to distinguish distinct fibroblast subsets (the vast majority of fibroblasts were simply lumped together as “fibroblasts”) might limit their ability to perform this analysis.

We thank the reviewer here for a very thoughtful and in-depth consideration of the complex issues that warrant addressing. Most points are specifically addressed below. Briefly, we found the reviewer’s comments regarding the “atlas” aspects of the manuscript including rigor and fidelity of cell annotation to be very salient and we regret

that these were underdeveloped in the initial submission. To this end, we have added a new Figure (Figure 2) and numerous supplemental figures (Supplemental Figures 3-7) that provide a more comprehensive description of the data underlying cell-annotation. We also agree that the contamination/segmentation issues (while improved by use of Cellbound) are a substantive limitation; we have added text specifically addressing this point, and developed a QC filtering strategy for differential expression analyses to mitigate these impacts. We regret that the reviewer perceived that there were “no new insights” into PF biology. We suggest that the novel molecular characterization of aberrant basaloid cells detaching from their underlying basement membrane is an impactful observation (further supported by a preprint recently submitted by another group), and that the use of niche-based trajectory analysis provides integrative and innovative conceptual advances. Concerns regarding the number of features analyzed were recognized and we have supplemented pathologic feature annotation and increased our sample n by 50% to add robustness to these analyses. We appreciate the suggestion of more formal proximity analyses, and now include these in Supplementary Figures 9, 27, and 28.

Specific comments:

C1) Figure 1 – As noted above, the uMAP plot shown does not appear to show clear separation of even the major cell categories labeled on the figure. The authors should explain how they chose to label specific regions of the uMAP as specific cell types, along with uMAPs showing overlay of a number of genes of interest that should be specific to specific cell types, ideally including uMAPs based on sub-clustering of each major cell type. That would provide more confidence that the labels are accurate.

R1) We appreciate this comment, and regret that insufficient attention was devoted to these considerations in the initial submission. In the revised manuscript, we have 1) added new data from 17 additional samples, 2) Added new Figure 2 which provides details regarding spatial and transcriptomic validation of cell identification, 3) Added supplementary figures 3-7 which add details supporting fidelity of cell annotation at the lineage-level. With this refined approach and inclusion of additional data, we now resolve 47 distinct cell-types/states and provide specific spatial and marker-based data to support the robustness of these annotations. We have also, to the best of our ability, aligned our nomenclature with recent cell nomenclature working groups from the community. We hope that these additions and edits make this dataset clearer and more useful for readers and the lung biology community et-large.

C2) Figure 4 –It is hard to see gene expression in individual cells with the large spots used to mark individual genes on relatively low power images. It would be important to show individual cells at much higher power so readers can see individual amplified transcripts and their relationship with individual nuclei and calculated cell boundaries. It is difficult to know what to make of the finding of cell detachment since from the supplementary figures it looks like very

few regions of *KRT17+/KRT5-* cells were seen. As noted above, detachment might also be an artifact of fixation and sectioning.

R2) We thank the reviewer for their suggestion about visualization of markers – made both here and above. We’ve found this visualization difficult to implement at scale as finding a balance between showing multiple markers, not obscuring low expressed genes, and providing sufficient spatial context is non-trivial. We have expanded the number of representative images – both for the *KRT5-/KRT17+* findings and others (Fig. 5, Fig. 6, Fig. 7, Supplementary Fig. 6g,h, Supplementary Fig. 8, Supplementary Fig. 25, and Supplementary Fig. 26.), but acknowledge this is certainly not comprehensive. In addition to the main/supplementary figures we have we have released full “Xenium Explorer” files for all of the data from this manuscript on GEO. This enables anyone to download these data and explore the results at high resolution in an interactive setting. We have found this interactive analysis to be the best visualization of these results and this is now a freely available resource as part of our dataset.

Focusing specifically on the *KRT5-/KRT17+* findings (now Fig. 5), we have updated this figure to include *KRT5* and *MMP7*. Additionally, we included a new panel figure 5j to show a more comprehensive expression profile across annotated *KRT5-/KRT17+* cells. We have also added a supplementary figure with several additional examples of epithelial detachment, including *KRT5-/KRT17+* cells along with *KRT5+/KRT17+* and *SFTPC+/SCGB3A2+* cells (Supplementary Figure 23). More markers of *KRT5-/KRT17+* cells and activated fibrotic fibroblasts in areas of epithelial detachment are also shown on Xenium images that are matched to Visium HD data showing composite expression of multiple cell type markers, including some not present on the Xenium panel (Supplementary Figure 31).

C3) Figure 5 – The structures called alveoli don’t really look like alveoli and the SFTPC signal seems to be present in stratified epithelial cells surrounding the lumens shown. In IPF, many of the macrophages are in the interstitium, but the sections shown seem to only show large clusters of macrophages within large lumens. Why aren’t more than single genes shown for each cell type, to allow readers to get a better sense of how cell types were named?

R3) In line with comments above, we regret that key “atlas”-related considerations were initially underdeveloped, and have added new Figure 2 as well as several supplemental figures that provide more details regarding justification of cell-annotation. Supplementary Figure 3 presents a single dotplot of 122 key marker genes and their expression across all 47 cell-types. The reviewer astutely points out that we did not originally include images of *FABP4+* and *SPP1+* macrophages in the interstitium or minimally remodeled alveoli, instead focusing on the novel findings that macrophage accumulations in PF occur 1) in both highly remodeled alveoli and airways and 2) are both *FABP4+* and *SPP1+*. In the interest of transparency and thoroughness, we now include a representative image of interstitial and alveolar macrophages in the new Figure

6c. *AGER* (a marker of AT1 cells) is included in this new panel, though notably there is low expression of *AGER* even in these minimally remodeled alveoli.

The alveoli included in panels (d) and (e) express AT2 marker *SFTPC* and transitional AT2 markers *SCGB3A2* and *KRT8*; however, these are substantially remodeled and no longer maintain typical alveolar morphology on H&E stained images, as the reviewer notes. These notable differences between the minimally and substantially remodeled alveoli shown in Figure 5 are now clarified in the caption and main text. As noted by the reviewer, the bottom area of the *SPP1+* alveolar lumen in this figure does contain stratified epithelial cells expressing alveolar markers including *SFTPC*, which underscoring how cellular arrangement and spatial relations differ in substantially remodeled regions compared to a “healthy” state.

MARCO has also been added as an additional marker gene for this figure to indicate that the *FABP4+* and *SPP1+* cells within the alveolar and airway lumens are macrophages.

C4) S2 – the cell numbers shown suggest some significant biases in cell naming. For example, although AT2 cells and AT1 cells are present in equal numbers in normal lungs, the authors identified more than 4 x more AT2 cells. This could be because AT1 cells numbers are reduced in fibrotic lungs, but it would be reassuring to show equal numbers in normal lungs. Previous scRNA sequencing and immunostaining have shown a reduction in AT2 cells in fibrotic lungs, but this paper suggests there is a large increase. I wonder if that is because of spill-over of the *SFTPC* signal, much as the authors saw for the highly expressed AT1 marker *AGER* (which I presume they ignored in their naming process). For mesenchymal cells, from scRNAseq in normal lungs most fibroblasts are alveolar fibroblasts and in fibrotic lungs they are mostly “activated” fibroblasts, with the remaining fibroblasts divided among peribronchial and adventitial. However, the overwhelming majority of fibroblasts shown are identified as just “fibroblasts” suggesting that much of the heterogeneity of this population of cells was missed or underestimated with the probes used for this study. This point should at least be acknowledged and discussed.

R4) After resolving the issue with probe labeling and reprocessing our data with the additional 17 samples, the proportion of cells we identified as AT1 and AT2 are higher and lower, respectively, compared to the previous manuscript submission. We have also identified fibroblast subtypes with more granularity.

This specific question regarding AT2 vs. AT1 cell numbers and relative abundance in the human lung is one which the literature is not entirely consistent and we are uncertain as to the reliability of some historical estimates in light of recent single-cell labeling studies in mice that demonstrate individual AT1 and AT2 cells contribute to multiple alveoli. Nonetheless, historical studies suggest that AT1 and AT2 numbers are not exactly equivalent. In one electron-microscopy-based study of normal lungs, the AT2:AT1 ratio was ~2 (Crapo et al 1982, DOI: 10.1164/arrd.1982.126.2.332), while in another report, normal human alveoli were estimated to contain an average of 40 AT1 cells and 67 AT2 cells, which is an AT2/AT1 ratio of 1.675 (Stone et al. 1992; <https://doi.org/10.1165/ajrcmb/6.2.235>). Although we did identify more AT2 cells than AT1

cells in unaffected samples in the present study (AT2/AT1 ratio ~2.53), this is much closer to an expected ratio as that estimated from scRNA-seq of disaggregated cells (AT2/AT1 ratio ~13.51). However, we agree that we may not be detecting all AT1 cells that are present in the tissue due to issues with cell segmentation, particularly since AT1 cells are very thin, elongated and irregularly shaped, and their associated nuclei may not be in the same imaging plane. This information is now included in the main text, with data presented on AT2/AT1 ratios for unaffected and PF samples in Figure 2b. Consistent with previous study we find a reduction in the proportion of AT2 cells found in fibrotic lungs across sample groups (Supplementary Fig. 12) and with increasing percent pathology (Supplementary Fig. 13).

During our updated analysis of these data we performed more granular annotation of fibroblasts, which included better aligning of our fibroblast naming conventions with the various lung cell nomenclature working groups.

C5) S3 - many of the cell types listed seem to have largely overlapping patterns of expression of the 70 genes shown within each broad cell type, so it would be important for the authors to show sections demonstrating with arrows how they distinguished among endothelial subsets, fibroblast subsets, epithelial subsets and lymphoid subsets. For example, differences were very subtle between peribronchial fibroblasts, fibroblasts and alveolar fibroblasts. Lumican was the only marker distinct for adventitial fibroblasts. It is not clear why proliferating mesenchymal cells aren't called proliferating fibroblasts, since they don't resemble mesothelial cells or SMCs at all. Where are pericytes? Did the authors not include any distinguishing markers?

R5) As noted in responses to prior comments, new Figure 2 as well as Supplementary Figures 3-7 provide considerably more detail regarding the marker gene expression and anatomic localization of different cell types. Some of the overlapping patterns were related to the probeset issue discussed above, and we acknowledge that the segmentation-related issues raised for AT1 cells are also relevant in fibroblasts. That said, working within the constraints of the probe-set that is not comprehensive, we suggest that the anatomic localization and marker gene expression for the subtypes of fibroblasts is well-aligned with descriptions in recent single-cell atlas and other publications, and address the comments raised about limited distinctions. For example, Adventitial Fibroblasts are PI16+ and MFAP5+. Subpleural fibroblasts express WT1 and HAS1; Myofibroblasts are WNT5A+; Fibrotic Fibroblasts are CTHRC1+/FAP+; Inflammatory Fibroblasts express CCL2 and SOD2. In the new cell type annotation, we label proliferating mesenchymal cells as "Proliferating FBs" because they primarily express fibroblast markers.

In the previous submission, pericyte marker *CSPG4* was incorrectly labeled as *APLNR*, leading to difficulty annotating this cell type. Though the gene label issue has been resolved, SMCs and pericytes could not be reliably subclustered based on the markers in this probeset into populations we felt confident were distinctly pericytes and SMC's, thus we categorized together as "SMCs/Pericytes". We agree with the reviewer that it would be preferable to be able to refine this, but we suspect an expanded panel of distinguishing markers would be needed.

C6) S4 – only shows expression of individual genes described as showing specific fibroblasts in panels b and c and double positive cells in panel a. However, at the resolution shown it is not possible to tell whether and if the cells are actually double positive and there are no other markers included to show that the cells positive for WNT5A or PI16 were actually fibroblasts. While PI16 is largely restricted to fibroblasts, the heat maps shown in supplementary figures suggest that WNT5A might be more broadly expressed. With 383 markers used, it would be better for the authors to show better examples of how they integrated multiple markers to identify each cell type they say they can distinguish. Some uMAPs for subclusters of each major cell type showing how good the distinctions were and how the markers that characterized each cluster allowed separation would be especially helpful.

R6) For Supplementary Figure 4 (now Supplementary Figure 8), zoomed-in images for panels (a-c) have been added for better resolution of the example areas. For panel (a), zoomed-in images show *SCGB3A2+/SCGB1A1+*, *SCGB3A2+/SCGB1A1-*, and *SCGB3A2-/SCGB1A1+* cells in airways and *SFTPC+/SCGB3A2+* cells in alveoli. For panels (b) and (c), new marker genes have also been added to demonstrate that the *WNT5A+* cells in panel (b) are myofibroblasts (*ACTA2*, *COL1A1*) and that the *PI16+* cells in panel (c) are adventitial fibroblasts (*MFAP5*, *COL1A1*) which are in close proximity to vasculature, but distinct from the smooth muscle cells (*ACTA2*) that directly surround vasculature.

C7) S6 – seems to show that there are no significant cell composition differences between normal and severe fibrosis. The only significant difference marked were for slides selected from regions thought to include mild fibrosis. Do the authors have any hypothesis for why this was the case? Did they not statistically compare severe fibrosis to normal?

R7) In the previous version of the manuscript, results were only reported for significant ANOVAs across the 3 groups. The ANOVAs with significant P-values were marked in the middle of the plot for each cell type, which we regret was visually confusing. To remedy this, we have revised the plot (now Supplementary Figure 12) to show significance for ANOVAs directly next to the cell type names and clarified the boxplot caption. Additionally, we now report results for post-hoc t-tests comparing groups (Unaffected vs. Less Affected, Less Affected vs. More Affected, and Unaffected vs. More Affected) in Supplementary Table 4.5. As would be anticipated, the majority of cell-types showed differential abundance across conditions.

C8) S8 legend says *s represent significance, but I couldn't see any *s on any panel in the figure.

R8) We think the reviewer may be referring to a different figure. Figure S8 was a volcano plot and no such text was referred to in the legend. Nevertheless, we have reviewed all figures with an eye on ensuring proper significance markers are included.

C9) S9 – It is not clear what to make of these volcano plots. In nearly every cell type, AGER, which is mainly expressed in AT1 cells, appears to be upregulated in the normal lung in most cell types. It looks like PPARG, DUOX1, RTKN2 and MUC5B, at least, have similar patterns of non-specific expression. KRT17 shows up high in FABP2 macs in fibrosis but is not shown in KRT5-/KRT17+ epithelial cells, where the only gene highlighted in fibrosis is CXCL9, a chemokine mainly expressed by myeloid cells and fibroblasts.

R9) The limitations of cell-segmentation and assignment of transcripts to the “correct” cell impact this sort of analysis, and we share the reviewer’s sense that some of these findings were likely “contamination” of transcripts from nearby cells. Further, several of these observations were related to the probe-set issue outlined above. In the revised manuscript, we have adjusted our approach to include more robust filtering of low level expression contamination before performing this analysis by first scaling the gene counts using a count-per-1K normalization across cells and only keeping genes with at least 5 counts in 30% of cells within a cell type. The new results are presented in Supplementary Figure 15 and Supplementary Table 8.

C10) S10 seems to confirm that very few nuclei were analyzed from fibroblastic foci, and areas of cellular detachment. Severe fibrosis, granulomas, areas of honeycombing and mixed inflammation and normal small airways and other structures predominated. Although there were only two TLS, many cells were analyzed from these two. One sample seems to have been composed entirely of a TLS and another TLS came from one other sample. It looks like both fibroblastic foci might have come from a single sample. It looks like the granulomas, that were extensively analyzed, came from just two samples, with one made up of more than 50% granulomas.

R10) Our initial pathology annotations were deliberately sparse, with the goal to demonstrate that computational niche analyses can be more thorough and less time-intensive than annotations performed by a clinician. However, we acknowledge that an inherent trade-off of that approach was a limited dataset, with low quantity and diversity of these annotations across samples. We have now added more annotations to the samples previously included in the manuscript along with those for the 17 newly-added samples. There are now 712 representative annotations across 27 features.

C11) S11a seems to show that many of the annotated histologic features were not well separated from one another by gene expression, with considerable overlap among many of the groups although I may have misread this since some of the colors were not easy to distinguish. The two exceptions seem to be granulomas and to a lesser extent, severe fibrosis.

R11) We have updated the color scheme – as well as included expression values from the newly annotated pathology features. There is now obvious separation between many features. However, there are still a number of features which have partially overlapping gene expression. We believe this is expected based on prior knowledge and the findings

presented in Figure 3c, which demonstrates a varying degree of cellular overlap between discrete features.

C12) S11b seems to show ITGB6, an epithelial restricted gene, and COL1A2, a fibroblast gene, as markers of histopathologic regions of airway smooth muscle. The figure also seems to confirm that only two fibroblastic foci and two tertiary lymphoid structures were annotated from the entire dataset.

R12) Indeed, ITGB6 and COL1A2 were both mislabeled probes in the original analysis (actually representing expression of FN1 and ACTA2 respectively). As mentioned above we now have a much deeper representation of annotations across the samples, including 7 TLS and 64 fibroblastic foci.

C13) S11c shows high levels of PDGFRA and ACTA2 in what are called normal alveolar epithelial cells and high levels of MUC5AC in what are called normal alveolar epithelium. The figure also shows MRC1 (expressed mainly in macrophages) and MSLN (mainly in mesothelial cells), in what are called hyperplastic aecs and epithelial cells in areas of detachment and CD68 (a macrophage marker) in normal alveolar epithelial cells.

R13) These confusing localizations were again a result of mislabeled probes. What was labeled ACTA2 was actually MRC1, what was labeled PDGFRA was actually ITGAX, what was labeled MRC1 was COL4A3. Looking at single cell RNA-sequencing data MSLN is expressed at moderate levels in AT1 and other epithelial populations, and CD68 is likely marking alveolar macrophages. We also wish to clarify that the “normal alveoli” histological annotation intentionally includes all of the cells that make up the alveoli, not just epithelium (see Figure 3c).

C14) S19 – Shows that cell identities were either unreliable or assignment of expressed genes to specific cells was inaccurate. This figure includes numerous examples of patterns that are dramatically different from expression patterns shown from public RNAseq databases including: PDGFRA and PDGFRb – very low in mesenchymal cells, high in all myeloid cells and some lymphocytes

CD4 in most mesenchymal cells and some myeloid cells

Col3a1 in lymphocytes

Krt6 in a population of mesenchymal cells and most lymphoid cells

Muc5ac high in some mesenchymal cells

Acta2 highest in all myeloid cells, not in mesenchymal cells, even smooth muscle

Fgf2 – mostly in myeloid cells

Itgb6 mostly in most mesenchymal cells and some myeloid cells

WT1 most highly expressed in myeloid cells and some lymphocytes.

R14) Most of these issues were driven by the mislabeling of genes, and the corrected labels align much better with results expected based on prior knowledge and scRNA-seq datasets. To the reviewer’s specific genes highlighted:

**What was labeled PDGFRA is actually ITGAX.
What was labeled PDGFRB is actually LYZ.
What was labeled as MUC5AC is actually PPARG.
What was labeled ACTA2 is actually MRC1.
What was labeled FGF2 is actually PLIN2.
What was labeled ITGB6 is actually FN1.
What was labeled WT1 is actually TREM2.**

Examining the remaining genes, and reprocessing samples with inclusion of new data and correct gene labeled, we observed much cleaner expression distributions. The results (now presented in Supplementary Fig. 31) show COL3A1 is almost exclusively in Mesenchymal cells (corroborated by Supplementary Fig. 3), KRT6A is almost exclusively within epithelial cells with some expression in Mesothelial cells (consistent with previous reports). We observe CD4 to be most highly expressed in lymphoid populations, but observe more modest expression in myeloid populations as well. This is consistent with results from prior scRNA-seq studies.

C15) S13 WNT5a seems to be expressed at low levels in 6 of 12 cellular niches. Yet figure S4 uses this single gene to suggest that subepithelial fibroblasts are distributed around conducting airways. The T3 niche, which is described in the text as the aberrant basaloid cell niche, contains only a single lane (sample?) that is KRT17 high and KRT5 low. All of the other lanes in this niche seem to express equal levels of KRT17 and KRT5. How are these data consistent with the idea that this niche represents aberrant basaloid cells?

R15) We thank the reviewer for pointing out the single gene annotation in Supplementary Fig. 4. We have now added multiple genes to this figure to better demonstrate our findings (now Supplementary Fig. 8). We suggest that the observation that for some cell-types, key marker genes may be found in multiple “niches” - WNT5A is such an example. We suggest this is part of the insight that comes from these sorts of analyses - certain cells/states may be quite niche-specific, while others are found in a variety of niches. Regarding the T3 niche, while this niche contains the vast majority of KRT5-/KRT17+ cells, these niches are not defined by a single cell type - i.e. this niche does not represent aberrant basaloid cells, but rather represents a molecular process that is often associated with KRT5-/KRT17+ cells. As shown in Figure 4b and Supplementary Figure 22 this niche contains both KRT5-/KRT17+ and basal cells. We suggest this niche is associated with more broad distal airspace remodeling and also includes a high proportion of RASC cells. Importantly, while this niche contains the majority of KRT5-/KRT17+ cells, there are still a small number of these cells overall.

C16) Methods – the authors described iterative rounds of clustering and sub-clustering using marker genes including analyzing uMAP outputs to refine clusters as well as analysis of cell size and shape. They should show uMAP plots validating the final clusters since those would help

readers to evaluate the validity of the annotation methods, ideally with overlay of multiple well-characterized genes described to characterize clusters previously identified by scRNAseq.

R16) We have included additional UMAPs and expanded the sections describing our cell type annotation in greater detail, particularly in Supplemental Figures 4-7. These are discussed extensively above.

C17) How did the authors decide that alveolar lumens could have up to 500 cells? These spaces seem too large to be defined as alveoli.

R17) We agree with the reviewer that a normal alveolus is unlikely to contain 500 cells. Our objective in the airspace-segmentation analyses was to capture both normal alveoli and airspaces of alveolar origins that may be already extensively remodeled. We observed that many of these remodeled airspaces included large numbers of inflammatory cells (largely macrophage accumulations) which drastically increases the number of cells per “alveolus”. We have now added this reasoning to the Methods section.

Reviewer #3:

Remarks to the Author:

Vannan et al used 97 genes from Haberman et al.'s (PMID: 32832598) plus a pre-existing probe set to measure gene expression in 19 lungs, both control and pulmonary fibrosis. The results as the authors themselves point out are circumscribed, because of the approach of using a small number of markers that were previously discovered, to at best a validation of cell types suggested by the previous scRNAseq work, now with spatial information added. Therefore, while the work complements the published scRNAseq data in terms of spatial information, I am concerned that it in any event merely recognizes structures that are well known to the lung pathology of fibrosis, such as fibroblastic foci and epithelial hyperplasia, etc.; not much was discovered by the study, nor were there surprises. Finally, I'm not sure how this will be a genetic resource, given the limited gene number. A hypothesis-driven gene set would at least uncover the activity of a given pathway or pathways, whereas the number of markers used here, and their apparent limitation to known overrepresented markers in fibrosis globally, limits depth.

We appreciate the reviewer's comments here, and in general agree that there are competing considerations regarding breadth and depth of analyses possible with these methodologies that rely upon a fixed number of potential analytes. We also recognize that there is a tradeoff between panel size and sensitivity; in pilot work performed during revision, we found that per-gene sensitivity using the 10X Xenium 5K (5000 gene) panel performed considerably worse (30-90% worse) than what we observed with this targeted panel, and using 8 μm binning, the Visium HD platform has even lower per-gene sensitivity. Thus, it is clear to us that current technologies are powerful, but each has limitations that inform how they can be best deployed. We attempted to design this panel to strike a balance of sufficient number of markers for reliable cell-type identification, while retaining some "disease biology" genes to enable spatial contextualization of disease activity. With that in mind, we agree this study was semi-targeted and informed by prior knowledge, and find the reviewer's idea of "pathway" targeted probe sets an intriguing future direction.

Acknowledging these tradeoffs (which we think it is important for the field to be clear-eyed about), this dataset is the largest and most pathologically diverse lung spatial transcriptomic dataset we are aware of, and the only single cell resolution spatial atlas of healthy and fibrotic tissue, enabling high-resolution characterization of evolving cellular makeup, spatial relations, and expression profiles. Beyond the "atlas" related features (which are expanded in this revised manuscript in response to reviewer comments), we suggest that defining the cellular makeup of specific pathologic features informs understanding of disease biology, and comprehensive quantitative characterization of cellular relations and how they evolve during disease would not be possible with other methods. Further, we are enthusiastic about the potential of cell-agnostic "niche" analysis methods as a means to understand "functional pathology", and we offer a novel approach to explore the trajectory of pathologic remodeling. Overall, we agree with the reviewer that some of the impact of this manuscript is integrating and contextualizing

prior work, but we suggest that there are both new biological insights and methodologic advances that we hope are useful for the field.

Some specific queries:

C1) Sloughing of the epithelium: Even if this is not artifactual, it is not clear what was learned from reporting this phenomenon, which is anyway observable by standard H&E and as the authors mentioned, previously reported. Is there something about the sloughing that reveals new aspects of the pathophysiology, or is there a clue to its cause?

R1) We too shared the reviewer’s concern that this could be an artifact, although the observation that this feature was rather consistently characterized by a specific epithelial cell state (aberrant basaloid/KRT5-/KRT17+ cells), is found across multiple donors and generally at specific anatomic locations lends some confidence that this may occur in-vivo. We hesitate to speculate too extensively in the manuscript text, but have expanded the text in the discussion, to suggest this could explain why alveoli are “missing” in PF lungs - fusion of opposing basement membranes exposed by sloughing, and or create a physical barrier that precludes mucociliary transport and enables accumulation of “honeycomb cysts” – features with key biological relevance to IPF progression. We also note another preprint reported similar findings recently (bioRxiv 2024.06.17.599411). This feature is predominantly found where KRT5-/KRT17+ cells are adjacent to activated/fibrotic fibroblasts where MMP and other secreted proteases involved in matrix remodeling could disrupt interactions between adherence molecules and the underlying basement membrane and one could envision that this creates “space” for underlying fibrosis to progress. We fully recognize that additional evidence is necessary to fully validate these processes, however results from our study suggest the concept that warrants further investigation.

C2) Airway macrophages: I am not clear on how the airway-localized macrophages could be important for progression of fibrosis in the parenchyma.

R2) In light of extensive prior work in the field investigating the role of macrophages in lung injury-repair and fibrosis, we suggest that a role of macrophages in disease biology is reasonably well established albeit certainly incompletely understood. We also wish to highlight that our findings suggest this is not an early feature of remodeling alveoli, but becomes evident in more remodeled areas, and we acknowledge that assertions about causality cannot be made from these sorts of data. A complete mechanistic interrogation of these cells is beyond the scope of this study, but we suggest that reciprocal feedback and regulation between airspace macrophages, the underlying epithelium, and adjacent fibroblasts is a concept that has previously been proposed, and is supported by paracrine signaling mechanisms. As noted below, we acknowledge that this probe-set has limited ability to assess these cell-communication questions.

C3) Disease heterogeneity: The sample contains IPF and non-IPF fibrosis. The authors should focus more analyses on the differences between these subgroups globally and with a focus on immune cell types and locations.

R3) We agree with the reviewer that these methods offer an opportunity to explore disease heterogeneity. In light of the variety of diagnoses and small numbers per group of non-IPF samples, we have limited power to explore specific subgroup-related differences, however we are excited to pursue these sorts of analyses in greater depth using larger datasets in the future. We did perform exploratory compositional analyses as suggested by the reviewer - somewhat surprisingly, if any trend emerged there was more lymphocytic inflammation in IPF samples compared to other ILD samples, although sample-to-sample variability was considerable and in general there were more inflammatory cells in more severely disease samples.

C4) The pseudotime ordering provides little context for the reasons for the evolution within airways. Are there ligand-receptor pairs that could be interrogated by other methods to understand this progression, if the Xenium markers used are insufficient for this analysis? Can the data shed light on capillary dropout? The loss of interstitial macrophages in favor of later SPP1+ macrophages is based on markers, but it is not clear whether the cells that were there all along—the macrophages—had a physiological role in fibrosis progression but merely began to express the putative profibrotic marker SPP1 “late” in the pseudotime trajectory.

R4) With the incorporation of new data and correction of the probe-set issues, this pseudotime analysis has been extensively revised. As noted above, this specific panel

has a relatively limited number of ligand-receptor pairs and is unfortunately not well-suited for these sorts of analyses. To explore this further, we performed “full transcriptomic” spatial transcriptomic profiling using the VisiumHD platform on a small set of samples with the hope that this would enable more “interaction” analyses. Unfortunately, the per-cell sensitivity of that platform is lower than Xenium, and resulted almost entirely in “binary” scenarios of either 0 or 1 count of a feature per-cell; this implied that there was a stochasticity to detection of ligand-receptor pairs, but also meant that there was no “dynamic range” that could be detected to quantify differential communication. This was disappointing but appears to be a fundamental limitation of the methodology.

The trajectory analysis supports a general model of “phases” of remodeling, with the earliest events involving loss of AT1 cells and capillaries; later there is increased abundance of inflammatory cells and changes in fibroblast phenotypes, while extensive cell-identity changes in the epithelium are observed later. The SPP1+ macrophages generally express at least low-levels of CD14, which is consistent with findings in scRNA-seq datasets, and would imply a recruited (monocyte-derived) origin rather than a phenotype shift of resident alveolar macrophages. We found that expression of CCL2, for example, is highest in “late remodeling” which would fit with a model of “late” recruitment. The specific pathologic role of SPP1 is a topic beyond the scope of this study, however we agree with the reviewer that the expression of SPP1 may result from the local environment, although a recent manuscript suggested a direct profibrotic role as well (PMID 39129313).